# The proteolysis of ZP proteins is essential to control cell membrane structure and integrity of developing tracheal tubes in *Drosophila*

Leonard Drees[1], Susi Schneider[2], Dietmar Riedel[3], Reinhard Schuh[1], Matthias Behr[2]*

[1]Research Group Molecular Organogenesis, Department of Molecular Developmental Biology, Max Planck Institute for Multidisciplinary Sciences, Göttingen, Germany; [2]Cell biology, Institute for Biology, Leipzig University, Leipzig, Germany; [3]Facility for electron microscopy, Max Planck Institute for Multidisciplinary Sciences, Göttingen, Germany

*For correspondence:
matthias.behr@uni-leipzig.de

Competing interest: The authors declare that no competing interests exist.

**Abstract** Membrane expansion integrates multiple forces to mediate precise tube growth and network formation. Defects lead to deformations, as found in diseases such as polycystic kidney diseases, aortic aneurysms, stenosis, and tortuosity. We identified a mechanism of sensing and responding to the membrane-driven expansion of tracheal tubes. The apical membrane is anchored to the apical extracellular matrix (aECM) and causes expansion forces that elongate the tracheal tubes. The aECM provides a mechanical tension that balances the resulting expansion forces, with Dumpy being an elastic molecule that modulates the mechanical stress on the matrix during tracheal tube expansion. We show in *Drosophila* that the zona pellucida (ZP) domain protein Piopio interacts and cooperates with the ZP protein Dumpy at tracheal cells. To resist shear stresses which arise during tube expansion, Piopio undergoes ectodomain shedding by the Matriptase homolog Notopleural, which releases Piopio-Dumpy-mediated linkages between membranes and extracellular matrix. Failure of this process leads to deformations of the apical membrane, tears the apical matrix, and impairs tubular network function. We also show conserved ectodomain shedding of the human TGFβ type III receptor by Notopleural and the human Matriptase, providing novel findings for in-depth analysis of diseases caused by cell and tube shape changes.

## Editor's evaluation

Using the *Drosophila* tracheal system as a model for apical membrane expansion of tubes, the authors convincingly demonstrate that ectodomain shedding of the Zona Pellucida (ZP) domain protein Piopio by the Matriptase homolog Notopleural releases its linkage with the ZP protein Dumpy and thus ensures proper apical membrane function and tube expansion. Given the high degree of conservation of these proteins in other species, the results presented are important for future analysis and will have further implications on tubular development and homoestasis in other systems, including human.

## Introduction

Tube networks are essential for organisms transporting liquids, gases, or cells across bodies. Endothelial and epithelial cells generate such networks with strict hierarchical order and precise tube dimensions

as a prerequisite for proper tube network functioning (*Ochoa-Espinosa and Affolter, 2012*; *Potente and Mäkinen, 2017*). Defective cell shapes and tube dimensions result in severe syndromes such as chronic obstructive pulmonary diseases (384 million people in 2010) with high global mortality as well as tube dysfunctions like aortic aneurysms (152,000 deaths worldwide) in blood vessel systems and polycystic kidney diseases (1 in 1000 people) (*Adeloye et al., 2015*; *Barnes et al., 2015*; *Harris and Torres, 2009*; *Quintana and Taylor, 2019*; *Zhai et al., 2021*). One fundamental question regarding the formation of functional tube systems is how cells balance forces at the cell membranes emerging during organ growth and the simultaneous maintenance of the tubular network integrity. *Drosophila melanogaster* embryos form a tracheal system, an excellent model for studying molecular mechanisms controlling cell expansion in combination with tube elongation.

The development of tracheal tubes is subject to precise genetic control. Genes that control cell polarity, junction formation at the lateral membranes, cytoskeletal organization, and intracellular trafficking of tube size determinants support apical cell membrane expansion (*Behr et al., 2003*; *Dong et al., 2014*; *Förster and Luschnig, 2012*; *Laprise et al., 2010*; *McSharry and Beitel, 2019*; *Nelson et al., 2012*; *Olivares-Castiñeira and Llimargas, 2017*; *Skouloudaki et al., 2019*; *Syed et al., 2012*; *Tonning et al., 2005*; *Tsarouhas et al., 2007*). In contrast, genes controlling the meshwork of chitinous apical extracellular matrix (aECM) formation restrict excessive cell expansion to prevent tube overexpansion (*Luschnig et al., 2006*; *Moussian et al., 2006*; *Öztürk-Çolak et al., 2016*; *Petkau et al., 2012*; *Tiklová et al., 2013*; *Wang et al., 2006*). Subsequently, genetically controlled mechanisms establish tracheal airway clearance, aeration, and tube stabilization (*Behr et al., 2007*; *Behr and Riedel, 2020*; *Drees et al., 2019*; *Stümpges and Behr, 2011*; *Tsarouhas et al., 2007*; *Tsarouhas et al., 2019*).

Previous studies revealed that axial and radial forces affect tracheal tube elongation. The apical membrane grows axially, pulling on the associated aECM until the aECM's elastic resistance balances the elongation force throughout the tubes (*Dong and Hayashi, 2015*). Given the association between cell membranes and aECM, ongoing tube expansion and luminal shear stresses inevitably lead to problematic membrane tension. Once forces are out of equilibrium, tracheal tubes show curvy appearance. Similarly, also blood vessels can appear unstable, twist, kink, and buckle. High stresses even lead to vascular damage and aneurysm rupture (*Dong et al., 2014*; *Han et al., 2013*). However, it is not known how cells manage to integrate the axial forces to stabilize the cell membrane and aECM.

Zona pellucida (ZP) domain proteins are critical components of apical cell membranes and aECM (*Plaza et al., 2010*) and assemble into extracellular fibrillar polymers (*Jovine et al., 2005*; *Litscher and Wassarman, 2020*). For example, Uromodulin plays a role in chronic kidney diseases and hypertension (*Rampoldi et al., 2011*). Secreted Uromodulin requires proteolysis at the apical cell membrane for shedding and polymerization within the tube lumen (*Brunati et al., 2015*). Similarly, the ZP domain protein Piopio (Pio) (*Figure 1—figure supplement 1A*) is secreted into the tracheal tubes of *Drosophila* embryos (*Grieder et al., 2008*; *Massarwa et al., 2009*). Pio restricts the elongation of autocellular junctions (*Jaźwińska et al., 2003*). Further, Pio is involved in relocating microtubule organizing center components γ-TuRC (γ-tubulin and Grips, gamma-tubulin ring proteins). This requires Spastin-mediated release from the centrosome and Pio-mediated γ-TuRC anchoring in the apical membrane (*Brodu et al., 2010*). However, while Pio is a transmembrane protein, it was detected in the tube lumen (*Jaźwińska et al., 2003*), but the release mechanism remains unknown. A promising candidate for Pio proteolysis is Notopleural (Np), the functional homolog of the human Matriptase. It is a type II single-transmembrane serine protease in tracheal and lung epithelia and is capable of ectodomain shedding (*Bugge et al., 2009*). Initial in vitro studies prove that Np cleaves the Pio ZP domain (*Drees et al., 2019*). The elastic luminal matrix is essential for the integrity of the tubular network. During tube elongation, the matrix balances elongation forces in the anterior-posterior direction (*Dong et al., 2014*). Here, we show that the ZP domain proteins Pio and Dumpy and the protease Np respond to mechanical stresses when tracheal tubes elongate to ensure normal membrane-aECM morphology.

## Results

### Pio maintains structural cell membrane continuity

The tracheal lumen matrix consists of a viscoelastic material that is coupled to the apical membrane. The precise balance between apical membrane growth and luminal matrix resistance determines tube shape (*Dong et al., 2014*). Based on these observations, we expect the following scenario: first, apical membrane growth and opposing restriction by the extracellular matrix produce increased tensile stress during tube expansion of stage 16 embryos (*Figure 1A*). Second, increasing tension impacts membrane-matrix couplings, which provide the proper balance between both. Thus, enhanced tensile stress may lead to either release or remodeling of the membrane-matrix couplings to avoid potential deformations. The Pio protein contains a transmembrane and an extracellular ZP domain, suggesting that it may link tracheal cells to the aECM. Using CRISPR/Cas9, we generated three *pio* lack of function alleles (*Figure 1—figure supplement 1*), we analyzed the two independent alleles *pio^{5m}* and *pio^{17c}* which showed embryonic lethality and identical tracheal mutant phenotypes. The tracheal phenotypes of *pio^{5m}* are shown in the supplement (*Figure 1—figure supplement 1B–F*). In all other figures, we show images of the *pio^{17c}* allele. The *pio^{17c}* and *pio^{5m}* null mutant embryos revealed the dorsal and ventral branch disintegration phenotype known from a previously described *pio^{2R-16}* mutation allele which contains an X-ray induced single point mutation that led to an amino acid replacement (V159D) in the ZP domain (*Jaźwińska et al., 2003*). In addition, late-stage 16 *pio^{17c}* and *pio^{5m}* null mutant embryos showed over-elongated tracheal dorsal trunk tubes (see below) in contrast to the *pio^{2R-16}* mutation. We compared the dorsal trunk morphology between control and *pio* mutant embryos by using the septate junction (SJs) marker Megatrachea (Mega) (*Behr et al., 2003*). The early stage 17 control embryos revealed tight appearance of tracheal cells and adjacent luminal extracellular matrix. In contrast, the corresponding *pio* mutant embryos showed irregular bulge-like gaps between the Mega-marked cell membrane and apical matrix (*Figure 1B*). Such gaps were not detectable in late-stage 15 wild-type (*wt*) or *pio* mutant embryos (*Figure 1—figure supplement 2A*). This suggests that the gaps arise in stage 16 *pio* mutant embryos during tube length expansion.

To study the role of Pio during tube length expansion, we examined *pio* mutant stage 16 embryos using the apical cell membrane marker Uif. This revealed unusual apical cell membrane deformations most prominent at the dorsal trunk (*Figure 1C*). The corresponding control embryos of the same fixations and staining did not show membrane deformations (*Figure 1C*). Additional analysis of orthogonal projections of confocal Z-stacks revealed straight apical cell membrane of dorsal tracheal trunks in the control embryos while comparable *pio* mutants contained numerous small membrane deformations along the dorsal trunk (*Figure 1C'*). Moreover, these membrane deformations may compromise the shape of the dorsal trunk tube lumen in *pio* mutant. The control dorsal trunk tubes showed straight and uniform lumens in three-dimensional reconstructions (*Figure 1C*). The corresponding orthogonal projections confirmed the appearance of a ring-like structures of the apical cell membrane (*Figure 1C*), which was detected with Uninflatable (Uif) (*Zhang and Ward, 2009*) and with the cell surface marker WGA (wheat germ agglutinin), indicating the tube lumen formation in the control embryos. In contrast, projections of the *pio* mutant dorsal trunk tubes showed bulge-like deformations that appeared as small protrusions reaching into the tracheal cells in the corresponding orthogonal projections (*Figure 1C and C'*).

Furthermore, the orthogonal projections of the *pio* mutant dorsal tubes did not show straight but unusually narrow and collapsed lumen at the site of apical membrane deformations (*Figure 1C'*; *Figure 1—figure supplement 1D*). The additional ultrastructural analysis confirmed the appearance of apical cell membrane deformations at the dorsal trunks of stage 17 *pio* mutant embryos. The TEM study revealed unusual gaps between the membrane and aECM, while control embryos lacked such gaps. These gaps appear like membrane bulges that protrude into the cells (*Figure 1D and E*; *Figure 1—figure supplement 2B*). These results indicate that Pio is required to stabilize or maintain structural membrane-matrix formation.

Anti-Pio antibody (kindly provided by Markus Affolter) detects a short stretch within the Pio ZP domain (*Jaźwińska et al., 2003*). Immunostainings confirmed Pio protein expression in the tracheal system of stage 16 embryos when tubes expand (*Figure 2A*). In these embryos, tracheal Pio staining is detectable at the membrane and is enriched within the tracheal lumen, consistent with previous findings (*Dong et al., 2014*; *Massarwa et al., 2009*). At the membrane of stage 16 control embryos, Pio overlaps at discrete points with the apical cell membrane determinant Crumbs (Crb) and WGA

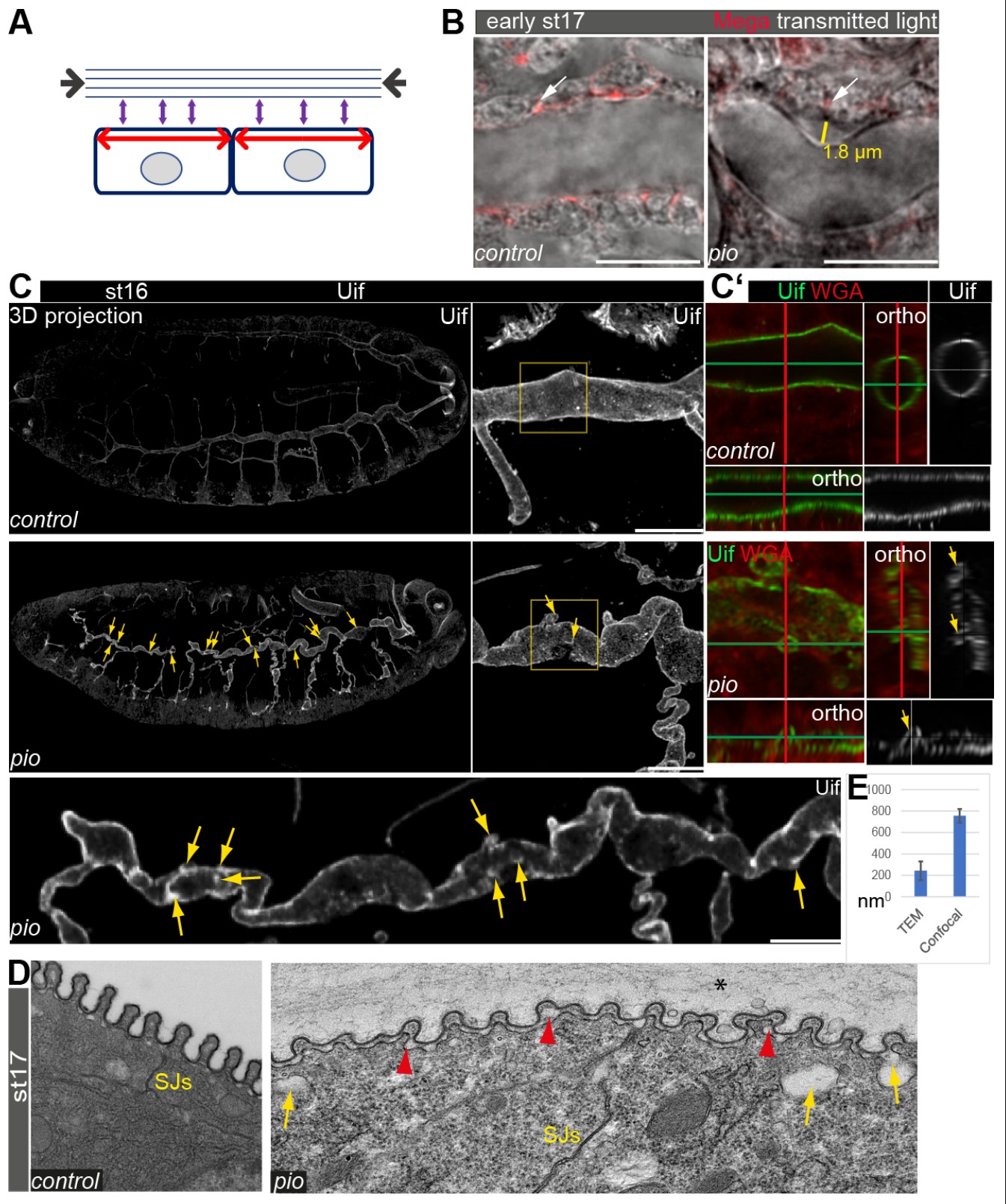

**Figure 1.** Pio supports structural continuity of the apical cell membrane. (**A**) Model implicates the axial and longitudinal forces (arrows) acting apical on cell membrane and extracellular matrix of stage (st) 16 embryos when tracheal lumen expands in growing tubes. (**B**) Confocal images of *wt* and *pio* mutant stage 17 embryos. Lateral membrane is marked by Megatrachea (Mega; red, arrows) immunostaining, and transmission light visualizes the tracheal cells and lumen. The yellow line indicates the distance between the apical cell surface and the detached luminal apical extracellular matrix

*Figure 1 continued on next page*

*Figure 1 continued*

(aECM). Scale bars represent 10 μm. (**C, C'**) Confocal LSM Z-stack (overview) and Airyscan (close up) images of immunostainings are displayed as maximum intensity (3D projection) and orthogonal (ortho) projections (**C'**) using Uif antibody and WGA. Scale bars indicate 10 μm. Stage 16 control embryo showed straight apical cell membrane and tracheal tubes. All *pio* null mutant (n=10) embryos revealed curly elongated tracheal tubes and unusual bulge-like apical cell membrane deformations (yellow arrows in overview and close-up images). The Uif (green) and WGA (red) stainings on the right panel show airyscan images of the region, which is marked by the yellow frame in close left. Position of membrane deformations are marked by the green and red lines (XY axes) with the ZEN orthogonal projection. The corresponding orthogonal projections are indicated. Note that membrane bulge-like structures interfere with the tube lumen integrity (ortho-target cross in mutant). Control and *pio* mutant embryos were fixed and stained together. (**D**) TEM analysis of late-stage 17 *wt* embryos reveal SJs and chitin-rich taenidial folds and a cleared tube lumen. The corresponding *pio* mutant embryos (n=4) showed normal SJs formation but unusual apical cell membrane bulge-like deformations (yellow arrow), reduced chitin (red arrowheads), taenidial folds with disorganized pattern, and unusual extracellular matrix contents within the tube lumen (*). Scale bars represent 500 nm. (**E**) Quantification of *pio* mutant bulge-like apical cell membrane deformations sizes in nm of airyscan Z-stacks (mean value 750 nm, n=15) and TEM (mean value 230 nm, n=61) images. Standard deviations are indicated. It is of note that measurements of TEM images do not always capture the three-dimensionality of bulges and may show only parts of them.

The online version of this article includes the following source data and figure supplement(s) for figure 1:

**Source data 1.** Bulge-like gaps between the Mega-marked cell membrane and apical matrix in stage 17 embryos.

**Source data 2.** Bulge-like gaps between the Mega-marked cell membrane and apical matrix in stage 17 embryos.

**Source data 3.** Uif marked unusual apical cell membrane deformations at the dorsal trunk in stage 16 embryos and quantification.

**Figure supplement 1.** Generation and ultrastructure analysis of *pio* mutant embryos.

**Figure supplement 2.** Cell membrane deformations in *pio* mutant embryos.

---

that stains the cell membranes (*Figure 2—figure supplement 1A and B*). Similarly, Z-stack projections and fluorescence intensity profile analysis in stage 16 control embryos showed overlapping staining of discrete Pio puncta with Uif at the apical cell membrane (*Figure 2C–E*; *Figure 2—figure supplement 2A and B*). In *crb* mutant embryos, the apical membrane is compromised (*Laprise et al., 2010*), but luminal content is still secreted (*Olivares-Castiñeira and Llimargas, 2018*; *Stümpges and Behr, 2011*). These *crb* mutant embryos lacked Pio staining at the membrane, and instead, Pio concentrated within the tube lumen. Control embryos showed normal Pio distribution at the membrane (*Figure 2B*).

These observations prompted us to address whether Pio misdistribution depends on apical cell membrane organization. Crb genetically interacts with *wurst* on late airway maturation, including gas-filling (*Stümpges and Behr, 2011*). The transmembrane protein Wurst (DNJAC22) is a critical component of clathrin-mediated endocytosis (*Behr et al., 2007*) and controls the internalization of proteins at the apical membrane (*Stümpges and Behr, 2011*). Stage 16 *wurst* mutant embryos and tracheal-specific wurst RNAi (interference) knockdown embryos revealed unusually increased Pio accumulation at the apical cell membrane compared to control embryos (*Figure 2A and B*; *Figure 2—figure supplement 1B and C*).

These findings suggest Wurst-mediated internalization of Pio and raise the question of how intracellular Pio trafficking may occur. Retromer and ESCRT-mediated endosomal sorting regulate essential proteins of tracheal tube size control (*Dong et al., 2014*). Pio is not affected by the retrograde transport from endosomes to the trans-Golgi network (*Dong et al., 2013*). Vps32/Shrub is a subunit of ESCRTIII, which regulates endocytic sorting of membrane-associated proteins leading to lysosomal degradation and is known to be involved in tracheal tube size control in stage 16 embryos (*Dong et al., 2014*). Loss of *shrub* leads to the formation of swollen endosomes that accumulate Crb within tracheal cells (*Dong et al., 2014*). We observed intracellular Crb staining that overlapped with Pio in *shrub* mutant embryos (*Figure 2—figure supplement 1A*). Additionally, Crb and Pio overlapped at the apical cell membrane, suggesting that newly synthesized Pio was secreted as it is known for Crb (*Dong et al., 2014*). These results indicate that Pio localization relies on apical membrane formation, turnover, and intracellular protein trafficking.

To understand the distribution of Pio under different stress situations, we examined mutants that either increase the apical cell surface or cause severe chitin defects. We generated voxel data of 3D Pio staining projections (Imaris, 3D surface rendering) using Airyscan Z-stacks and deconvolution (SVI Huygens). These data identified Pio voxel in punctuate pattern at and next to the apical cell membrane in *wt* embryos. However, most voxel accumulated around the inner luminal chitin matrix structure (*Figure 2C*; *Figure 2—figure supplement 2A*), resembling the confocal pattern of control embryos (*Figure 2A and B*). The disruption of SJs in *mega* mutant embryos caused apical cell surface

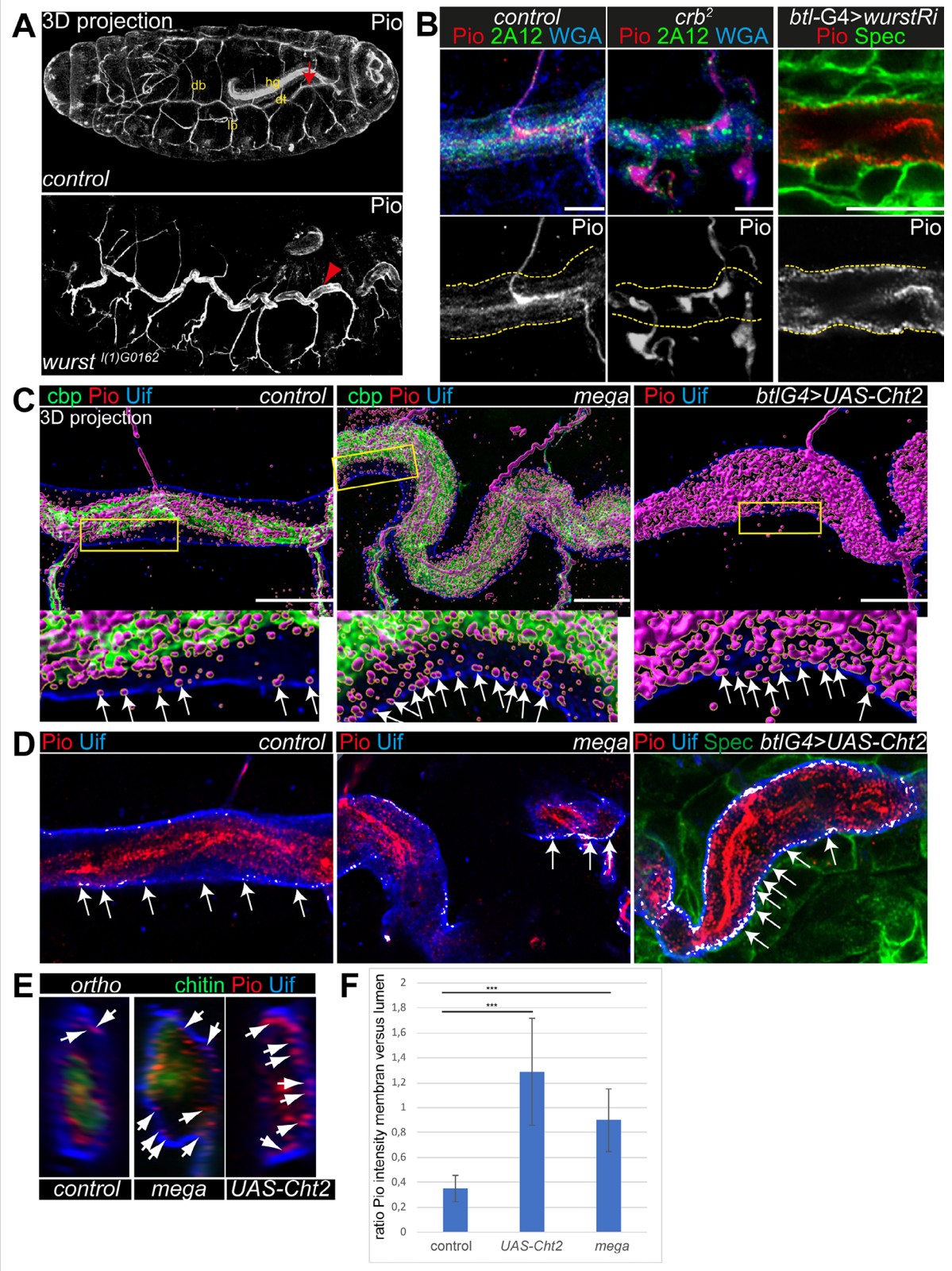

**Figure 2.** Pio localization depends on the apical membrane and supports tracheal air-filling. Shown are stage 16 embryos. Maximum intensity projections of confocal Z-stacks are shown in A, single confocal images in B, and Airyscan microscopy in C–E. Scale bars indicate 10 μm. (**A**) Pio protein is expressed in tracheal tubes (db, dorsal branches; lb, lateral branches; dt, dorsal trunk) and other ectodermal epithelial organs. Pio accumulates in the tracheal lumen (arrow) of control embryos. In contrast, Pio staining showed unusual accumulation at the apical cell membrane of *wurst* mutant

*Figure 2 continued on next page*

*Figure 2 continued*

embryos (arrowheads point to Pio accumulation). It is of note that embryos were stained together. Shown are whole-mount embryos. hg, hindgut. (**B**) In *crb* mutant embryos, Pio staining is within the luminal matrix but not at the apical cell membrane (indicated by yellow dashes). Corresponding control embryos showed Pio at the membrane and predominantly within the lumen. The tracheal-specific *wurst* knockdown shows Pio accumulation at the apical cell membrane similar to the *wurst* mutant embryo (compare with A). 2A12 detects the apical extracellular matrix (aECM) protein Gasp. WGA stains cell membrane surfaces and chitin predominantly. (**C**) Imaris 3D projection. In wt Pio (magenta) is detectable in a punctuate pattern at the tracheal apical cell membrane, partially overlapping with Uif (blue) in stage 16 control embryos. Chitin-binding probe (cbp; green) labels chitin. The stage 16 *mega* mutant embryo (n=5) showed Pio accumulation at the apical cell surface. The tracheal expression of the Chitinase 2 (n=5) showed a disturbed Pio pattern, including accumulation of Pio puncta at the apical cell surface. The lower panel shows close-ups of the apical cell membrane of the framed area in the upper images. The white arrows indicate Pio puncta at the Uif marked apical cell membrane. We chose comparable regions where a gap formed between the cell membrane and chitin matrix to detect the apical Pio puncta. Note that *mega* mutant embryo (20 Pio puncta) contains twice as much Pio puncta at the Uif stained apical cell membrane as the control (9 Pio puncta); both show tracheal metamers 7–9. (**D**) ZEN co-localization, which compares histograms of fluorescence intensities between the two channels, airyscan images. The overlapping Pio (red) and Uif (blue) puncta are colored in white. Arrows point to such overlapping Uif and Pio puncta at the apical cell membrane. (**E**) Images show representative orthogonal projections of dorsal trunks (metamere 6–8) of control, *mega* mutant, and Cht2 overexpression embryos. Arrows point to Pio puncta (red) at and near the apical cell membrane marked with Uif (blue). Chitin is shown in green. (**F**) Quantification of the ratio of maximum fluorescence intensity values between membranes and tube lumen. The control embryos showed a ration at 0.35, the Cht2 overexpression at 1.29, and *mega* mutants a ration of at 0.9. Bars represent ± SD and p-values from t-test are indicated with asterisks (***, p<0.0001).

The online version of this article includes the following source data and figure supplement(s) for figure 2:

**Source data 1.** Quantification of Pio distribution across the dorsal trunks in different stress situations in stage 16 embryos.

**Figure supplement 1.** Tube luminal matrix and Pio turnover in tracheal cells.

**Figure supplement 2.** Pio localization in *mega* mutant and Cht2 overexpressing embryos.

---

expansion and increased tube length in stage 16 embryos (*Behr et al., 2003*). These *mega* mutant embryos showed Pio localization at the chitin matrix but increased staining at and near the apical cell membrane (*Figure 2C*; *Figure 2—figure supplement 2A*). The ectopic expression of the *chitinase 2 (cht2)* in tracheal cells leads to excessive tube dilation due to strongly reduced luminal chitin (*Behr et al., 2003*; *Petkau et al., 2012*; *Tonning et al., 2005*). Also, the tracheal *cht2* expression led to increased Pio staining at and near the apical cell membrane (*Figure 2C*; *Figure 2—figure supplement 2A*). Additional Z-stack analysis methods support such a shift of Pio localization. The ZEN co-localization tool and orthogonal projections across the tube lumen showed enriched Pio puncta at the membrane of *mega* mutant and Cht2 overexpressing tracheal cells compared with control (*Figure 2D and E*).

Further, the fluorescence intensity profile confirmed that fluorescence peaks of Pio antibody staining overlapped with peaks of Uif staining at the apical cell membrane and with the chitin-binding probe (cbp) at the luminal chitin cable (*Figure 2—figure supplement 2B–D*). The statistical analysis of the fluorescence intensity profiles of Pio, Uif, and cbp confirmed a significant difference between control, *mega* mutant, and Cht2 overexpression (*Figure 2F*). Together, these findings provide evidence that Pio staining is distributed in a punctuate pattern at the apical cell membrane in stage 16 embryos, which elongate tracheal tube length. Furthermore, the genetic modifications of the apical membrane and chitin matrix morphology affect the Pio pattern at the cell membrane.

Next, we investigated if Pio affects the tracheal airway function. First, orthogonal projections of confocal Z-stacks of late-stage 16 *pio* mutant embryos revealed that membrane deformation compromised the shape of the tube lumen, which includes deformations that protrude into cells, which constricted the lumen (*Figure 1C and C'*; *Figure 1—figure supplement 1D*). Second, the ultrastructure analysis of stage 17 *pio* mutant embryos revealed, in addition to the membrane protrusions, reduced chitin of taenidial folds at the aECM of dorsal trunks while control embryos established chitin-loaded taenidial folds at the apical cell surface (*Figure 1D*; *Figure 1—figure supplement 2B*). Also, confocal microscopy revealed reduced tracheal chitin staining in stage 16 *pio* null mutant embryos (see chitin stainings below). Third, *pio* mutant embryos showed an irregular pattern of taenidial folds in TEM and airyscan analysis, while control embryos showed a regular and narrower spacing of taenidial folds (*Figures 1D and 3A*; *Figure 1—figure supplements 1D and 2B*). Fourth, ultrastructure TEM images revealed aECM remnants in the airway lumen of *pio* mutant stage 17 embryos, while control embryos cleared their airways (*Figure 1—figure supplement 2B*). Consistently, the in vivo analysis of airways in stage 17 *pio* mutant embryos revealed lack of tracheal air-filling (*Figure 3B*).

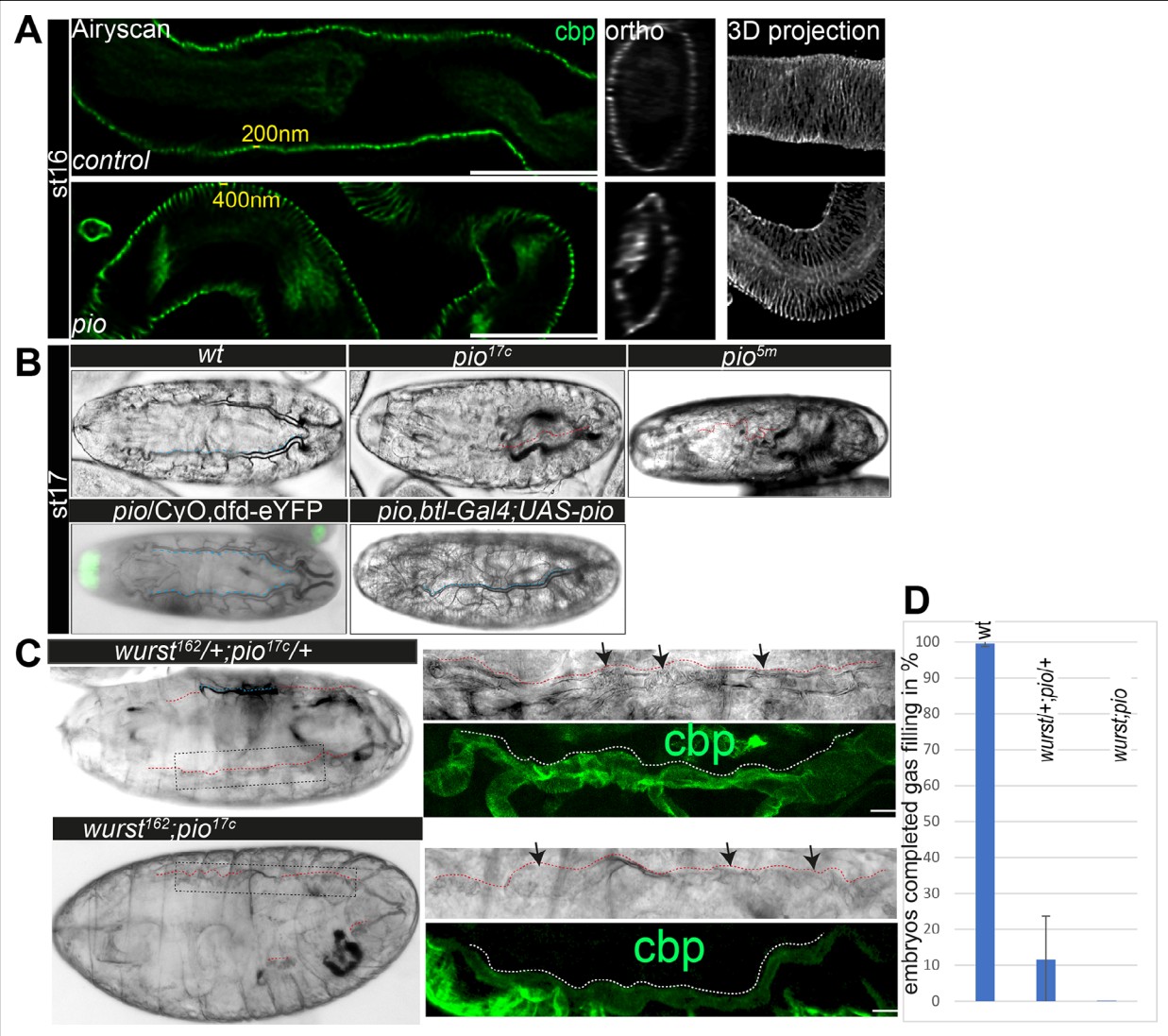

**Figure 3.** Pio is required for taenidial fold morphology and airway gas-filling. (**A**) Airyscan images (left) and orthogonal (middle) and 3D projections (right) of control and *pio* mutant late-stage 16 embryos. cbp (in green and gray) detects chitin at the taenidial folds and within the tracheal lumen of control and *pio* mutant embryos. The *pio* mutant embryos show loose taenidial fold patterns with enlarged distances between the ridges. (**B**) Late-stage 17 *wt* embryos revealed normal tracheal air-filling (indicated with blue dashes). All *pio* (n=20) mutant embryos mutant embryos showed tracheal air-filling defects (red dashes indicate liquid filled airways), while almost none of the control embryos showed defects (n=100). Tracheal air-filling defects in embryos were displayed as dark gray bars, and normal air-filling as light gray bars. Error bars indicate the standard deviation. Heterozygous *pio* mutant embryos (n=10) as well as the *btl-Gal4* driven tracheal expression of Pio in *pio* mutant embryos (n=10) revealed normal tracheal air-filling at the end of stage 17 (indicated with blue dashes). Green signal indicates eYFP expression from the balancer chromosome in the heterozygous *pio* mutant embryo. (**C**) The transheterozygous *wurst;pio* and *wurst;pio* double mutant stage 17 embryos showed tracheal air-filling defects (red dashes) in bright-field microscopy. Whole-mount embryos are shown left, framed parts of dorsal trunks are shown as close-ups right. Dorsal trunk tubes are indicated with red dashes. Confocal 3D projection of cbp (chitin, green) revealed collapsed and irregular tube lumen morphology in the mutant embryos. Scale bars indicate 10 μm. (**D**) Quantification of completed air-filling in late-stage 17 embryos. 89% of the transheterozygous (n=27) and all 11 *wurst;pio* double mutant embryos revealed incomplete air-filling whereas nearly all control embryos managed to complete air-filling. Error bars indicate the standard deviation.

The online version of this article includes the following source data and figure supplement(s) for figure 3:

**Source data 1.** Pattern of taenidial folds in stage 16 embryos.

**Source data 2.** Quantification of tracheal air-filling in stage 17 embryos.

**Figure supplement 1.** CRISPR-Cas9-mediated *pio* mutagenesis and caused phenotypes and mCherry:Pio generation and expression.

**Figure supplement 2.** Localization of chitin-matrix proteins in *pio* mutant embryos.

The pan-tracheal expression of Pio in *pio* mutant embryos rescued the lack of gas-filling (*Figure 3B*). Thus, TEM images suggest that *pio* mutant embryos showed impaired tube lumen clearance of aECM, which prevented subsequent airway gas-filling.

Since Pio localization depends on Wurst, we addressed putative genetic interaction by investigating tracheal air-filling in stage 17 transheterozygous *wurst* and *pio* mutant embryos. Heterozygous *wurst* and *pio* mutant control embryos showed normal tracheal air-filling (*Figure 3B*; *Behr et al., 2007*). In contrast, 88.4% of transheterozygous embryos bearing one copy of the *pio* and one of the *wurst* mutant alleles showed air-filling defects, accompanied by lethality before larval hatching (*Figure 3C and D*), suggesting genetic interaction of *pio* and *wurst* mutants. Consistently, all *wurst;pio* double mutant embryos failed to complete tracheal air-filling and are lethal as late-stage 17 embryos (*Figure 3C and D*). Moreover, higher resolution images of the not-gas-filled dorsal trunks of *wurst;pio* transheterozygous and double mutant stage 17 embryos revealed compromised tube lumen morphology accompanied by dilatations in bright-field microscopy and confocal analysis of chitin staining (*Figure 3C*). The *wurst* mutant embryos failed to clear luminal matrix content (*Behr et al., 2007*), including Pio, thus preventing airway clearance at stage 17 (*Figure 2—figure supplement 1D*). Also, the *pio* mutant embryos showed tracheal lumen clearance defects of chitin fibers in ultrastructure (TEM) analysis (*Figure 1D*, *Figure 1—figure supplement 2B*). In contrast, confocal analysis revealed that well-known chitin matrix proteins, such as Obstructor-A (Obst-A) and Knickkopf (Knk), are removed from the lumen of *pio* mutants (*Figure 3—figure supplement 1A*). These results suggest that the Pio function did not affect airway clearance of Obst-A and Knk and therefore did not play a central role in airway clearance like Wurst. Nevertheless, airway clearance defects observed in TEM images in *pio* null mutant embryos and, in addition, defective tube lumen morphology in *wurst;pio* transheterozygous mutant embryos explain the occurrence of airway gas-filling defects.

The chitin matrix is required in tracheal tube size control, taenidial fold formation, and tube lumen stability (*Behr and Riedel, 2020*; *Dong et al., 2014*; *Öztürk-Çolak et al., 2016*; *Tonning et al., 2005*). Essential chitin matrix proteins, such as chitin-binding protein Obst-A, chitin deacetylases Serpentine (Serp), and Vermiform (Verm), as well as chitin-protein Knk restrict tube expansion (*Luschnig et al., 2006*; *Moussian et al., 2006*; *Petkau et al., 2012*; *Wang et al., 2006*). In the *pio* mutant stage 16 embryos, luminal Obst-A, Knk, Serp, and Verm antibody staining showed *wt*-like distribution, but Serp and Verm appeared in reduced levels in the lumen (*Figure 3—figure supplement 2*). Thus, Pio does not control tracheal chitin-matrix secretion, formation, and organization but may affect their maintenance in stage 16 embryos.

Apical cell membrane growth is another essential cellular mechanism of tube growth (*Dong et al., 2014*; *Laprise et al., 2010*). However, the apical cell membrane marker Uif showed *wt*-like localization in *pio* mutant trachea (*Figure 1C*). Further, Crb immunostainings showed normal localization in *pio* mutant embryos (*Figure 4A*). Also, actin and tubulin cytoskeleton, which may impact apical cell membrane and tracheal tube formation, did not show gross differences between control and *pio* mutant stage 16 embryos. The apical cell cortex enriched tubulin and F-actin staining (*Figure 4—figure supplement 1*). Further, the confocal Mega immunostainings and appearance of SJs in ultrastructure analysis were similar between control and *pio* mutant embryos (*Figure 1B and D*; *Figure 1—figure supplement 2*). Finally, adherens junctions (AJs) revealed a *wt*-like appearance in *pio* mutant embryos in ultrastructure images (*Figure 1—figure supplement 2B*), and also the Armadillo (Arm; *Drosophila* β-Catenin) immunostainings showed *wt*-like pattern at the apicolateral membrane of tracheal cells (*Figure 4B*). These data indicate that Pio is not involved in either apical polarity or lateral membrane formation, which is consistent with previous findings (*Brodu et al., 2010*). However, we observed an increased distance in the axial direction between the AJs of the dorsal trunk fusion cells. In addition, we determined an enlarged apical cell surface area due to unusual cell elongation in the axial direction (*Figure 4B–D*). These findings indicate that Pio is required to prevent excessive apical cell growth and membrane deformation.

To understand Pio dynamics in living embryos, we generated CRISPR/Cas9-mediated homology-directed repair *pio*$^{mCherry::pio}$ embryos (*Figure 3—figure supplement 1B*). The mCherry::Pio expression in stage 16 embryos (*Figure 3—figure supplement 1C*) resembled Pio protein expression pattern in the tracheal system, digestive organs, epidermis, and spiracles (*Figure 2A*). Early stage 16 embryo dorsal trunk cells showed apical enrichment of mCherry::Pio (*Figure 5A*). Airyscan images and fluorescence intensity profiles of Z-stacks revealed a punctuate pattern of mCherry::Pio overlapping

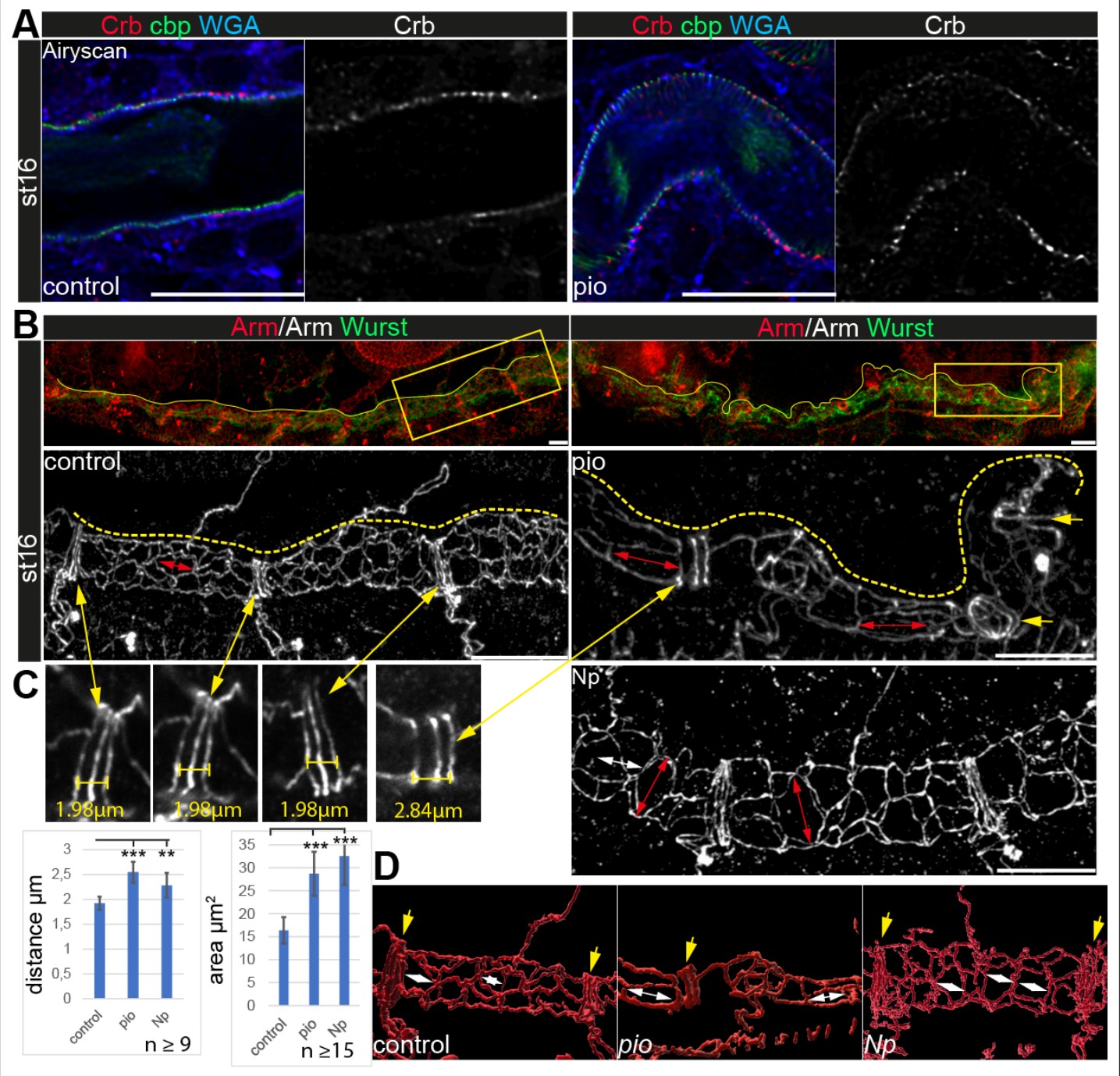

**Figure 4.** Apical polarity and AJs localization in *pio* and *Np* mutant embryos. Confocal LSM Z-stacks of tracheal dorsal trunk show single layer (**A**) and 3D projections (**B, C**) of stage 16 and 17 embryos. (**A**) Control and *pio* mutant late-stage 16 embryos show Crb (red) staining at the apical membrane, the cell surface marker WGA (blue), and cbp (green) in the tracheal apical extracellular matrix (aECM) at the apical cell surface and in the luminal cable-like ECM. A single Crb channel is indicated in gray. (**B**) Maximum intensity projections of confocal Z-stacks of control and *pio* mutant late-stage 16 embryos and *Np* mutant early stage 17 embryo. Upper panels show immunostainings with Armadillo (red) and Wurst (green) at the dorsal tracheal trunk. Yellow dashes mark the tracheal tube. Magnifications of the framed regions in the top panel show Armadillo staining in gray (bottom). Red double arrows indicate tracheal cells. Yellow arrows point to AJs of fusion cells. (**C**) Yellow double arrows point to magnifications of the Armadillo staining of dorsal trunk fusion cells. The distance of AJs of fusion cells in control, *pio*, and *Np* mutant embryos are indicated in representative images. Plots show AJ distances of fusion cells (n>9) in μm and apical cell area in μm² (n>15). Bars represent ± SD and p-values for AJs distance (*pio* p=5.8e-5; *Np* p=0.0022) and cell area (*pio* p=1.6e-6, *Np* p=2,5e-10), unpaired t-test. (**D**) 3D reconstruction (Imaris surface rendering) of confocal Armadillo immunostainings

*Figure 4 continued on next page*

*Figure 4 continued*

marking the AJs of control, *pio,* and *Np* mutant embryos. Yellow arrows point to AJs of fusion cells; white double arrows indicate cell length in the axial direction.

The online version of this article includes the following source data and figure supplement(s) for figure 4:

**Source data 1.** Pattern of adherens junctions in dorsal trunk fusion cells in stage 16 pio mutant embryo and quantifications.

**Source data 2.** Pattern of adherens junctions in dorsal trunk fusion cells in stage 16 Np mutant embryos.

**Source data 3.** Quantification of pattern of adherens junctions in dorsal trunk fusion cells.

**Figure supplement 1.** Tubulin and F-actin show normal distribution in pio null mutant embryos.

with chitin in the chitin cable and at the apical cell membrane surface where taenidial folds form in stage 16 embryos (*Figure 5—figure supplement 1A–C*). When tube expansions stopped at the end of stage 16, the tracheal mCherry::Pio signal shifted toward the lumen. This mCherry::Pio signal shift is distinct from the chitin matrix pattern (*Figure 5A*; *Figure 5—video 1*). Fluorescence recovery after photobleaching (FRAP) experiments demonstrated recovery of mCherry::Pio expression after photobleaching (*Figure 5B and B'*; *Figure 5—video 2*). In stage 16 embryos, mCherry::Pio puncta reappeared in tracheal cells within 2 min of bleaching and in the tubular lumen within 6 min. This demonstrates the dynamic mCherry::Pio relocation in tracheal cells and the lumen during tube expansion.

## Pio binds Dumpy to organize the luminal ZP protein matrix

Dumpy (Dpy) is a giant (3.2 mDa) and stretchable ZP domain protein. In stage 16 embryos *Dpy::eYFP* (*Lye et al., 2014*) appears at the tracheal apical cell surface and predominantly within the lumen (*Figure 5C*; *Dong et al., 2014*; *Jaźwińska et al., 2003*). Airyscan images and fluorescence intensity profiles of Z-stacks in stage 16 embryos revealed predominant *Dpy::eYFP* staining, which overlapped with chitin in the chitin cable. In addition, we observed a punctate *Dpy::eYFP* pattern at the apical cell surface overlapping with taenidial chitin and mCherry::Pio (*Figure 5—figure supplement 1D and E*). In contrast, the Dpy::eYFP signal predominantly remained intracellularly in *pio* mutants (*Figure 5C*), showing that Dpy secretion depends on Pio. We also performed cell culture experiments to extend our analysis of Dpy secretion. We generated constructs of RFP-tagged Dpy that lacked a portion of EGF and DPY repeats but contained the essential Dpy C-terminal region (ZPD domain, transmembrane domain, cytoplasmic region) (*Figure 5D*). Only the co-expression of RFP-tagged Dpy with FLAG-tagged Pio resulted in extracellular RFP::Dpy localization in the S2*R*+ cells (*Figure 5—figure supplement 2A*). Since extracellular RFP::Dpy is not released from the cells but overlaps with FLAG::Pio at the membrane, it suggests that they co-localize at the cell surface. The S2*R*+ cells expression products of RFP::Dpy constructs were only pulled down together with Strep::Pio in Strep-IP samples (*Figure 5D*). These data demonstrate that Pio co-localizes and interacts with Dpy. In contrast to Pio, tracheal Dpy::YFP was immobile in our FRAP experiments (*Figure 5B and B'*; *Figure 5—video 2*), supporting findings that the exchange of Dpy is negligible (*Dong et al., 2014*). Importantly, the current model suggests that the tracheal Dpy matrix stretches very likely by force applied from the cells during tube expansion and thus must be connected to the epithelium (*Dong et al., 2014*). Our FRAP data suggest that Pio is dynamic at the tracheal ZP matrix, while the static and stretchable Dpy modulates mechanical tension within the matrix, as discussed for Dpy previously (*Dong et al., 2014*; *Wilkin et al., 2000*).

Embryos of the lethal *dpy^olvR* allele showed *pio^2R-16* mutant-like tracheal branch disintegration (*Jaźwińska et al., 2003*) and additionally twisted and broken dorsal trunks in late-stage 17 embryos (*Bökel et al., 2005*). Also, previously shown quantification revealed a significantly increased dorsal trunk length in *dpy^olvR* mutant embryos (*Dong et al., 2014*). These findings suggest that also *dpy* mutation may impact tube expansion. Chitin staining of *dpy^olvR* mutants showed a sinusoidal appearance of tracheal dorsal trunk tubes and branch disintegration in stage 16 embryos (*Figure 5—figure supplement 3A*). The higher resolution airyscan images of Uif stainings at the dorsal trunks and three-dimensional projections revealed blister-like apical membrane deformations but less frequent as in the *pio* mutant embryos (*Figure 5—figure supplement 3B*). The orthogonal projections of such blisters in *dpy* mutant dorsal trunks revealed Uif-marked membrane protrusion into tracheal cells similar to *pio*-mutant embryos (*Figure 5—figure supplement 3C*). Additionally, we observed enriched puncta of

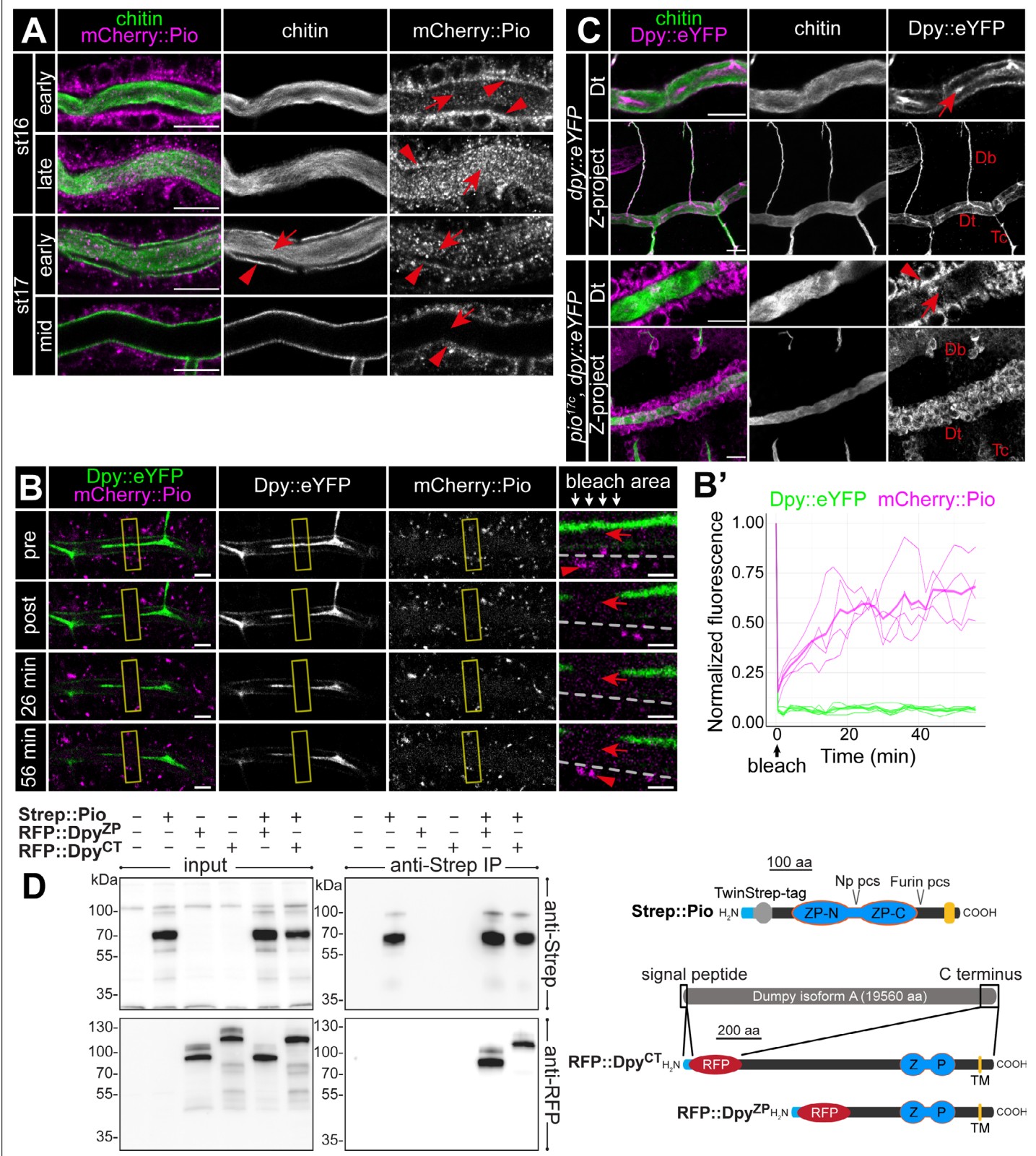

**Figure 5.** Pio is dynamically localized at the membrane and controls Dumpy secretion. (**A**) Confocal LSM images of dorsal trunks of embryos with endogenous mCherry:: Pio expression stained with anti-mCherry antibody (magenta) and cbp (chitin; green) at indicated embryonic stages. In early stage 16 embryos, mCherry:: Pio enriches apically (arrowheads) and is present in the lumen (arrow). In contrast, at the end of stage 16 mCherry::Pio predominantly localizes within the tracheal lumen (arrows). The luminal mCherry:: Pio staining disappeared during stage 17. (**B, B'**) Confocal images of

*Figure 5 continued on next page*

*Figure 5 continued*

a representative fluorescence recovery after photobleaching (FRAP) experiment (n=4) in a live embryo with endogenous expression of Dpy::eYFP and mCherry::Pio and quantification of normalized fluorescence in the bleached area of n=4 embryos (right) are shown. Yellow frames indicate the bleached area. Close-ups (right-most images) show details of the bleached area (below arrows in header). The dashed line indicates apical cell membranes, red arrows indicate luminal mCherry::Pio, and red arrowheads indicate intracellular or membrane-associated mCherry::Pio. A representative movie of an FRAP experiment is presented in Movie S2. Fluorescence intensities refer to the bleached regions of interest (ROIs) as indicated with the frame in corresponding Movie S2 and was measured after correction for embryonic movements. The mCherry::Pio (magenta) reveals recovery of small Pio puncta in the bleached area including the tracheal lumen, while Dpy::eYFP (green) shows no recovery even after 56 min. Scale bars indicate 5 μm in overview panels and 2 μm in bleach close-ups. (**C**) Confocal LSM images of endogenous expression of Dumpy:eYFP stained with anti-GFP antibody. The *wt*-like stage 16 control embryos show extracellular Dumpy:eYFP (magenta) in the apical extracellular matrix (aECM) at the cell surface and in the luminal cable (arrow) overlapping with cbp (chitin; green). In contrast, in *pio* mutant embryos Dumpy::eYFP did not overlap with chitin (cbp, green), but remained intracellularly (arrowhead). Upper rows focus on the dorsal trunk, lower rows show 3D maximum intensity projections of whole tracheal segments. Note the dorsal branch disruption known from hypomorphic *pio* point mutation allele (*Jaźwińska et al., 2003*). Single channels are indicated in gray. Db, dorsal branch; Dt, dorsal trunk; Tc, transverse connective. Scale bars indicate 10 μm. (**D**) Immunoblotting of co-immunoprecipitation (Co-IP) assay of RFP-tagged Dpy constructs and Strep-tagged Pio expressed in *Drosophila* S2R+ (Schneider) cells reveals binding of Dpy and Pio. Schemata of expressed proteins used in the assay are shown on the right. Strep::Pio is the full-length Pio protein with a Twin-Strep tag inserted C-terminal to the signal peptide (light blue). RFP::Dpy$^{ZP}$ and RFP::Dpy$^{CT}$ both contain the endogenous Dpy signal peptide (light blue) followed by mCherry (RFP) and different length of the C-terminal region of the Dpy isoform A protein as indicated. Transmembrane (TM) domains (yellow), ZP domains (blue), and Furin and Np protease cleavage sites (pcs) in Pio are indicated. Western blots of input cell lysates (left) and anti-Strep IP elutions (right) stained with anti-Strep (top) and anti-RFP (bottom) antibodies are shown. Both RFP::Dpy proteins are only detectable in IP elutions when they were co-expressed with Strep::Pio.

The online version of this article includes the following video, source data, and figure supplement(s) for figure 5:

**Source data 1.** Confocal Z-stack images of dorsal trunk showing mCherry::Pio expression during stages 16 and 17 of embryogenesis.

**Source data 2.** Confocal Z-stack images of dorsal trunk showing mCherry::Pio expression during late stages 16.

**Source data 3.** Confocal Z-stack images of dorsal trunk showing mCherry::Pio expression during early stage 17.

**Source data 4.** Confocal Z-stack images of dorsal trunk showing mCherry::Pio expression during mid stage 17.

**Source data 5.** Confocal Z-stack images of dorsal trunk showing Dpy::YFP expression in control and *pio* mutant embryos.

**Source data 6.** Confocal Z-stack images of dorsal trunk showing Dpy::YFP expression in pio mutant embryos.

**Source data 7.** Uncropped western blots.

**Figure supplement 1.** The mCherry::Pio shows localization at the apical cell surface and in the tube lumen.

**Figure supplement 2.** Pio is involved in Dpy secretion and Notopleural and Matriptase activity controls Pio localization.

**Figure supplement 3.** Tracheal phenotypes and Pio mislocalization in *dpy* mutant embryos Airyscan images (left), orthogonal (middle) and 3D projections (right) of of late-stage 16 *dpy*$^{olvr}$ mutant embryos (n=8).

**Figure 5—video 1.** Confocal time-lapse movie of endogenously expressed Dpy::eYFP (magenta) and mCherry::Pio (green) in *wt*.
https://elifesciences.org/articles/91079/figures#fig5video1

**Figure 5—video 2.** Representative confocal time-lapse movie of a fluorescence recovery after photobleaching (FRAP) experiment in a *wt* control embryo with endogenous expression of Dpy::eYFP (green) and mCherry::Pio (magenta).
https://elifesciences.org/articles/91079/figures#fig5video2

Pio staining in the Uif-marked membrane blisters (*Figure 5—figure supplement 3C*). The subcellular localization studies in stage 16 *dpy*$^{olvR}$ mutant embryos revealed a punctuate Pio pattern at the apical cell membrane overlapping with Uif but not within the chitin cable (*Figure 5—figure supplement 3D and E*). These findings suggest that *Dpy* is involved in Pio release at the apical cell surface and in apical cell membrane stability during tube length elongation.

## Pio release involves the serine protease Notopleural at the apical cell membrane

Furin-like enzymes cleave ZP precursors (*Jovine et al., 2005*). Pio contains a Furin **p**roteolytic **c**leavage **s**ite (Furin pcs) followed by a C-terminal transmembrane domain. Surprisingly, we detected in stage 16 embryo lysate three mCherry::Pio variants, one correlating with the predicted mass of a full-length protein (115 kDa), a second after furin site cleavage (90 kDa), and a third one correlating with the size after proteolysis within the ZP domain (60 kDa) (*Figure 6A*). The latter was even the predominant variant in stage 17 (*Figure 6A*). In contrast, the Np mutant embryo lysates of stages 16 and 17 contained only faint amounts of the 60 kDa mCherry::Pio and an increased level of the small variant

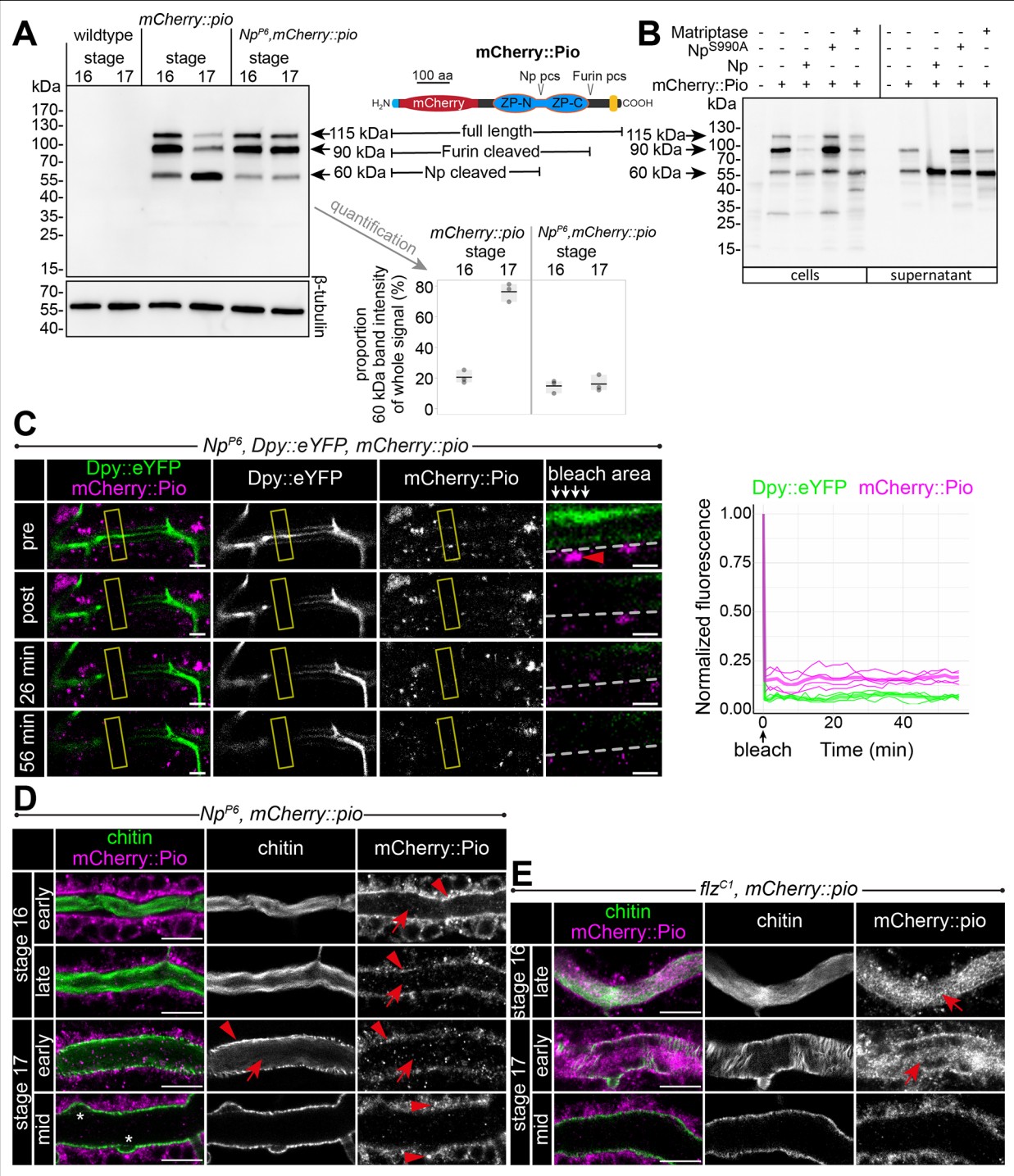

**Figure 6.** Np-mediated ZP domain shedding controls Pio dynamics at the apical cell membrane. (**A**) Left: Immunoblot of protein lysates from embryos of stages 16 and 17 stained with anti-mCherry antibody show three specific bands in samples from mCherry::Pio expressing embryos. Middle: Schematic presentation of the mCherry::Pio fusion protein with signal peptide (light blue), mCherry (red), ZP domain (blue), and transmembrane domain (yellow). Furin and Np protease cleavage sites (pcs) and expected molecular weights of resulting fragments are indicated. Right: Proportional intensity of the Np-cleaved 60 kDa mCherry::Pio fragment to the whole signal from all three mCherry::Pio fragments normalized to β-tubulin intensity. Data from three biological replicates show that proteolytic processing of mCherry::Pio at the Np cleavage site is highly increased in stage 17 embryos compared to stage 16 embryos in *wt* genetic background. This difference is not detectable in samples from *Np* mutant embryos. The endogenous Pio protein has a calculated mass of about 50.82 kDa. (**B**) Cleavage assay within the Pio ZP domain is mediated by proteolytic activity of Np and the human Matriptase. Immunoblotting of cell lysates and supernatant precipitates from *Drosophila* S2R+ cells expressing mCherry::Pio alone or together with Np, catalytically inactive Np$^{S990A}$ or human Matriptase with anti-mCherry antibody. Np and human Matriptase cleave mCherry::pio, causing shedding of

*Figure 6 continued on next page*

*Figure 6 continued*

the mCherry::Pio extracellular domain and a substantial increase of the 60 kDa mCherry::Pio, which correlates in size with cleavage at the ZP domain, in cell culture supernatants. This effect is not observable for catalytically inactive Np^S990A. (**C**) Confocal images of a representative fluorescence recovery after photobleaching (FRAP) experiment in a live *Np* mutant embryo with endogenous expression of Dpy::eYFP and mCherry::Pio and quantification of normalized fluorescence in the bleached area (yellow frames) of n=4 embryos (right) are shown. Close-ups (right-most images) show details of the bleached area (below arrows in header). The dashed line indicates apical cell membranes, arrowheads indicate intracellular or membrane-associated mCherry::Pio. Arrowhead points mCherry::Pio at the apical cell surface in the untreated area. Representative movie of the FRAP experiments is presented in *Figure 6—video 1* (*Np* mutant), compare with *Figure 5—video 2* (*wt*). The fast recovery of small mCherry::Pio puncta in the tracheal lumen is impeded in *Np* mutant embryos (compare with *wt* in *Figure 4D*). As in *wt* embryos, Dpy::eYFP (green) shows no recovery even after 56 min. Scale bars indicate 5 μm in overview panels and 2 μm in bleach close-ups. (**D**) Confocal images of tracheal dorsal trunks of *Np* mutant embryos with endogenous expression of mCherry::Pio at indicated developmental stages stained with cbp (chitin; green) and anti-mCherry antibody (magenta). Single channels are indicated in gray. Stage 16 *Np* mutant embryos show intracellular mCherry::Pio at the apical cell surface (arrowhead), which is similar to control embryos (see *Figure 4A*). In contrast to control embryos, the luminal mCherry::Pio (arrow) is strongly reduced in stage 16 and 17 *Np* mutant embryos, while the non-luminal mCherry::Pio accumulates in stage 17. The luminal chitin cable is degraded normally in *Np* mutant embryos but does not condense (*Drees et al., 2019*) during early stage 17 and instead, remains attached to the tracheal cell surface and fills the whole lumen during degradation (compare with control in *Figure 5A*). The asterisks mark bulges in *Np* mutant tubes. Note the two layers of chitin visible at the membrane bulges and the adjacent apical extracellular matrix (aECM) indicating disintegration of the tracheal chitinous aECM (see also Figure 6). Scale bars indicate 10 μm. (**E**) The tracheal trypsin-like S1A Serine transmembrane protease Filzig (Flz) shows high sequence homology to Np (*Drees et al., 2019*) and acts in processing the lumen matrix (*Rosa et al., 2018*). Confocal images of dorsal trunks of *flz* mutant embryos with endogenous expression of mCherry::Pio at indicated developmental stages stained with cbp (chitin; green) and anti-mCherry antibody (magenta). Single channels are indicated in gray. The *flz* mutant embryos revealed normal Pio expression, luminal shedding, and clearance from airways (compare with *wt* in *Figure 4A*). In contrast to *Np* mutant embryos, the luminal aECM cable condensed during early stage 17 luminal clearance (arrows) as in *wt* embryos (compare with *Np* mutant in D and *wt* in *Figure 4A* and **A**). Scale bars indicate 10 μm.

The online version of this article includes the following video, source data, and figure supplement(s) for figure 6:

**Source data 1.** Confocal Z-stack images of dorsal trunk showing mCherry::Pio expression in *Np* mutant embryos during stages 16 and 17.

**Source data 2.** Confocal Z-stack images of dorsal trunk showing mCherry::Pio expression in Np mutant late stage 16 embryo.

**Source data 3.** Confocal Z-stack images of dorsal trunk showing mCherry::Pio expression in Np mutant early stage 17 embryo.

**Source data 4.** Confocal Z-stack images of dorsal trunk showing mCherry::Pio expression in Np mutant mid stage 17 embryo.

**Source data 5.** Confocal Z-stack images of dorsal trunk showing mCherry::Pio expression in *flz* mutant embryos during stages 16 and 17.

**Source data 6.** Confocal Z-stack images of dorsal trunk showing mCherry::Pio expression in flz mutant stage 17 embryo.

**Source data 7.** Uncropped western blots (*Figure 6A*).

**Source data 8.** Uncropped western blots (*Figure 6B*).

**Figure supplement 1.** The tracheal expression and localization of the serine protease Filzig (Flz).

**Figure 6—video 1.** In *Np* mutant embryo, mCherry::Pio foci are visible intracellularly as in *wt*, but almost no mCherry::Pio is detectable in the tracheal lumen.

https://elifesciences.org/articles/91079/figures#fig6video1

**Figure 6—video 2.** Confocal time-lapse movie of endogenously expressed Dpy::eYFP (magenta) and mCherry::Pio (green) in *Np* mutant embryo.

https://elifesciences.org/articles/91079/figures#fig6video2

---

was not apparent in stage 17 embryos (*Figure 6A*). These findings support our recent study suggesting that Np cleaves the Pio ZP domain in vitro and in vivo (*Drees et al., 2019*). Co-expression of the catalytically inactive Np^S990A with mCherry::Pio in *Drosophila* Schneider cells showed as a prominent signal the 90 kDa mCherry::Pio variant in the cell lysate (*Figure 6B*), and live imaging revealed mCherry::Pio localization at the cell surface (*Figure 5—figure supplement 2B*). This was comparable with control cells that expressed mCherry::Pio alone (*Figure 6B*, *Figure 5—figure supplement 2A*). In contrast, cells that co-expressed either the functional Np or its homolog, the human Matriptase, revealed as a predominant signal the 60 kDa mCherry::Pio variant in the supernatant fraction (*Figure 6B*) and no mCherry::Pio localization at the cell surface (*Figure 5—figure supplement 2B*). These results demonstrate that the enzymatic activity of the serine protease Np is sufficient for ZP domain cleavage, which results in the ectodomain shedding of Pio in Schneider cells.

Next, we investigated the consequences of NP-mediated ZP cleavage. FRAP experiments showed only minor intracellular recovery of mCherry::Pio in *Np* null mutant embryos (*Figure 6—video 1*). In contrast to the control, extracellular mCherry::Pio is not released into the tube lumen within 56 min after bleaching in *Np* mutant embryos (*Figure 6C*, *Figure 6—video 1*). The luminal mCherry::Pio immobility was comparable to the immobile Dpy::eYFP fraction in control and *Np* mutant embryos,

suggesting that Pio is hampered to diffuse in *Np* mutant embryos. Second, mCherry::Pio was not released into the tracheal lumen in late-stage 16 Np mutant embryo stainings (*Figure 6D*, *Figure 6—video 2*). Third, the tracheal trypsin-like S1A Serine transmembrane protease Filzig had no influence on the mCherry::Pio pattern (*Figure 6E*, *Figure 6—figure supplement 1*). The *flz* mutant (*Figure 6—figure supplement 1*) and *Np* mutants stage 17 embryos showed no Dpy clearance from the tracheal lumen (*Drees et al., 2019*). However, the *flz* mutation did not affect the release of Pio at the apical cell surface (*Figure 6E*). Further, the luminal aECM cable condensed during early stage 17 in *flz* mutant embryos as observed in wt (compare *Figure 5A* with *Figure 6E*), indicating that these processes require Np. In summary, our findings prove the inhibition of luminal Pio release in *Np* mutant embryos. Furthermore, the Np-mediated proteolysis of the Pio ZP domain is specific and plays a central role in Pio dynamics at and near the apical cell membrane during tube expansion.

To further analyze the Np function, we used Uif to examine the tracheal apical cell membrane structure. This revealed unusual bulge-like membrane deformations in late-stage 16 and 17 Np mutant embryos, while control embryos did not show such bulges (*Figure 7A and B*). Confocal time-lapse series revealed that bulge-like deformations emerged during early stage 16 embryos and grew in size (*Figure 7C*). Furthermore, the increasing size of the membrane bulges led to the detachment of α-tubulin::GFP marked cells from the aECM at the cell surface (*Figure 7C*). The control embryos did not show such a separation between cells and adjacent aECM (*Figure 1B*). However, confocal images of Uif and chitin show residual chitin at the Uif marked membrane and chitin at the detached aECM (*Figure 7A*). These results indicate the tearing of the tracheal aECM at the apical cell surface in *Np* mutant embryos.

Finally, our time-lapse series and the analysis of stage 17 embryos proved that the bulge-like cell membrane structures remained in *Np* mutant embryos. Moreover, these cell membrane bulges destabilize the normal epithelial barrier. Upon tracheal expression of myr-RFP in Np mutants, we detected accumulation of RFP signal first at the membrane bulge and subsequently in the tube lumen (*Figure 7—figure supplement 1*). The *Np* and *pio* mutant embryos show apical membrane deformations (compare *Figures 1A and 7A*). Np is located at the membrane (*Drees et al., 2019*), where it controls Pio release into the tracheal lumen. This suggests that membrane deformations in *Np* mutant embryos are caused by impaired function of the Pio-mediated ZP matrix due to a lack of Pio shedding.

Any imbalance between membrane and matrix during tube expansion causes tube deformations. Chitin staining was reduced and revealed sinusoidal over-elongated tubes in stage 16 *pio* mutant embryos (*Figure 8A and B*; *Figure 1—figure supplement 1B and C*), proving that *pio* function prevents tracheal tubes from over-elongation. In *Np* mutant stage 16 embryos, Pio was not shed into the lumen but remained at the cell, while Dumpy showed normal and immobile localization within the tube lumen (*Figure 6C and D*; *Figure 6—videos 1; 2*). This indicates that the Pio-mediated ZP matrix can still restrict tube expansion of the membrane in Np mutants independently from membrane deformations. The *Np* mutant stage 16 embryos did not show tube overexpansion (*Figure 8A and B*). Further, *Np,pio* double mutants did not exacerbate the *pio* mutant tube length defects suggesting that both act in the same genetic pathway (*Figure 8A and B*). Our data assumes that Np overexpression may enhance Pio shedding in stage 16 embryos, affecting the Pio-mediated ZP matrix function. Upon *breathless (btl)*-Gal4-mediated expression of UAS-Np in tracheal cells, we observed a high amount of Pio puncta across the entire tracheal tube lumen, specifically in stage 16 embryos but not in earlier stages (*Figure 8—figure supplement 1*). Consistently tracheal Np overexpression led to tube overexpansion in stage 16 embryos resembling the *pio* mutant phenotype (*Figure 8A and B*). Thus, Np-mediated Pio shedding controls Pio function.

Since our findings show that Np controls tracheal Pio function by ZP domain cleavage, we addressed whether this is also a putative mechanism of human ZP domain proteins. The type III transforming growth factor-β receptor (TβRIII) acts as a signaling modifier and co-receptor of TGFβ and contains a ZP domain (*López-Casillas et al., 1991*; *Moustakas et al., 1993*). TβRIII ectodomain shedding in lung cancer cell models induces epithelial-to-mesenchymal transition and promotes the growth of tumors (*Huang et al., 2019*). For an in vitro cleavage assay, we co-expressed GFP- and Strep-tagged combined variants of TβRIII together with either human Matriptase or Np in *Drosophila* cells. The C-terminal GFP tag detects the intracellular part of TβRIII. The N-terminal Strep-tag follows the extracellular ZP domain and recognizes the ectodomain. Cells expressing TGFβRIII without Matriptase or Np revealed strong GFP and Strep co-localization signals. Co-expression of either human Matriptase

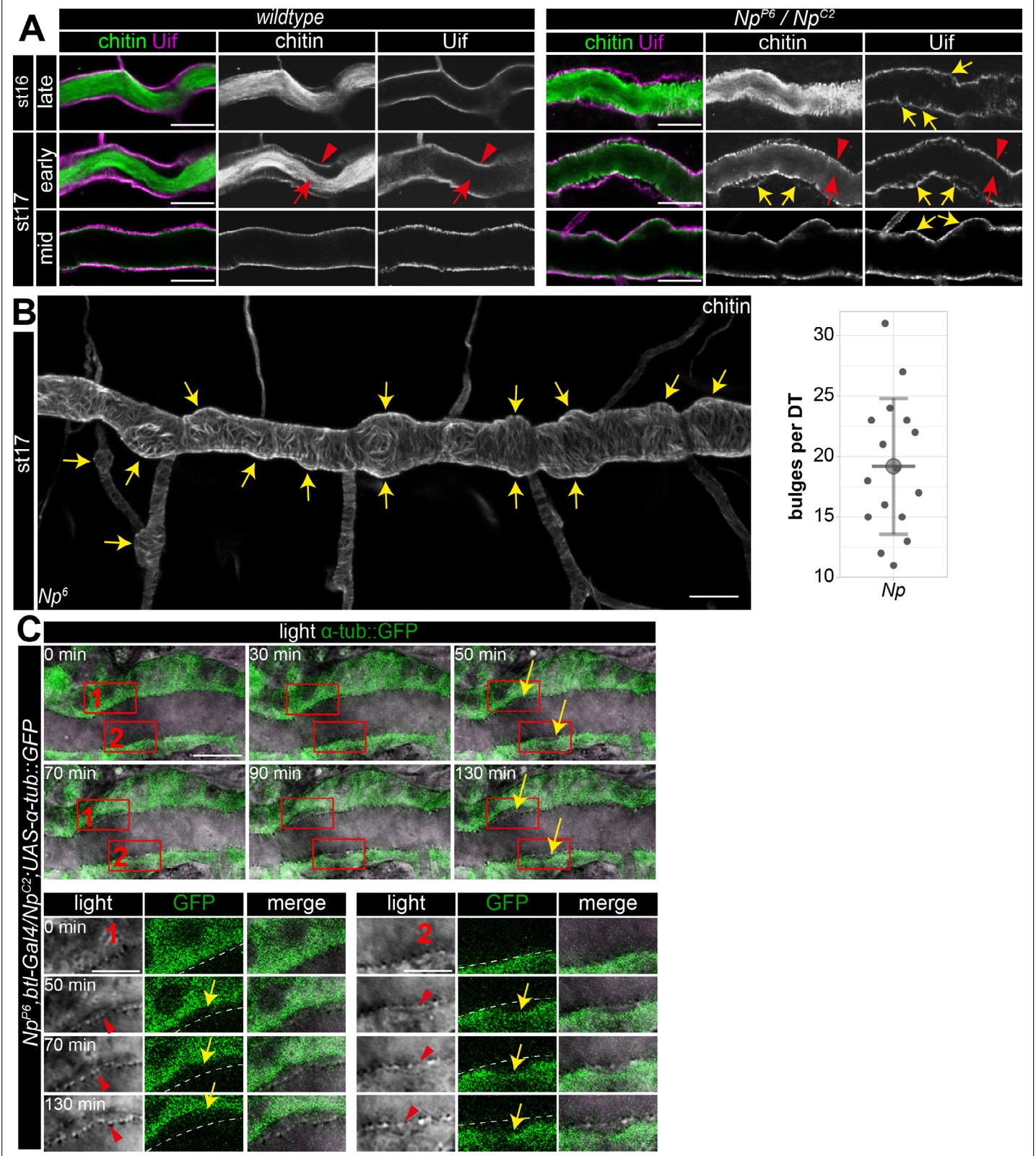

**Figure 7.** Np supports structural cell membrane integrity. (**A**) Bulge-like tracheal apical membrane deformations appeared in *Np* mutant embryos as stable structures that grow during late stage 16. Confocal images of dorsal trunks of *wt* embryos and *Np^P6^/Np^C2^* embryos stained with cbp (chitin; green) and anti-Uif antibody (magenta) as a marker for apical tracheal membranes. (**B**) Confocal Z-stack projections of cbp staining of stage 17 *Np* mutant shows several bulges (arrows) at the dorsal trunk. Quantification of bulges per dorsal trunk (n=16) in *Np* mutants is shown right. (**C**) The in

*Figure 7 continued on next page*

*Figure 7 continued*

vivo time-lapse series of 130 min show bulges arising at Np mutant embryos' dorsal trunk cell membranes. Tracheal cells express Tubulin::GFP in transheterozygous $Np^{P6};Np^{C2}$ mutant embryos. Frames 1 and 2 are shown as close-ups (below) of forming bulges. The cell membrane of GFP-expressing cells and parts of the misorganized tracheal cuticle apical extracellular matrix (aECM) (dashed line) separate, as shown in the time-lapse. Yellow arrows point to membrane deformations; red arrowheads point to the tracheal cuticle at the apical cell surface, and red arrows to the luminal aECM cable. Note that chitin is detectable in two layers at the sites of bulges, while Uif is detectable only at the bulges, indicating a disintegration of the tracheal chitinous aECM (**A**). Scale bars are 10 µm (overviews) and 3.5 µm (details). Note that bulges grew as time progressed (up to 130 min).

The online version of this article includes the following source data and figure supplement(s) for figure 7:

**Source data 1.** Quantification and confocal Z-stack images of bulges at dorsal trunks of Np mutant embryos.

**Source data 2.** Quantification of bulges at dorsal trunks of Np mutant embryos.

**Figure supplement 1.** Tracheal expression of myr-RFP in *Np* mutant embryos.

or Np caused a substantial reduction of the extracellular Strep signal. In contrast, the intracellular GFP signal remained (*Figure 8C*). This shows that matriptases catalyze the extracellular proteolysis and the extracellular localization of the human TβRIII ZP domain protein. This indicates a new and unexpected conserved mechanism capable of controlling TβRIII function.

## Discussion

Tracheal tube lumen expansion requires mechanical stress regulation at apical cell membranes and attached aECM. This involves the proteolytic processing of proteins that set local membrane-matrix linkages. Thus, the membrane microenvironment exhibits critical roles in regulating tube and network functionality.

ZP domain proteins organize protective aECM in the kidney, tectorial inner ear, and ZP (*Jovine et al., 2005*; *Litscher and Wassarman, 2020*), as well as in *Drosophila* epidermis, tendon cells, and appendages (*Bökel et al., 2005*; *Plaza et al., 2010*; *Ray et al., 2015*). *Drosophila* ZP domain proteins link the aECM to actin and polarity complexes in epithelial cells (*Fernandes et al., 2010*). Dumpy establishes force-resistant filaments for anchoring tendon cells to the pupal cuticle (*Chu and Hayashi, 2021*).

We identify that ZP protein-mediated microenvironmental changes increase the flexibility of membrane-matrix association, resulting from the activity of ZP domain proteins (*Figures 1 and 7*). Shear stress stimulates the activity of membrane-anchored proteases (*Kang et al., 2015*) and potentially also Np since we did not observe the misdistribution of the tracheal cytoskeleton when blisters arise. The dynamic membrane-matrix association control is based on our findings that loss of Np prevents Pio ectodomain shedding at the apical cell membrane resulting in immobile localization of Pio at the membrane and Dpy localization within the matrix (*Figure 6*). Direct interaction and overlapping subcellular localization at the cell surface showed that both proteins form a ZP matrix that potentially attaches membrane and ECM (*Figure 5*; *Figure 5—figure supplement 1*). Deregulation of Pio shedding blocks ZP matrix rearrangement and release of membrane-matrix linkages under tube expansion and subsequent shear stress. This destabilizes the microenvironment of membranes, causing blister formation at the membrane due to ongoing membrane expansion (*Figure 7*). Additionally, Pio could be part of a force-sensing signal transduction system destabilizing the membrane and matrix. Our observation that the membrane deformations are maintained in *Np* mutant embryos supports our postulated Np function to redistribute and deregulate membrane-matrix associations in stage 16 embryos when tracheal tube length expands. In contrast, Np overexpression potentially uncouples the Pio-Dpy ZP matrix membrane linkages resulting very likely in unbalanced forces causing sinusoidal tubes (*Figure 8*).

The membrane defects observed in both Pio and Np mutants indicate errors in the coupling of the membrane matrix due to the involvement of Pio (*Figures 1 and 7*). In *pio* mutants, gaps appear between the deformed membrane and the apical matrix (*Figure 1B–D*). These changes in apical cell membrane shape are consistent with increased cell and tube elongation in *pio* mutant embryos because the matrix is uncoupled from the membrane in such mutants (see model *Figure 9*). In contrast to *pio* mutants, the large membrane bulges in *Np* mutants affect the membrane and the apical matrix (*Figure 7*). Since apical Pio is not cleaved in *Np* mutants (*Figure 6D*), the matrix is not uncoupled from the membrane as in *pio* mutant embryos but is likely more intensely coupled, which leads to tearing

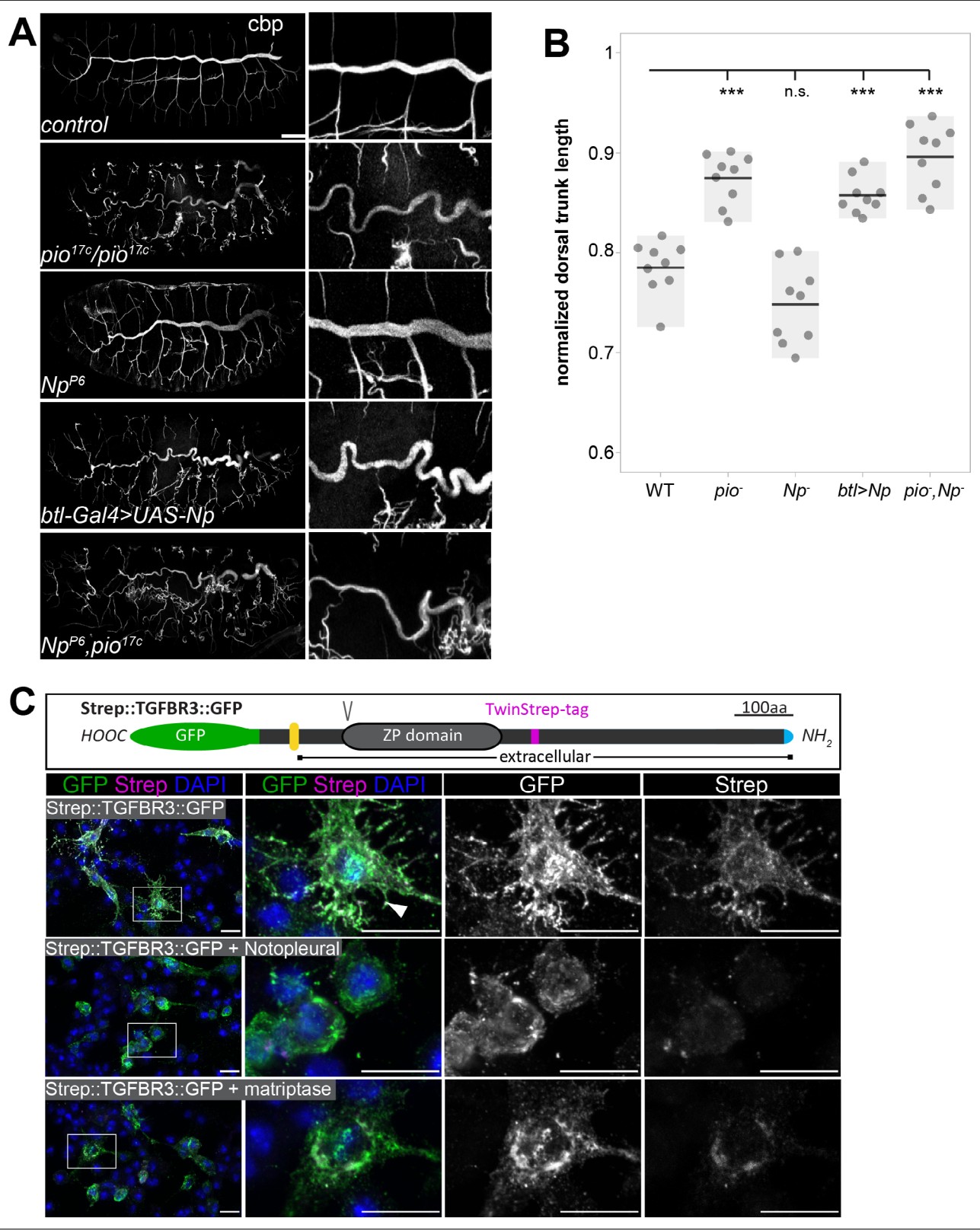

**Figure 8.** Pio and Np control tube size and their regulatory mechanisms of ZP domain shedding is conserved. (**A**) Confocal Z-stack projections of whole-mount stage 16 embryos stained with cbp (chitin) focusing on the tracheal system and close-ups at the right. In contrast to straight branches of control embryos, *pio¹⁷ᶜ* null mutant embryos revealed curly elongated tubes indicating excess tube expansion. Note dorsal branch disruption known from a hypomorphic *pio* point mutation (*Jaźwińska et al., 2003*). *Np* mutant embryos show straight *wt*-like tubes. Embryos that overexpress

*Figure 8 continued on next page*

*Figure 8 continued*

*Np* in the tracheal system (*btl-Gal4>UAS*-Np) show curly elongated tubes and dorsal branch disruption, phenocopying the *pio* mutant phenotype. *Np;pio* double mutant embryos do not exacerbate *pio* mutant tube size defects and show a similar phenotype as *pio* mutant embryos, respectively. (**B**) Quantification of normalized dorsal trunk length from 9 stage 16 embryos of each genotype. Anterior-posterior dorsal trunk length was divided by the anterior-posterior length of the embryo. Normalized dorsal trunk length in *pio* mutant embryos, *btl-Gal4>UAS*-Np embryos and *Np,pio* double mutant embryos is significantly increased when compared with *wt* (p=0.00013, p=0.00007, p=0.00019). Notably, the Np mutant dorsal trunk is relatively straight, while control embryos show slightly convoluted tubes. Also, statistical analysis reveals the tendency of slightly shortened dorsal trunk length in Np mutant. Individual points represent the respective embryos. (**C**) Human TGFβ type III receptor (TGFβRIII) is a widely expressed ZP domain containing protein. Human TGFβRIII with a cytoplasmic GFP tag and an extracellular Strep tag was expressed in *Drosophila* S2R+ cells either alone or together with *Drosophila* Np or human Matriptase. A schema of the tagged TGFβRIII is shown (top). The ZP and transmembrane (yellow) domains, the N-terminal Strep tag (magenta), C-terminal GFP (green) Furin protein cleavage site, (**V**) and the signal peptide (blue) are indicated. Images display maximum intensity projections of confocal Z-stacks. Shown are S2R+ cells that expressed the Strep::TGFβRIII::GFP construct alone or together with Np or human Matriptase stained with DAPI (blue) anti-GFP (green) and anti-Strep (magenta) antibodies. Single channel panels are indicated in gray. Control cells contain co-localizing GFP and Strep signals. The co-expression of Np or Matriptase reveals strong GFP but faint Strep signals due to extracellular cleavage and shedding of the TGFβRIII ectodomains. Framed boxes in overview images display details in panels on the right side. Scale bars indicate 10 µm.

The online version of this article includes the following source data and figure supplement(s) for figure 8:

**Source data 1.** Quantification of dorsal trunks lengths.

**Figure supplement 1.** Np controls Pio shedding into the tube lumen during tube expansion.

---

of the matrix axially along the membrane bulges (*Figures 7 and 9*; *Figure 7—figure supplement 1*), when the tube expands in length. If apical Pio detachment reduces coupling between the matrix and apical membrane, then it is likely that *Np* mutant embryos may exhibit a reduced tube length phenotype. In Np mutant embryos, average tracheal dorsal trunk length tends to be reduced compared to *wt* embryos (*Figure 8B*), suggesting that Pio shedding is critical in controlling tracheal tube lumen length.

The *btl*-Gal4-driven Np expression mimics the endogenous Np from stage 11 onward in all tracheal cells throughout embryogenesis (*Drees et al., 2019*), suggesting that Np is not expressed at a wrong time point. However, the ratio between Np and Pio is essential. We assume that tracheal Np overexpression increases Pio shedding in stage 16 embryos (*Figure 8—figure supplement 1*), resulting in a *pio* loss-of-function-like phenotype. Thus, the tube length overexpansion upon Np overexpression indicates that Pio cleavage is required for tube length control.

Is Pio ectodomain shedding in response to tension? We did not measure tension directly. However, the developmental profile of mechanical tension during tracheal tube length elongation in stage 16 embryos (*Dong et al., 2014*) is consistent with the profile of Pio shedding. Np cleaves apical Pio during stage 16 when tube length expands (*Figures 5 and 6*). In contrast, Pio shedding decreases sharply at early stage 17 when tube elongation is completed (*Figures 5A and 6A*). Our model, therefore, predicts that loss of Pio or increased Pio secretion at stage 16 may reduce the coupling of the membrane matrix so that increased tracheal tube elongation is maintained until the end of stage 16, which is found in *pio* mutants and upon Np overexpression (*Figure 8A and B*). Unknown proteases may likely be involved in Pio processing since cleaved mCherry::Pio is also detectable in inactive NpS990A cells. Previously we identified a mutation at the Pio ZP domain (R196A) resistant to NP cleavage in cell culture experiments (*Drees et al., 2019*). Establishing a corresponding mutant fly line would be essential in determining whether the observed phenotype resembles the phenotype of the *Np* mutant embryos. In addition, unknown mechanisms, such as distinct membrane connections during development and emerging links to the developing cuticle, may also influence tension at the apical membrane during tube length control.

Indeed, the anti-Pio antibody, which detects all different Pio variants, showed a punctuate Pio pattern overlapping with the apical cell membrane markers Crb and Uif at the dorsal trunk cells of stage 16 embryos (*Figure 2*; *Figure 2—figure supplement 1*, *Figure 2—figure supplement 2*). Additionally, Pio antibody also revealed early tracheal expression from embryonic stage 11 onward, and due to Pio function in narrow dorsal and ventral branches, strong luminal Pio antibody staining is detectable from early stage 14 until stage 17, when airway protein clearance removes luminal contents. In the *pio⁵ᵐ* and *pio¹⁷ᶜ* mutants, Pio stainings were strongly reduced although some puncta were still detectable in the trachea (*Figure 1—figure supplement 1G and H*). Similarly, Pio antibody

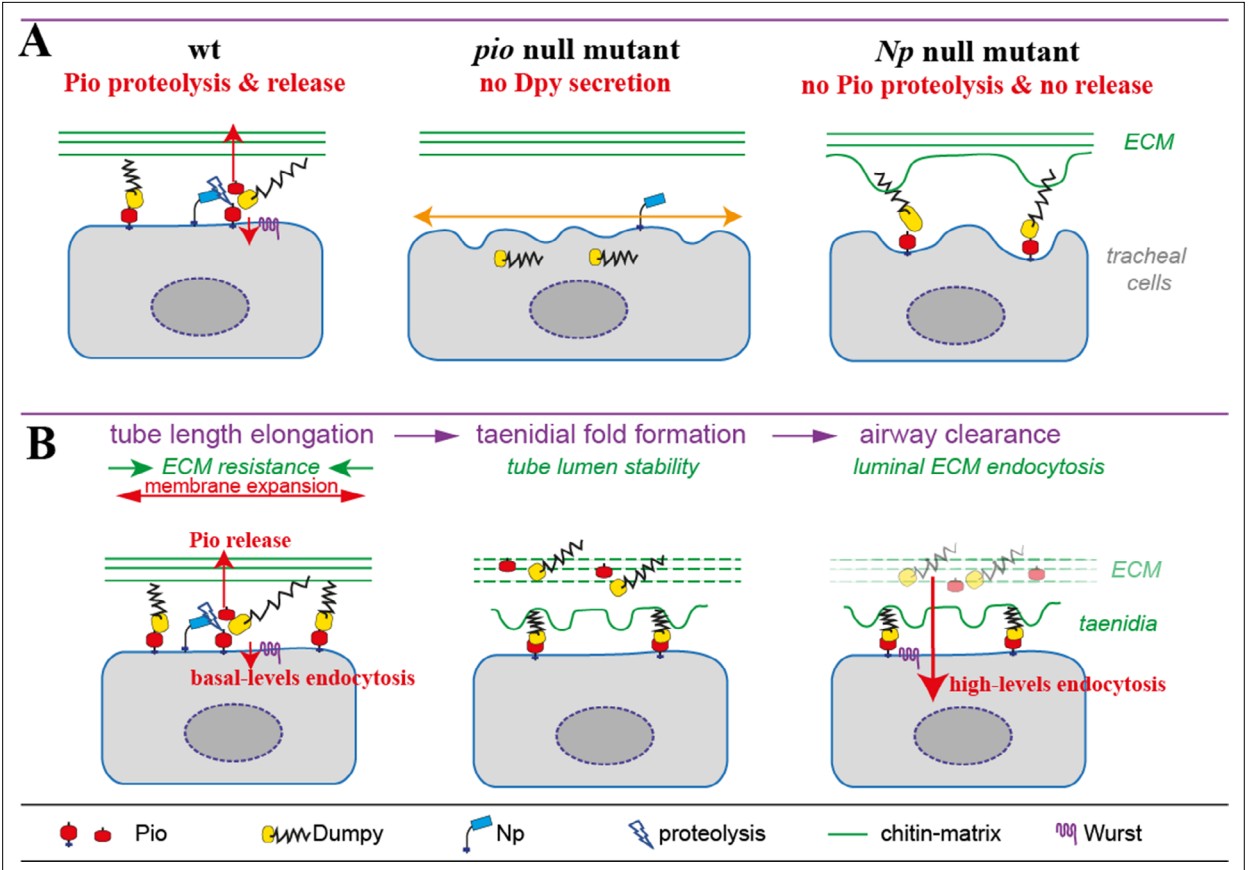

**Figure 9.** Model of apical Pio and Dpy matrix at the apical cell surface and Pio proteolysis and release. (**A**) The simplified model shows a Pio-mediated Dpy secretion, the apical ZP domain protein matrix formation, and subsequent Pio ectodomain shedding by Np at wt tracheal cells. In *pio* null mutant embryos, Dpy is not secreted. The loss of the apical ZP proteins led to membrane matrix disconnection. This results in unstable membrane structures with numerous unusual gaps between membrane and matrix, excess apical cell shape expansion (indicated by the orange double arrow) in the axial direction, and dorsal trunk overexpansion. In contrast, apical ZP domain protein matrix forms in *Np* mutant embryos, but apical Pio ectodomain shedding is prevented. This led to membrane bulging and rupture (indicated by green arrows) of the apical ECM at bulges and slightly shortened dorsal trunks. (**B**) Simplified illustration of late embryonic tracheal development. The ZP proteins stabilize the apical membrane-matrix during tube expansion, subsequent taenidial fold formation, and airway clearance. Np cleaves Pio ZP domain at the membrane during tube expansion and sheds luminal Dpy during airway protein clearance. Wurst enables endocytosis at the apical membrane. This altogether enables precise tube length expansion (stage 16), lumen stability and airway gas-filling (stage 17). Basal (stage 16) and high-level (stage 17) endocytosis at the apical cell membrane have been described recently by *Tsarouhas et al., 2019*.

staining is intracellular in the trachea of stage 11 *pio$^{2R-16}$* point mutation embryos (*Jaźwińska et al., 2003*). Interestingly, also *dpy* mutants showed strongly reduced and intracellular Pio antibody staining (*Figure 5—figure supplement 3E*).

We generated mCherry::Pio as a tool for in vivo Pio expression and localization pattern analysis during tube lumen length expansion. The mCherry::Pio resembled the Pio antibody expression pattern from early tracheal development onward. However, luminal mCherry::Pio enrichment occurs specifically during stage 16, when tubes expand. The stage 16 embryos showed mCherry::Pio puncta accumulating apically in dorsal trunk cells. Moreover, mCherry::Pio puncta partially overlapped with Dpy::YFP and chitin at the taenidial folds, forming at apical cell membranes. Supported by several observations, such as antibody staining, video monitoring, FRAP experiments, and western blot studies (*Figure 5*; *Figure 8—figure supplement 1*; *Figure 5—videos 1 and 2*; *Figure 6—videos 1; 2*), these findings indicate that Pio may play a significant role at the apical cell membrane and matrix in dorsal trunk cells of stage 16 embryos.

Furthermore, we show that Np mediates Pio ZP domain cleavage for luminal release of the short Pio variant during ongoing tube length expansion. The luminal cleaved mCherry::Pio is enriched at the end of stage 16 and finally internalized by the subsequent airway clearance process during stage

17 after tube length expansion (*Figures 5 and 6*). Such rapid luminal Pio internalization is consistent with a sharp pulse of endocytosis rapidly internalizing the luminal contents during stage 17 (*Tsarouhas et al., 2007*). Wurst is required to mediate the internalization of proteins in the airways (*Behr et al., 2007*; *Stümpges and Behr, 2011*). In consistence, during stage 17, luminal Pio antibody staining fades in control embryos but not in Wurst deficient embryos.

Nevertheless, Pio and its endocytosis depend on its interaction with the chitin matrix and the Np-mediated cleavage. In stage 16 *wurst* and *mega* mutant embryos, we detect Pio antibody staining at the chitin cable, suggesting that Pio is cleaved and released into the dorsal trunk tube lumen. Also, the Cht2 overexpression did not prevent the luminal release of Pio. However, reduced *wurst*, mega function, and Cht2 overexpression caused an enrichment of punctuate Pio staining at the apical cell membrane and matrix (*Figures 1 and 2*). Although the three proteins are involved in different subcellular requirements, they all contribute to the determination of tube size by affecting either the apical cell membrane or the formation of a well-structured apical extracellular chitin matrix, indicating that changes at the apical cell membrane and matrix in stage 16 embryos affect the Pio pattern at the membrane. It also shows that local Pio linkages at the cell membrane and matrix are still cleaved by the Np function for luminal Pio release, which explains why those mutant embryos do not show *pio* mutant-like membrane deformations and Np-mutant-like bulges. This is in line with our observations that tracheal Pio overexpression cannot cause tube size defects as the Np function is sufficient to organize local Pio linkages at the membrane and matrix. Therefore, it is unlikely that tracheal tube length defects in *wurst* and *mega* mutants as well as in Cht2 misexpression embryos are caused by the apical Pio density enrichment.

Nevertheless, oversized tube length due to the misregulation of the apical cell membrane and adjacent chitin matrix may cause changes to local Pio set linkages and the need for Np-mediated cleavage. Strikingly, we observe a lack of Pio release in *Np* mutants. This shows that Pio density at the membrane versus lumen depends predominantly on Np function. The molecular mechanisms that coordinate the Np-mediated Pio cleavage are unknown and will be necessary for understanding how tubes resist forces that impact cell membranes and matrices. On the other hand, Pio is required for the extracellular secretion of its interaction partner Dpy (*Figure 5—figure supplement 2*). At the same time, Dpy is needed for Pio localization at the cell membrane and its distribution into the tube lumen (*Figure 5—figure supplement 3*). Consistently, in vivo, mCherry::Pio, and Dpy::eYFP localization patterns overlap at the apical cell surface and within the tube lumen (*Figure 5—figure supplement 1*). These observations support our model that Pio and Dpy interact at the cell surface where Np mediates Pio cleavage to support luminal Pio release by the large and stretchable matrix protein Dpy (*Figure 9*).

Taenidial organization prevents the collapse of the tracheal tube. Therefore, cortical (apical) actin organizes into parallel-running bundles that proceed to the onset of cuticle secretion and correspond precisely to the cuticle's taenidial folds (*Matusek et al., 2006*; *Öztürk-Çolak et al., 2016*). Mutant larvae of the F-actin nucleator formin DAAM show mosaic taenidial fold patterns, indicating a failure of alignment with each other and along the tracheal tubes (*Matusek et al., 2006*). In contrast, *pio* mutant dorsal tracheal trunks contained increased ring spacing (*Figure 3A*). Fusion cells are narrow doughnut-shaped cells where actin accumulates into a spotted pattern. Formins, such as diaphanous, are essential in organizing the actin cytoskeleton. However, we do not observe dorsal trunk tube fusion defects as found in the presence of the activated diaphanous.

On the other hand, ectopic expression of DAAM in fusion cells induces changes in apical actin organization but does not cause any phenotypic effects (*Matusek et al., 2006*). DAAM is associated with the tyrosine kinase Src42A (*Nelson et al., 2012*), which orients membrane growth in the axial tube dimension (*Förster and Luschnig, 2012*). The Src42 overexpression elongates tracheal tubes due to flattened axially elongated dorsal trunk cells and AJ remodeling. Although flattened cells and tube overexpansion are similar in *pio* mutant embryos, we did not observe a mislocalization of AJ components, as found upon constitutive Src42 activation (*Förster and Luschnig, 2012*). Instead, we detected an unusual stretched appearance of AJs at the fusion cells of *pio* mutant dorsal trunks (*Figure 4B and C*), which to our knowledge, has not been observed before and may play a role in regulating axial taenidial fold spacing and tube elongation.

Self-organizing physical principles govern the regular spacing pattern of the tracheal taenidial folds (*Hannezo et al., 2015*). The actomyosin cortex and increased actin activity before and turnover at

stage 16 drive the regular pattern formation. However, the cell cortex and actomyosin are in frictional contact with a rigid apical ECM. The Src42A mutant embryos contain shortened tube length but increased taenidial fold period pattern due to decreased friction. In contrast, the chitinase synthase mutant *kkv[1]* has tube dilation defects and no regular but an aberrant pearling pattern caused by zero fiction (*Hannezo et al., 2015*).

In contrast, *pio* mutant embryos do not contain tube dilation defects or shortened tubes but increased tube length (*Figures 1 and 8*; *Figure 1—figure supplement 1*). Furthermore, our cbp and antibody stainings reveal the presence of a luminal chitin cable and a solid aECM structure in *pio* mutant stage 16 embryos (*Figure 8*; *Figure 1—figure supplement 1*; *Figure 3—figure supplement 2*). In addition, apical actin enrichment in tracheal cells of *pio* mutant embryos appeared *wt*-like. Nonetheless, *pio* mutant embryos show an increased taenidial fold period compared with wt, indicating a decreased friction. Thus, we propose that the lack of Pio reduces friction. Reasons might be subtle defects of actomyosin constriction or chitin matrix, which we have not detected in the *pio* mutant tracheal cells. Further reasons for lower friction might also be the loss of Pio set local linkages between apical cortex and aECM in stage 16 embryos, which are modified by Np, as proposed in our model (*Figure 9*).

Heterozygous and homozygous *pio* mutant embryos generally do not show tubal collapse. However, the loss of Pio and accompanying lack of Dpy secretion in stage 17 *pio* mutant embryos led to the loss of a Pio/Dpy matrix, impacting the late embryonic maturation and differentiation of a normal chitin matrix at the apical cell surface. TEM images reveal reduced dense chitin matrix material at taenidial folds and misarranged taenidial fold pattern (*Figure 1*; *Figure 2—figure supplement 1*), suggesting impaired taenidial function prevents tube lumen from collapsing after tube protein clearance. *Wurst* knockdown and mutant embryos do not show general tube collapse, but luminal chitin fiber organization is disturbed in stage 17 embryos (*Behr et al., 2007*). Therefore, transheterozygous *wurst;pio* mutant embryos may combine both defects and suffer from maturation deficits of the chitin/ZP matrix at the apical cell surface and within the tube lumen, which finally causes a high number of embryos with incomplete gas-filling due to tube collapse. These maturation deficits are even more dramatic in the *wurst;pio* double mutants, which show no gas-filling.

Our studies on human Matriptase provide evidence for a mechanistic conservation of ZP domain protein as a substrate for ectodomain shedding (*Figure 8*). The upregulation of Matriptase activity and increased TGFβ receptor density affect human and mouse model idiopathic pulmonary fibrosis cells on pulmonary fibrogenesis (*Bardou et al., 2016*; *Naik et al., 2012*). Furthermore, the human Matriptase induces the release of proinflammatory cytokines in endothelial cells, which contribute to atherosclerosis and probably also to abdominal aortic aneurysms (*Seitz et al., 2007*). The membrane bulges arising in our *Drosophila* model during tracheal tube elongation upon Np loss of function showed analogy to the appearance of artery aneurysms. Bulges with varying phenotypic expression in different organs can lead to aortic rupture due to fragile artery walls or degeneration of layers in responses to stimuli, such as shear stresses (*Kubo et al., 2015*). Indeed, aneurysms development is forced by alterations in the ECM (*Yoon et al., 1999*) and are characterized by extensive ECM fragmentation caused by shedding of membrane-bound proteins (*Antalis et al., 2016*; *Quintana and Taylor, 2019*; *Yoon et al., 1999*; *Zhong and Khalil, 2019*).

We identified a dynamic control of matrix proteolysis, very likely enabling fast and site-specific uncoupling of membrane-matrix linkages when tubes expand. Such a scenario has not yet been studied in angiogenesis. It may represent a new starting point for genetic studies to decipher the putative roles of ZP domain proteins and Matriptase in clinically relevant syndromes, including the formation of aneurysms caused by membrane deformation and defects in size determination of airways and vessels.

## Materials and methods

**Key resources table**

| Reagent type (species) or resource | Designation | Source or reference | Identifiers | Additional information |
|---|---|---|---|---|
| Antibody | Mouse, anti-Crb, monoclonal | DSHB | Cq4 | 1:10 |

*Continued on next page*

*Continued*

| Reagent type (species) or resource | Designation | Source or reference | Identifiers | Additional information |
|---|---|---|---|---|
| Antibody | Mouse anti-Flag, monoclonal | Merck | F3165 | 1:500 |
| Antibody | Mouse anti-Gasp/Obst-C, monoclonal | DSHB | 2A12 | 1:5 |
| Antibody | Chicken anti-GFP, polyclonal | Abcam | ab13970 | 1:1000 |
| Antibody | Rabbit anti-GFP, polyclonal | Synaptic Systems132003 | 132003 | IF: 1:500 WB: 1:10,000 CIF: 1:1000 |
| Antibody | Rabbit anti-Knk, polyclonal | *Moussian et al., 2006* | Moussian | 1:50 |
| Antibody | Mouse anti-Mega, monoclonal | *Jaspers et al., 2012* | Schuh | 1:50 |
| Antibody | Rabbit anti-mCherry, polyclonal | Rockland | 600-401P16 | IF: 1:500 WB: 1:10,000 |
| Antibody | Rabbit anti-Obst-A, polyclonal | *Petkau et al., 2012* | Behr | 1:50 |
| Antibody | Rabbit anti-Pio, polyclonal | *Jaźwińska et al., 2003* | Affolter | 1:100 |
| Antibody | Rabbit anti-Serp, polyclonal | *Luschnig et al., 2006* | Luschnig | 1:50 |
| Antibody | Rabbit anti-Spalt, polyclonal | *Kühnlein et al., 1994* | Schuh | 1:25 |
| Antibody | Mouse anti-α-Spectrin, monoclonal | DSHB | 3A9 | 1:10 |
| Antibody | Anti-Strep-HRP, mouse monoclonal | IBA | 1-1509-001 | 1:10,000 |
| Antibody | Mouse anti-β-Tubulin, monoclonal | DSHB | E7 | 1:100 |
| Antibody | Guinea pig ani-Uif, polyclonal | *Zhang and Ward, 2009* | Ward | 1:100 |
| Antibody | Rabbit anti-Verm, polyclonal | *Luschnig et al., 2006* | Luschnig | 1:50 |
| Antibody | Donkey anti-goat Alexa488, polyclonal | Dianova | 705-545-003 | 1:400 |
| Antibody | Donkey anti-guinea pig Cy3, polyclonal | Dianova | 706-165-148 | 1:400 |
| Antibody | Donkey anti-mouse Alexa647, polyclonal | Dianova | 715-605-020 | 1:400 |
| Antibody | Donkey anti-mouse Alexa647, polyclonal | Dianova | 715-605-020 | 1:400 |
| Antibody | Donkey anti-rabbit Cy3, polyclonal | Dianova | 711-167-003 | 1:400 |
| Antibody | Donkey anti-rabbit-AlexaFluor488, polyclonal | Thermo Fisher Scientific | A-11034 | 1:500 |
| Antibody | Donkey anti-rabbit-AlexaFluor568, polyclonal | Thermo Fisher Scientific | A-21069 | 1:500 |
| Antibody | Goat anti-mouse-HRP, polyclonal | Thermo Fisher Scientific | G-21040 | WB: 1:10,000 |
| Antibody | Goat anti-rabbit-HRP, polyclonal | Thermo Fisher Scientific | Thermo Fisher ScientificG-21234 | 1:10,000 |
| Reagent | WGA, Alexa Flour 633 | Invitrogen | W21404 | 1:100 |
| Reagent | Cbp, Alexa488 | New England Biolabs | | 1:200 |

*Continued*

| Reagent type (species) or resource | Designation | Source or reference | Identifiers | Additional information |
|---|---|---|---|---|
| Reagent | Phalloidin- PromoFluor-488 | PromoKine, VWR | PROMOPK-PF488P-7 | 1:75 |
| Genetic reagent | btl-Gal4 | Bloomington Drosophila Stock Center (BDSC) | | |
| Genetic reagent | crb[2] (crb[11A22]) | BDSC | Stock ID 3448 | https://flybase.org/reports/FBal0001817.html |
| Genetic reagent | dpy[olvR]/SM5 | BDSC | Stock ID 280 | https://flybase.org/reports/FBal0002971#phenotypic_data_sub |
| Genetic reagent | Dumpy::eYFP [CPTI-001769] | *Lye et al., 2014* | Sanson | |
| Genetic reagent | mega[G0012]/FM7, act-GFP | *Behr et al., 2003* | Schuh; U Schäfer | |
| Genetic reagent | shrub[4]/Cyo | *Dong et al., 2014* | Hayashi | |
| Genetic reagent | w[1118] | BDSC | | https://flybase.org/reports/FBal0018186.html |
| Genetic reagent | w*; mCherry::pio/CyO, dfd-eYFP | This manuscript | Drees | Generation as described in the supplement, available from MB |
| Genetic reagent | w*; flz[C1] | This manuscript | Drees | Generation as described in the supplement, available from MB |
| Genetic reagent | w*; flz[C1], mCherry::pio/CyO, dfd-eYFP | This manuscript | Drees | Generation as described in the supplement, available from MB |
| Genetic reagent | w[1118];PBac{681  .P.FSVS-1} flz[CPTI001902] | Kyoto Stock Center | Stock ID 115246 | https://flybase.org/reports/FBti0143804 |
| Genetic reagent | w*; pio[5M]/CyO, dfd-eYFP | This manuscript | Drees | Generation as described in the supplement, available from MB |
| Genetic reagent | w*; pio[17C]/CyO, dfd-eYFP | This manuscript | Dress | Generation as described in the supplement, available from MB |
| Genetic reagent | w*; Np[P6]/CyO, dfd-eYFP | *Drees et al., 2019* | Drees | Available from MB |
| Genetic reagent | w*; Np[P6], P{Gal4-btl}/CyO, dfd-eYFP | *Drees et al., 2019* | Drees | Available from MB |
| Genetic reagent | w*; Np[C2], P{UAS-Np[S990A]}/CyO, dfd-eYFP | *Drees et al., 2019* | Drees | Available from MB |
| Genetic reagent | w*; Np[C2]/CyO, dfd-eYFP; P{UASp-GFPS65C-alphaTub84B}3/TM3, Sb[1] | *Drees et al., 2019* | Drees | Available from MB |
| Genetic reagent | w*; Np[P6], mCherry::pio/CyO, dfd-eYFP | *Drees et al., 2019* | Drees | Available from MB |
| Genetic reagent | w*; P{UAS-Np[S990A]}/P{UAS-Np[S990A]} | *Drees et al., 2019* | Drees | Available from MB |
| Genetic reagent | w*; P{UAS-Np}/P{UAS-Np} | *Drees et al., 2019* | Drees | Available from MB |
| Genetic reagent | wurst[162]/FM7-actin-GFP | *Behr et al., 2007* | Behr | Available from MB |
| Genetic reagent | UAS-Cht2 | *Tonning et al., 2005* | Uv | |
| Genetic reagent | w[1118]; P{w[+mC]=UAS-myr-mRFP}1 | BDSC | Stock ID 7118 | https://flybase.org/reports/FBst0007118.html |
| Genetic reagent | UAS-wurst-RNAi | *Stümpges and Behr, 2011* | VDRC | |
| Cell line (*D. melanogaster*) | S2R+ cells | *Drees et al., 2019* | DGRC | https://flybase.org/reports/FBtc0000150.html#:~:text = S2R%2B%20 is%20an%20isolate%20of,to %20the%20original%20S2%20line.&text = S2R%2B%20is%20an%20 isolate%20of%20S2 %20that%20has%20receptors%20for%20wg%20signalling. |
| Cell line (*D. melanogaster*) | Kc167 | *Drees et al., 2019* | DGRC | https://flybase.org/reports/FBtc0000001.html |
| Sequence-based reagent | pio-sgRNA-sense | Eurofins Genomic | | CTTCGATTGGGACACCGAGCCACT |
| Sequence-based reagent | pio-sgRNA-antisense | Eurofins Genomics | | AAACAGTGGCTCGGTGTCCCAATC |

*Continued*

| Reagent type (species) or resource | Designation | Source or reference | Identifiers | Additional information |
|---|---|---|---|---|
| Sequence-based reagent | flz-sgRNA-sense | Eurofins Genomics | | CTTCGTGGGTTACGCCGG CCTCAA |
| Sequence-based reagent | flz-sgRNA-antisense | Eurofins Genomics | | AAACTTGAGGCCGGCGTA ACCCAC |
| Sequence-based reagent | UAS-mCherry::pio-for | Eurofins Genomics | | GAATTCATGAAGACAGGCACTCGAATGGACGCTTTCCACA CGGCGCTGCACTTAATCACAATCGCAGCTCTGACGACG |
| Sequence-based reagent | UAS-mCherry::pio-rev | Eurofins Genomics | | CTCGAGGCCGCCTTTGTAAAGCTCATCC |
| Sequence-based reagent | Pio-5'-HA1-for | Eurofins Genomics | | ACTAGTCCGAATTCGCAGG TGATTATCGCCTCTCGGCC ATCAG |
| Sequence-based reagent | Pio-5'-HA1-rev | Eurofins Genomics | | AAGCTTCTTTAATTAAAGG GGAAATTTCG |
| Sequence-based reagent | Pio-5'-HA2-for | Eurofins Genomics | | ACTAGTGGCAAGCTTACTG GCGATGGATTAGGCC |
| Sequence-based reagent | Pio-5'-HA2-rev | Eurofins Genomics | | CACCTGCGATCTTAATCTT GCCAGCGTCTGTC |
| Sequence-based reagent | Pio-3'-HA-for | Eurofins Genomics | | TTAAGGAAGAGCACACAG TTGGGCGCTTTGTTAGTCG |
| Sequence-based reagent | Pio-3'-HA-rev | Eurofins Genomics | | CGGGGAAGAGCGACGAGA TTGCGCCGGAAAATAAG |
| Sequence-based reagent | UAS-pio-ORF-for | Eurofins Genomics | | CTCGAGCCAACGGCAATGAAAGATGCCC |
| Sequence-based reagent | UAS-pio-ORF-rev | Eurofins Genomics | | TCTAGATTAGCTGCTGTGCGAGAAG |
| Sequence-based reagent | Dpy-ZP-for | Eurofins Genomics | | GCTTTACAAAGGTTACACGGGTAATCCG |
| Sequence-based reagent | Dpy-CT-for | Eurofins Genomics | | GCTTTACAAAGGTGGAAATGCCAGGATTG |
| Sequence-based reagent | Dpy-CTZP-rev | Eurofins Genomics | | GTGGAGCCGGCCACCATTTATGGAGGTTTC |
| Sequence-based reagent | Dpy-ZP-for | Eurofins Genomics | | GGCCACCATTTATGGAGGTTTC |
| Sequence-based reagent | Dpy-ZP-rev | Eurofins Genomics | | GGTTCCTTCACAAAGATCCTTTAGGATATGTAATCCGGCG |
| Sequence-based reagent | Strep::TGFBR3::GFP1-for | Eurofins Genomics | | CTGAATAGGGAATTGGGAATTCATGACTTCCCATTATG |
| Sequence-based reagent | Strep::TGFBR3::GFP1-rev | Eurofins Genomics | | CACCGCTGCCACCTCCTGATCCGCCACCCTTTTCAAACTGC GGATGACTCCATGCACTTTGCACCTCTTCTGGCTCTC |
| Sequence-based reagent | Strep::TGFBR3::GFP2-for | Eurofins Genomics | | ATCAGGAGGTGGCAGCGGTGGAAGTGCATGGAGCCATCCCC AATTCGAGAAGGGGAGCGTGGATATTGCCCTG |
| Sequence-based reagent | Strep::TGFBR3::GFP2-rev | Eurofins Genomics | | TCACCATACCGCCGCTAGCGGCCGTGCTGCTGCTG |
| Plasmid | pJet1.2 | Thermo Fisher Scientific | | |
| Plasmid | pUAST | GAL4/UAS-mediated overexpression; *Brand and Perrimon, 1993* | | |
| Plasmid | pBFv-U6.2 | Expression of single sgRNA; *Kondo and Ueda, 2013* | | |
| Plasmid | pBFv-U6.2B | Expression of two sgRNAs; *Kondo and Ueda, 2013* | | |

*Continued on next page*

*Continued*

| Reagent type (species) or resource | Designation | Source or reference | Identifiers | Additional information |
|---|---|---|---|---|
| Plasmid | pHD-ScarlessDsRed | Scarless genome editing via HDR | DGRC | |
| Plasmid | actin5C-Gal4 | Expression of Gal4 in cultured cells; *Usui et al., 1999* | | |
| Software, algorithm | Clustal omega algorithm | https://www.ebi.ac.uk/Tools/msa/clustalo/ | | |
| Software, algorithm | DNASTAR software suite | Lasergene Software | Lasergene Software | |
| Software, algorithm | Flybase | https://www.flybase.org | https://www.flybase.org | BLASTP algorithm |
| Software, algorithm | Huygens professional | SVI | 20.10 | |
| Software, algorithm | Illustrator | Adobe | CS6 | https://www.adobe.com |
| Software, algorithm | Imaris 9.7.2 | Oxford Instruments | Oxford Instruments | https://imaris.oxinst.com/ |
| Software, algorithm | NetOGlyc | DTU Health Tech | | https://services.healthtech.dtu.dk/ |
| Software, algorithm | Office 365 (Word, Excel) | Microsoft | Microsoft | https://www.microsoft.com |
| Software, algorithm | ProP | DTU Health Tech | | https://services.healthtech.dtu.dk/ |
| Software, algorithm | SignalP | DTU Health Tech | | https://services.healthtech.dtu.dk |
| Software, algorithm | Photoshop CS6 | Adobe | CS6 | https://www.adobe.com |
| Software, algorithm | SMART | EMBL | EMBL | |
| Software, algorithm | Serial Cloner | Serial basics | 2.6.1 | |
| Software, algorithm | TMHMM 2.0 algorithm | DTU Health Tech | | https://services.healthtech.dtu.dk/ |
| Software, algorithm | ZEN 2.3 | Zeiss | Zeiss | 2.3, black |

## Fly husbandry, gas-filling, and statistics

For collection of *D. melanogaster* embryos and larvae, flies of the desired genotype took place at 25°C for collection of embryos for RNAi-mediated knockdown. All used fly strains are listed in the Materials and methods section in the supplement. In all other cases, egg-laying took place at 22°C. The apple-juice agar plates were exchanged with fresh plates at respective points of time to obtain embryos or larvae at certain developmental stages. The mutant alleles were kept with 'green' balancers to recognize mutant embryos.

We used w[1118] as control (referred to as *wt*). Rescue experiments: *w*; pio17C, btl-Gal4/Cyo, dfd-eYFP* were mated with *w*; pio5M /Cyo, dfd-eYFP; UAS-Pio* to receive the rescue in the progenoty *w*; pio17C, btl-Gal4/pio5M; UAS-pio / +.*

For gas-filling assay, we transferred stage 17 embryos and freshly hatched larvae onto agar plates and studied those by bright-field microscopy. Significance was tested using t-tests in Excel 2019; asterisks indicate p-values (*$p<0.05$, **$p<0.01$, ***$p<0.001$); error bars indicate the standard deviation. Relevant guidelines (e.g. ARRIVE) were followed in this study.

## Embryo dechorionation, fixation, and immunostainings

In general, we analyzed for control minimum n>20 embryos, and for *pio* or other mutants n>10 embryos. Embryos were washed from the apple-juice agar plates into close-meshed nets, incubated for 3 min in a bleach solution (2.5% sodium hypochlorite) for dechorionation.

For subsequent fixation, embryos were incubated at 250 rpm for 20 min in 1 ml 10% (vol/vol) formaldehyde solution (50 mM EGTA, pH 7.0), 2 ml HEPES solution, and 6 ml heptane. The fixative was removed and 8 ml methanol added and incubated at 500 rpm for 3 min to detach the vitelline membrane. Finally, embryos were washed with methanol and stored at –20°C.

All used antibodies are listed in the Materials and methods section in the supplement. For antibody staining, embryos were 5 min washed three times with BBT followed by blocking for 30 min. Subsequently, pre-absorbed primary antibodies diluted in blocking solution were applied to the embryos

and incubated overnight at 4°C. Primary antibody was incubated overnight washed off six times with BBT. After blocking for 30 min pre-absorbed secondary antibodies diluted in blocking solution were added to the embryos and incubated for 2 hr. If required, Alexa488-conjugated cbp was added to the secondary antibody dilutions at a 1:200 dilution for staining of chitin. Finally, embryos were washed six times with PBT for 5 min and mounted either in Prolong mounting medium (Thermo Fisher Scientific) or in phenol-free Kaiser's glycerol-gelantine (Carl Roth).

## Light microscopy, image acquisition, and image analysis

For bright-field and dark-field light microscopy, we used a Zeiss Axiophot 2e upright microscope (A-PLAN 10×/0.25 and 25×/0.8 LCI Plan Neofluar Objectices and Zeiss Glycerin) and images were acquired with an AxioCam (HRc/mRc) and the acquisition software (Zeiss Axiovision 12). For handling confocal analysis, we used ZEN software (2.3, SP1 black) and Zeiss LSM780-Airyscan (Zeiss, Jena) microscopes. Overview imaging were taken with 25×/0.8 LD-PLAN Neofluar and magnifications with Plan-Apochromat 40×/1.4 Oil DIC M27 and with 63×/1.3 PLAN Neofluar M27 objectives and Zeiss oil/water/glycerol medium. For imaging standard ZEN confocal microscopy (Pinhole Airy1) settings and image processing (maximum intensity projection and orthogonal section) and orthogonal projections were used.

ZEN black 2.3 fluorescence intensity profile was used with the arrow toolset across the tube lumen to measure and compare the fluorescence pixel intensities of individual channels. For quantification, we determined the fluorescent intensities profile across the tube to identify values at apical membrane and tube lumen at a minimum 10 different positions of DTs (metameres 5–6) of two distinct embryos for each genetic background. The maximum values of membranes versus tube lumen were set into ratio and compared between control, *mega* mutant, and Cht2 overexpression. Statistical analysis was calculated with Excel 2019. To quantify protein co-localization and overlap in confocal and airyscan images, we used ZEN black 2.3 co-localization tool, set two channels for comparison, and considered quadrant 3, which represents a pixel with high-intensity levels in both channels, as co-localized pixels. Those pixels were set to white for better visualization in the images.

The 'express' deconvolution of SVI Huygens pro was used with standard settings. Images were transferred to ZEN for further analyses. For Airyscan acquisition, standard ZEN settings and optimal Z-stack distances were used. For 3D visualization (HP Z4 graphics workstation), deconvolved confocal and Airyscan Z-stacks were processed in Imaris (version 9.7.2) to convert voxel-based data into surface objects. We cropped Images with Adobe CS6 Photoshop and designed figures with CS6 Illustrator. For equipment details, see https://bioimaging.uni-leipzig.de/equipment.html.

## CRISPR/Cas9-mediated mutagenesis and genome editing

Using CRISPR/Cas9-mediated mutagenesis, we generated three independent *pio* lack of function alleles [*pio^17C^*, *pio^5M^*, and *pio^11C^*] and one *flz* lack of function allele (*flz^C1^*) that carry frameshift mutations in the ORF. The *pio* mutations led to truncated Pio proteins containing only short N-terminal stretches but lacking all critical for Pio function. These mutations in *pio^17C^*, *pio^5M^*, and *pio^11C^* alleles and *flz^C1^* allele caused embryonic lethality. The in vivo analysis of airways in *pio^17C^*, *pio^5M^*, and *pio^11C^* homozygous and *pio^17C^*/*pio^5M^* transheterozygous mutant stage 17 embryos revealed lack of tracheal air-filling. The sequences of sgRNAs are listed in the Materials and methods section in the supplement.

The CRISPR Optimal Target Finder tool was used to identify specific sgRNA targets in the *D. melanogaster* genome. DNA oligos corresponding to the chosen sgRNA target sequences were annealed and ligated into the *pBFv-U6.2* vector via BbsI restriction sites (*Kondo and Ueda, 2013*). For simultaneously targeting two sgRNA recognition sites, one sgRNA target encoding annealed DNA oligo was ligated into the *pBFv-U6.2* vector and the other one into the *pBFv-U6.2B* vector. Subsequently, the *U6.2* promotor and sgRNA target sequence of the *pBFv-U6.2* vector were transferred to the sgRNA target sequence containing *pBFv-U6.2B* vector via EcoRI/NotI endonuclease restriction sites. Vectors for sgRNA expression were injected into *nos-Cas9-3A* embryos by BestGene. Adult flies that developed from the injected embryos were then crossed with balancer chromosome strains. Single flies of the resulting F1 generation were again crossed with balancer chromosome strains and the resulting F2 stock was analyzed regarding lethality of the putatively mutated chromosome. The regions of the genomic sgRNA target sites from the obtained lethal stocks were amplified by PCR followed by purification of the amplicon and sequencing.

The CRISPR/Cas9 system was used for targeted insertion of transgenes by homology-directed repair. Sequences of homology arms (HAs) were amplified by PCR from genomic DNA of *nos-Cas9-3A* flies. HAs were then purified and cloned into the *pHD-ScarlessDsRed* donor vector for editing of *pio*. A mCherry encoding sequence was inserted into the *pio* ORF at the 3' end of the 5' HA in the *pHD-ScarlessDsRed* donor vector. The donor vector together with the vector for sgRNA expression were injected into *nos-Cas9-3A* embryos by BestGene. Adult flies that developed from the injected embryos were crossed with *w** flies and resulting F1 flies were selected for the presence of *3xP3-DsRed* marker. Stocks were then established by crosses with balancer chromosome strains. Correct insertion of the transgenes was verified by amplification of the targeted genomic regions by PCR and subsequent purification of the amplicons and sequencing. The *3xP3-DsRed* marker gene that was inserted in the *pio* locus was removed by crosses with the *tub-PBac* strain and subsequent selection of F2 flies that lacked the *tub-PBac* balancer chromosome and the *3xP3-DsRed* marker gene.

## Cell culture-based experiments

*D. melanogaster* S2*R*+ cells (DGRC) and Kc167 cells (DGRC) were kept in flasks containing Schneider's *Drosophila* medium (Thermo Fisher Scientific) supplemented with 1% penicillin/streptomycin (Thermo Fisher Scientific) and 10% FBS (Sigma-Aldrich) at 25°C. Handling of cells was performed in the sterile environment of a clean bench. The *Drosophila* S2*R*+ and Kc167 cells were ordered from the Drosophila Genomics Resource Center (DGRC) where extensive authentication of the cell lines is carried out. Negative mycoplasma status was tested by PCR. Frequent observation of cell shape, growth rate, and ability to adhere to surfaces was carried out to ensure that key parameters of the cell line remained constant during passaging of the cells for ongoing experiments.

Confluent cells were detached, diluted 1:6, and transferred either to 10 cm diameter Petri dishes, or to six-well plates (Greiner Bio One), or to glass bottom micro-well dishes (MatTek; 6 ml per dish), approximately 24 hr before transfection. The UAS/Gal4 system was used for protein overexpression in cultured cells. Cells in six-well plates and glass bottom micro-well dishes were transfected with 500 ng *actin5C-Gal4* vector and 500 ng *pUAST*-responder vector (1 µg total DNA), while cells in 10 cm diameter Petri dishes were transfected with twice the amount of each vector (2 µg total DNA). The Effectene transfection reagent (QIAGEN) was used for cell transfection according to suppliers' guidelines. For transfections of cells in six-well plates and glass bottom micro-well dishes, vector DNA was mixed with 190 µl Effectene EC buffer (QIAGEN) followed by adding 8 µl Effectene Enhancer (QIAGEN), vortexing and incubation for 5 min. Subsequently, 20 µl Effectene were added and the mix was vortexed for 10 s followed by incubation for 15 min. The transfection mix was then carefully trickled onto the cells with a pipette. Incubation steps were performed at room temperature. For transfections of cells in 10 cm diameter Petri dishes with 2 µg DNA, volumes of the transfection mix components were doubled.

Cells in glass bottom micro-well dishes were incubated at 25°C for approximately 48 hr after transfection followed by imaging with a confocal LSM.

For immunostainings cells were incubated at 25°C for 48 hr after transfection and then washed with PBS twice and fixed for 15 min by adding formaldehyde solution. The cells were then washed twice with PBS and incubated in PBTX for 150 s for permeabilization. Subsequently, cells were washed three times with PBS followed by 30 min blocking. Primary antibodies diluted in blocking solution were added to the cells and incubated for 2 hr. Then cells were washed three times with PBS followed by incubation with secondary antibodies diluted in blocking solution for 1 hr. Finally, the cells were washed twice with PBS followed by mounting in Prolong mounting medium with DAPI (Thermo Fisher Scientific). Images were acquired with a confocal LSM. All incubation and washing steps were performed at room temperature.

For pull-down assays Strep-tagged Pio was co-expressed in S2*R*+ cells together with either of two independent RFP-tagged Dpy fragments for pull-down assays. One RFP-tagged Dpy contains the Dpy C-terminal region, and the second exclusively the Dpy-ZP domain. Expression products of both RFP::Dpy constructs were only pulled down together with Strep::Pio in the Strep-IP samples. For western detection see below. All used oligos to generate constructs are listed in the Materials and methods section in the supplement.

## Protein purification and western blotting

We dechorionated embryo collection, squashed them with a needle and pulsed with ultrasound for 30 s, added 25 ml 4× SDS sample buffer, heated for 9 min at 96°C, centrifuged at 11,000 rpm for 20 min. Lysate was stored in a fresh cup at −80/–20°C. Schneider's cells were manually detached, centrifuged at 900 rpm, PBS was replaced by 35 μl 4× SDS sample buffer and heated 9 min at 96°C, and stored at –80°C. For supernatant samples, we centrifuged at 15,000 rpm at 4°C for 20 min, washed with acetone and resuspended in 100 μl 1× SDS sample buffer, heating it at 96°C for 9 min. We used 4–20% gradient Mini-Protean TGX Precast Protein Gels (Bio-Rad) together with PageRuler prestained protein ladder (Thermo Scientific), MiniProtean chamber (Bio-Rad), PowerPac Basic power supply (Bio-Rad) for 40 min at 170 V. The gels were then equilibrated in transfer buffer and packed into a western blotting sandwich. Sandwich blotting was performed with a PVDF transfer membrane with 0,2 μm pore size (Thermo Fisher Scientific) for embryos lysates or in other cases, Amersham Hybond-ECL membrane with 0.2 μm pore size (GE Healthcare). Protein transfer to the membrane was performed on ice in a MiniProtean chamber (Bio-Rad) that was filled with transfer buffer at 300 mA for 90 min.

## Fluorescence recovery after photobleaching

The stage 16 embryos were dechorionated, transferred, and fixed on the coverslip of small Petri dishes (ibidi, Germany; https://ibidi.com/dishes/185-glass-bottom-dish-35-mm.html) and mounted in PBS. We used LSM780 confocal for FRAP experiments. To define the region of interest (ROI) we used the Zeiss Zen software. The bleaching was performed with 405 nm full laser power (50 mW) at the ROI for 20 s. A Z-stack covering the whole depth of the tracheal tubes in the ROI were taken at each imaging step. The confocal images were taken every 2 min until 60 min after bleaching. To correct for movements of the embryos, images that are presented in figures and supplemental movies were manually overlayed to center them in the same focal plane and to correct for movements in the X and Y axis. Fluorescence intensity in the bleached ROIs was measured after correction for embryonic movements using Fiji.

## Electron microscopy

Stage 17 *Drosophila* embryos were dechorionated, transferred to a 150 μm specimen planchette (Engineering Office M. Wohlwend GmbH), and frozen with a Leica EM HBM 100 high-pressure freezer (Leica Microsystems). Vitrified samples were embedded with an Automatic Freeze Substitution Unit (EM AFS; Leica Microsystems) at −90°C in a solution containing anhydrous acetone, 0.1% tannic acid, and 0.5% glutaraldehyde for 72 hr and in anhydrous acetone, 2% $OsO_4$, and 0.5% glutaraldehyde for additional 8 hr. Samples were then incubated at −20°C for 18 hr followed by warm-up to 4°C. After subsequent washing with anhydrous acetone, embedding in Agar 100 (Epon 812 equivalent) was performed at room temperature. Counterstaining of ultrathin sections was done with 1% uranylacetate in methanol.

Alternatively, the dechorionated embryos were fixed by immersion using 2% glutaraldehyde in 0.1 M cacodylate buffer at pH 7.4 overnight at 4°C. The cuticle of the embryos was opened by a cut to allow penetration of fixative. Postfixation was performed using 1% osmium tetroxide. After pre-embedding staining with 1% uranyl acetate, tissue samples were dehydrated and embedded in Agar 100. Ultrathin sections were evaluated using a Talos L120C transmission electron microscope (Thermo Fisher Scientific, MA, USA).

## Acknowledgements

We are very grateful to Markus Affolter, Bernard Moussian, and Stefan Luschnig for sharing generously antibodies. We thank Christian Wolf for critical reading and helpful discussion. We thank Jochen Rink and the department of Tissue Dynamics and Regeneration for generous support of our study. The FlyBase, NCBI, and SMART internet tools were always important sources. Stocks obtained from the Bloomington *Drosophila* Stock Center (NIH P40OD018537), the Vienna Drosophila Resource Center (VDRC), and from DGRC (NIH grant 2P40OD010949) were used in this study. MB expresses his gratitude to Johannes Kacza for his support with Imaris. We acknowledge support from Leipzig University for Open Access Publishing. We appreciate the University Leipzig BioImaging Core Facility (BCF

equipment INST 268/230-1; INST 268/293-1; SFB-TR67; EFRE 100192650, 100195814, 100144684) for assistance.

## Additional information

### Funding
No external funding was received for this work.

### Author contributions
Leonard Drees, Conceptualization, Data curation, Supervision, Validation, Investigation, Visualization, Methodology, Writing - review and editing; Susi Schneider, Dietmar Riedel, Validation, Investigation, Visualization; Reinhard Schuh, Conceptualization, Supervision, Funding acquisition, Writing - review and editing; Matthias Behr, Conceptualization, Resources, Data curation, Software, Formal analysis, Supervision, Funding acquisition, Validation, Investigation, Visualization, Methodology, Writing - original draft, Project administration, Writing - review and editing

### Author ORCIDs
Leonard Drees ⓘ http://orcid.org/0000-0002-3191-8709
Dietmar Riedel ⓘ http://orcid.org/0000-0003-2970-6894
Matthias Behr ⓘ http://orcid.org/0000-0001-9869-1257

### Decision letter and Author response
Decision letter https://doi.org/10.7554/eLife.91079.sa1
Author response https://doi.org/10.7554/eLife.91079.sa2

## Additional files

### Supplementary files
• MDAR checklist

### Data availability
All data confocal raw data used for this this study are either included in the manuscript as supporting files or available from the research data repository of the saxon universities, Open Access Repository and Archive OpARA, https://doi.org//10.25532/OPARA-240.

The following dataset was generated:

| Author(s) | Year | Dataset title | Dataset URL | Database and Identifier |
|-----------|------|---------------|-------------|-------------------------|
| Behr M | 2023 | proteolysis of ZP protain Pio at the apical cell membrane | http://dx.doi.org/10.25532/OPARA-240 | Open Access Repository and Archive OpARA, 10.25532/OPARA-240 |

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
