## [Editor Report]

Using the *Drosophila* tracheal system as a model for apical membrane expansion of tubes, the authors convincingly demonstrate that ectodomain shedding of the Zona Pellucida (ZP) domain protein Piopio by the Matriptase homolog Notopleural releases its linkage with the ZP protein Dumpy and thus ensures proper apical membrane function and tube expansion. Given the high degree of conservation of these proteins in other species, the results presented are important for future analysis and will have further implications on tubular development and homoestasis in other systems, including human.

---

## [Decision Letter]

[Editors' note: this paper was reviewed by Review Commons.]

---

## [Author Response]

1. General Statements

We thank the three reviewers for carefully reading our manuscript and for all considerations, ideas, suggestions, and comments. These were all very helpful for us to strengthen the scientific statements of our manuscript. Please, note that all changes are marked in red in the manuscript and supplement. Below you will find, point by point, our responses to all questions and comments.

Reviewer #1Overall, this is an exciting work. There are, however, several open questions that the authors could address to facilitate understanding of their work. These points are:1. On page 5, lines 113ff, the authors mention the membrane bulges that they analyse in figure 1. They show these deformations by light (confocal) and electron microscopy. However, the bulges seen by confocal microscopy seem to be bigger that those seen by electron microscopy. The authors could quantify the sizes of the bulges for clarification.

We quantified the size of the membrane bulges. At the confocal we measured in average 750nm as mean value of identified bulges (n=12) with 650nm as minimal and 890nm as maximal sizes. At the TEM we measure ~243nm as mean value (n=61), with a range between 62nm as minimum and 442 as maximum value. These measurements are shown as Figure 1E.

Please note that measurements of TEM images do not always capture the three-dimensionality of bulges and may show only parts of them. In addition, ultrastructure is more sensitive and can easily detect small membrane changes that we cannot observe with confocal and airsycan microscopy. In contrast, even with our high-quality objective (63x Zeiss Plan Neofluar, Glycerin, 1.3 NA), standard confocal analysis is limited at ~200nm on the XY axis (airyscan ~110nm) and ~450nm on Z-axis. Therefore, TEM analysis detects smaller bulges than confocal analysis, and consequently, this method detected a large range of bulge lengths between 63nm and 441nm.

In contrast, the airyscan method detected a range of bulge length between 0.65 and 0.83 µm. However, confocal and TEM analyses provide evidence of membrane bulges in *pio* mutant embryos. Please note that we extended our studies and now show membrane bulges in two different *pio* mutant alleles (17C and 5M) with airsycan microscopy.

2. The subject of the manuscript is rather complicated; presentation of data from Figure 1C and D on lines 113ff and 169ff is confusing.

We apologize and thank the reviewer for careful reading. We revised both paragraphs (lines 108 – 123 and lines 166 – 174) and are confident that the descriptions are now much more understandable. All changes are marked in red.

3. The quality of the sub-images of Figure 2E differs. Especially, the phenotype of the wurst, pio transheterozygous embryo is not well visible.

We apologize for it. We repeated the experiment with *wurst;pio* transheterozygotes, and generated wurst;pio double mutant embryos to improve the quality. The gas filling assay is shown in Figure 3. With brightfield microscopy in overview images (10x air objective) and closeups of the dorsal trunks (25x Glycerin objectives). Both show the gas-filling defects of dorsal trunk tubes. In a subsequent confocal analysis of chitin stainings in late-stage 17 embryos, we found that tracheal tube lumens are collapsed in the transheterozygotes and double mutant embryos.

4. Lines 246ff: the protein size are given for the mCherry:chimeric proteins; an estimate of the native Pio portions should be given.

The endogenous Pio protein has a calculated mass of about 50.82 kDa. We state it now in the according legend of Figure 6.

5. In Figure 6A, the appearance of chitin in the wildtype tube is different compared to the Np mutant situation, more filamentous. Can the authors comment on that?

The author is correct. The chitin cable formation in Np mutant embryos is normal but lacks the condensation process, and, therefore, fiber structure of the chitin matrix differs from control embryos in late stage 16 and stage 17 embryos (see Drees et al., PLOS Genetics, 2019).

6. In the Discussion section, I would appreciate if the timing of events was discussed or even shown in a model. The central question is: how are the functions of Pio and Np coordinated in time? As I understand, Np should not cleave Pio before morphogenesis is completed. Is there any example in the literature for how such an interaction could be controlled? The overexpression of Np shows that either the ratio between Np and Pio is important, or the btl promoter expresses Np at the "wrong" time point.

We thank the reviewer for this interesting comment.

Of course, we did not measure forces, but it has been published that axial forces appear at the apical cell membrane during stage 16 tube expansion. Our data show that Np cleaves Pio ZP domain and subsequent release increase during stage 16. The cleaved and released Pio enriches in the lumen during stage 16, from where cleaved Pio is internalized during stage 17 with the help of Wurst-mediated endocytosis. This is supported by several in vivo studies, video microscopy, antibody stainings and biochemical data, such as the interaction of Pio and Dumpy as well as the identification of different Pio products with and without Np cleavage. Moreover, we found membrane bulges that increase in size during stage 16 and identified a subsequent tear-off of the chitin matrix in Np mutant embryos. Thus, we propose that Np is required to cleave Pio-Dumpy linkages at the membrane-matrix when tubes elongate and postulated forces appear at the cell membrane during tube elongation in stage 16 embryos.

We stated this in the discussion as follows:

“The membrane defects observed in both Pio and Np mutants indicate errors in the coupling of the membrane matrix due to the involvement of Pio (Figures 1,7). …, the large membrane bulges in *Np* mutants affect the membrane and the apical matrix (Figure 7). Since apical Pio is not cleaved in *Np* mutants (Figure 7D), the matrix is not uncoupled from the membrane as in *pio* mutant embryos but is likely more intensely coupled, which leads to tearing of the matrix axially along the membrane bulges (Figures 7, 9), when the tube expands in length.”

How could Np be regulated at the membrane? Np is a zymogen that very likely undergoes ectodomain shedding for activation, similar to what has been described for matriptases. Additionally, human matriptase requires transient interaction of the stem region with its cognate inhibitor HAI-2, which *Drosophila* lacks (see Drees et al., PLOS Gen, 2019). Thus, the regulation of Np activation is not known.

Further, we observed that Dumpy is not degraded in Np mutant embryos during stage 17.

Nevertheless, in a previous publication, we showed that btl-G4 driven Np expression rescues Np mutant phenotypes in a time-specific manner. We used the btl-G4 driver line for these rescue experiments to express Np in tracheal cells. This restored tracheal Dumpy degradation in Np mutant embryos. Thus, btlG-G4 driven Np overexpression is able to rescue Np mutant tracheal phenotypes in a time-specific manner, although Gal4 is expressed from early tracheal development onwards. Further, btl-Gal4 driven Np expression mimics the endogenous Np, which is expressed from stage 11 onwards in all tracheal cells throughout embryogenesis (see Drees et al., PLOS Gen, 2019).

Based on these experiments, we conclude that the btl-G4-driven Np overexpression can cleave Pio ZP domain in stage 16 embryos at the correct time.

However, the ratio of Np expression and Pio is essential in the way that btl-Gal4 driven Gal4 Np overexpression may cause cleavage of a higher number of Pio proteins and the release of critical Pio-Dumpy linkages at the cell membrane and matrix. Thus, increased Pio shedding into the lumen reduces Pio linkages at the membrane, resulting in a *pio* mutant like tracheal overexpansion in btl-Gal4 driven Gal4 Np overexpression.

Finally, we were able to prove the reviewer’s question in a new experiment. We used btl-Gal4 driven UAS-Np embryos for Pio antibody staining. This revealed Pio enrichment at the tracheal chitin cable in stage 14 and 15 embryos. In contrast, stage 16 embryos showed numerous Pio puncta appearing across the entire tube lumen, indicating that Np mediates Pio shedding specifically in stage 16 embryos and not before. This Np-controlled Pio releases modifies tube length control.

Therefore, we stated this in the manuscript as follows:

Results:

“Our data assumes that Np overexpression may enhance Pio shedding in stage 16 embryos, affecting the Pio-mediated ZP matrix function. Upon *breathless (btl)*-Gal4-mediated expression of UAS-Np in tracheal cells, we observed a high amount of Pio puncta across the entire tracheal tube lumen, specifically in stage 16 embryos but not in earlier stages (Figure S13). Consistently tracheal Np overexpression led to tube overexpansion in stage 16 embryos resembling the *pio* mutant phenotype (Figure 8A,B). Thus, Np-mediated Pio shedding controls Pio function.”

Discussion:

“The *btl*-Gal4-driven Np expression mimics the endogenous Np from stage 11 onwards in all tracheal cells throughout embryogenesis (Drees et al., 2019), suggesting that Np is not expressed at a wrong time point. However, the ratio between Np and Pio is essential. We assume that Np overexpression increases Pio shedding, resulting in a *pio* loss-of-function phenotype. Thus, the tube length overexpansion upon Np overexpression indicates that Pio cleavage is required for tube length control.

Our observation that the membrane deformations are maintained in *Np* mutant embryos supports our postulated Np function to redistribute and deregulate membrane-matrix associations in stage 16 embryos when tracheal tube length expands. In contrast, Np overexpression potentially uncouples the Pio-Dpy ZP matrix membrane linkages resulting very likely in unbalanced forces causing sinusoidal tubes.”

7. Also for the discussion: We have two situations where Pio amounts/density are enhanced at the apical plasma membrane. The wurst experiments on lines 136ff show that Pio amount and density depends on endocytosis; is the wurst phenotype (Figure 2), at least partially, due to over-presentation of Pio? Likewise, in Figure 2C, there is more Pio in Cht2 overexpressing tracheae (but there is overall more Pio in these tracheae) – is actually endocytosis reduced in chitin-less luminal matrices? First: does the Pio signal at the apical plasma membrane correspond to membrane-Pio or free-Pio? Second, as in the case of wurst: would more Pio on the membrane (density) affect tracheal dimensions in Cht2 over expressing tracheae? Or are the consequences of Pio accumulation in the apical plasma membrane different in Cht2 and wurst backgrounds? Maybe cleavage of Pio and its endocytosis are dependent on its interaction with the chitin matrix. These questions connect to the question immediately above: how are the functions of the different players coordinated in space and time? We need a discussion on this issue.

We thank the reviewer for this very important idea to discuss the functions of the different players in a coordinated space and time and apologize that we haven’t done before. As this is an important point, we tried to figure out all questions raised by the reviewer and discussed it in several new paragraphs in the discussion:

"Indeed, the anti-Pio antibody, which can detect all different Pio variants, showed a punctuate Pio pattern overlapping with the apical cell membrane marker Uif at the dorsal trunk cells of stage 16 embryos. Additionally, Pio antibody also revealed early tracheal expression from embryonic stage 11 onwards, and due to Pio function in narrow dorsal and ventral branches, strong luminal Pio staining is detectable from early stage 14 until stage 17, when airway protein clearance removes luminal contents.

[…]

Heterozygous and homozygous *pio* mutant embryos generally do not show tubal collapse. However, the loss of Pio and accompanying lack of Dpy secretion in stage 17 *pio* mutant embryos led to the loss of a Pio/Dpy matrix, impacting the late embryonic maturation and differentiation of a normal chitin matrix at the apical cell surface. TEM images reveal reduced dense chitin matrix material at taenidial folds and misarranged taenidial fold pattern (Figures 1; S2), suggesting impaired taenidial function prevents tube lumen from collapsing after tube protein clearance. *Wurst* knockdown and mutant embryos do not show general tube collapse, but luminal chitin fiber organization is disturbed in stage 17 embryos (Behr et al., 2007). Therefore, transheterozygous *wurst;pio* mutant embryos may combine both defects and suffer from maturation deficits of the chitin/ZP matrix at the apical cell surface and within the tube lumen, which finally causes a high number of embryos with incomplete gas filling due to tube collapse. These maturation deficits are even more dramatic in the *wurst;pio* double mutants, which show no gas filling.”

8. The sentence on line 242ff should be rephrased: "dynamic" and "elastic" are not opposites.

We thank the reviewer for careful reading. We revised the sentence as follows:

“Our FRAP data suggest that Pio is the dynamic part of the tracheal ZP-matrix, while the static Dpy modulates mechanical tension within the matrix”.

9. A central question to me is the amounts and the density of factors in different genetic backgrounds as mentioned above. Is there any mechanism adjusting the amounts or the density of the players according to the size of the apical plasma membrane or the tracheal lumen? Pio seemingly responds to these changes.

We would like to know the molecular mechanisms that control the density of players at the apical membrane. This question is important and could be the starting point for novel scientific investigations. Mechanisms of protein trafficking, such as exocytosis, recycling and endocytosis regulate delivery and internalization of proteins at the apical cell membrane. Furthermore, protein junctions at the lateral membrane may recognize and therefore may respond to low and high mechanical stresses between cells that appear during tube length expansion. However, we did not observe any hint for misregulation of Pio expression levels in the different mutants which affect endocytosis, SJs and luminal ECM. But we observed a shift of Pio levels between apical cell membrane/matrix and lumen in *wurst*, *mega* mutants and Cht2 overexpression. This shift is analyzed with diverse ZEN tools and quantified (Figure 2D-F; Figure S4B). As discussed in the new paragraph, this shift is very likely caused by changes at the apical cell membrane and chitin matrix which impact Pio shedding. Moreover, we observe the lack of Pio release in *Np* mutants. This shows that Pio density at the membrane versus lumen depends predominantly on Npmediated cleavage. As discussed above, how Np is activated at the apical cell membrane to cleave Pio is not known.

10. The connection of Pio and taenidia is mentioned in the Results section (page 7) but not discussed.

We appreciate the careful reading and comments of the reviewer very much. We included the connection of Pio and taenidial in the Discussion section as follows:

“Taenidial organization prevents the collapse of the tracheal tube. Therefore, cortical (apical) actin organizes into parallel-running bundles that proceed to the onset of cuticle secretion and correspond precisely to the cuticle's taenidial folds (Matusek et al., 2006; Öztürk-Çolak et al., 2016). Mutant larvae of the F-actin nucleator formin DAAM show mosaic taenidial fold patterns, indicating a failure of alignment with each other and along the tracheal tubes (Matusek et al., 2006). In contrast, *pio* mutant dorsal tracheal trunks contained increased ring spacing (Figure 3A). Fusion cells are narrow doughnut-shaped cells where actin accumulates into a spotted pattern. Formins, such as Diaphanous, are essential in organizing the actin cytoskeleton. However, we do not observe dorsal trunk tube fusion defects as found in the presence of the activated diaphanous.

[…]

Heterozygous and homozygous *pio* mutant embryos generally do not show tubal collapse.

However, the loss of Pio and accompanying lack of Dpy secretion in stage 17 *pio* mutant embryos led to the loss of a Pio/Dpy matrix, impacting the late embryonic maturation and differentiation of a normal chitin matrix at the apical cell surface. TEM images reveal reduced dense chitin matrix material at taenidial folds and misarranged taenidial fold pattern (Figures 1; S2), suggesting impaired taenidial function prevents tube lumen from collapsing after tube protein clearance. *Wurst* knockdown and mutant embryos do not show general tube collapse, but luminal chitin fiber organization is disturbed in stage 17 embryos (Behr et al., 2007). Therefore, transheterozygous *wurst;pio* mutant embryos may combine both defects and suffer from maturation deficits of the chitin/ZP matrix at the apical cell surface and within the tube lumen, which finally causes a high number of embryos with incomplete gas filling due to tube collapse. These maturation deficits are even more dramatic in the *wurst;pio* double mutants, which show no gas filling.”

11. Dp remains cytoplasmic in pio mutant background – is the pio mutant phenotype due to defects by lack of Pio AND Dp function? What is the tracheal phenotype of dp mutants?

It has been discussed that *dumpy^olvr^* and *pio* mutants show similar phenotypes in early tracheal development (Jazwinska, 2003) and it has been discussed that *dumpy^olvr^* mutant embryos compromise tube size in combination with *shrub* mutants. The additional quantifications of the *dumpy^olvr^* mutant showed significantly increased tube length (Dong 2014). We used *dumpy^olvr^* mutant [In(2L)dpy^olvr^], an X-ray induced mutation of the dumpy gene locus (Wilkin 2000).

*dumpy^olvr^* mutant resemble *pio* null mutant tracheal phenotypes including detached dorsal and ventral branches and oversized tracheal dorsal trunk with curly appearance in late embryos. We included chitin and Uif staining’s of stage 16 dumpy mutant embryos (Figure S10).

This data suggest that Pio mutant phenotype is due to a lack of Pio and Dumpy, which would support our model, of Pio and Dumpy protein interaction in the extracellular space of the tube lumen.

In wt embryos Pio is predominantly in the luminal chitin cable, in contrast in *dumpy* mutant embryos most Pio is predominantly not at the luminal chitin cable. Less luminal Pio staining in *dumpy* mutant embryos but Pio accumulation apically shows that Dumpy is required for luminal Pio release in stage 16 embryos. This supports our model that Pio and Dumpy interaction may link membrane and matrix and that this link reacts on mechanical stress during tube expansion by Np-mediated cleavage of Pio and its accompanied luminal release due to linked Dumpy.

12. Lines 374ff: the reduced dorsal trunk in Np mutants is not significant; the respective statement should be formulated carefully. If we believe the statistics (no significance), this would mean that attachment of the apical plasma membrane to the luminal chitin via Pio is needed to restrict axial extension; release of Pio is needed for differentiation (taenidia formation, luminal clearance) beyond morphogenesis.

We agree with the reviewer that the reduction of the dorsal trunks in Np mutant is statistically not significant. However, the mean value is clearly below that of WT. Therefore, we revised our statement as follow: “In Np mutant embryos, tracheal dorsal trunk length shows the tends to be reduced compared to *wt* embryos.” Further, the btlG4-driven UAS-Np overexpression of Np suggests strong Pio release from the apical membrane and therefore resembles the *pio* mutant tube length overexpansion (Figure 8A,B; Figure S13). Thus, our current observations indicate that Np-mediated Pio release at the cell membrane enables precise tube length elongation. We thank the referee for discussing that Pio is needed for taenidial fold formation which would fit to our findings in *pio* null mutant embryos. *Pio* mutant embryos show the appearance of taenidial folds in stage 16 embryos (airyscan) and stage 17 embryos (TEM images). However, TEM images also show chitin matrix reduction in pio mutant stage 17 embryos. Further, costainings of Pio with Crb and Uif, as well as co-stainings of mCherry::Pio with Dpy-GFP and cbp confirms that the Pio localize at the apical cell membrane where taenidial folds form in late stage 16 embryos. Thus, our observations suggest that Pio and Dumpy are required at the apical membrane and matrix to stabilize taenidial folds and tube lumen during 17. This also includes the Np-mediated Pio release at the apical cell membrane. As requested by the referee we summarized Pio function during late tracheal development in our simplified model (see Figure 9).

However, it is of note that Np-mediated Pio release increases at late stage 16 (Figure 5A, 6D; Figure S13) but is strongly reduced in stage 17 embryos. In contrast, thin taenidial fold are formed at late stage 16 and becomes thicker and form at fusion points during stage 17 and reach their most mature form when the intraluminal chitin cable is cleared (Öztürk-Colak et al., *eLife*, 2016). Thus, the pattern of Pio release and taenidial fold differentiation do not fully match. Moreover, in preliminary experiments we observe Pio antibody staining in stage 17 embryos at the apical cell membrane of dorsal trunks (data not shown). Furthermore, lumen clearance of Obst-A, Knk, Sepr and Verm are not affected in pio mutant embryos, but unknown luminal ECM contents remained (Figure 1D). Therefore, we will follow this very interesting idea in future experiments.

Nonetheless, we state in the results that Pio shedding is essential:

“Our data assumes that Np overexpression may enhance Pio shedding in stage 16 embryos, affecting the Pio-mediated ZP matrix function. Upon *breathless (btl)*-Gal4-mediated expression of UAS-Np in tracheal cells, we observed a high amount of Pio puncta across the entire tracheal tube lumen, specifically in stage 16 embryos but not in earlier stages (Figure S13). Consistently tracheal Np overexpression led to tube overexpansion in stage 16 embryos resembling the *pio* mutant phenotype (Figure 8A,B). Thus, Np-mediated Pio shedding controls Pio function.”

13. Why don't we see the apical Pio signal in Figure 4B?

The red arrowhead points to apical mCherry::Pio punctuate staining in the Figure 5B (before 4B) in the close up of the “bleached area” before bleaching and 56min post bleaching. However, in vivo bleaching experiments do not allow additional antibody stainings to detect precisely the apical cell membrane. Further, the Dpy::eYFP marks the tube lumen and the apical cell surface. The latter showed adjacent mCherry::Pio punctuate staining. However, due to bleaching Dpy signal was not detectable in the area.

14. The Strep signals in the merges in Figure 7C are not well visible.

We are not sure which Strep signal the reviewer is referring to in Figure 7C, which is now Figure 8C.

The top panel shows the Strep signal (right panel) overlapping with GFP in cells that do not express Np or human matriptase. Thus, the TGFB3 ZP domain is not cleaved, and the intracellular GFP and also the extracellular Strep signals are maintained and overlap. In contrast, when Np or human matriptase is added, the TGFB3 ZP domain is cleaved and only the intercellular GFP signal is retained, whereas the extracellular Strep signal is released from the cell surface. This explains why the Strep signal is barely detectable in the middle and lower panels of Figure 8C.

Reviewer #1 (Significance (Required)):This work brings together several factors (Pio, Dp, Np, Wst etc) already known to be needed for tracheal morphogenesis and differentiation in the embryo of *D. melanogaster*. Having worked myself with some of these factors, however, I recognize that the interaction between these factors is novel and very exciting. The experiments strongly indicate a new mechanism of cellECM connection that seems to be conserved to some extent (as they provide preliminary data on an example from humans). By integrating the functions of different factors, the work provides ample opportunity for future projects to elucidate this mechanism in detail. Therefore, I expect that it will have a significant impact not only on the field of developmental cell biology but also, due to the conserved proteins involved (ZP proteins, Matriptase), on the field of cell biology of human diseases.Reviewer #2The figures are clear, and the questions well addressed. However, I find that some of the claims are not completely backed by the data presented and have some suggestions that will hopefully make some points clearer.Major comments1. In the abstract and at the end of the introduction the authors claim that they show that Pio, Dpy and Np support the balancing of mechanical stresses during tracheal tube elongation. However, this is not shown in this manuscript, where tension or mechanical stress were not measured and it is therefore speculative.

As requested by the reviewer, we deleted “support balancing of” at the final sentence of the Introduction. Please, note that we did not use the term balancing of mechanical stresses at the abstract.

However, we revised the abstract.

It has been shown previously that forces and mechanical tension rise when apical membrane expands and elastic extracellular matrix, which is anchored to the membrane balances theses forces (Dong et al., 2014). Furthermore, its has been shown that the gigantic and elastic Dumpy protein modulates mechanical tension (Wilkin et al., 2000). Thus, these previous publications state that mechanical tension rise at the apical cell membrane and matrix when tubes expand during stage 16 and that Dpy is part of that molecular process, which we included in the abstract as essential background information.

“The apical membrane is anchored to the apical extracellular matrix (aECM) and causes expansion forces that elongate the tracheal tubes. The aECM provides a mechanical tension that balances the resulting expansion forces, with Dumpy being an elastic molecule that modulates the mechanical stress on the matrix during tracheal tube expansion.”

Nonetheless, our results show that Np-mediated Pio cleavage increases during stage 16 as response to tube length expansion which is accompanied by forces as postulated by others (see above). We further observe that the membrane bulges and chitin matrix tear off, when Pio cleavage does not occur in Np mutant embryos. Our data further show that Pio and Dumpy interact and that Pio release is prevented in Dpy mutant embryos. Altogether this suggests that the Np-mediated Pio cleavage responds to tube expansion and requires Dpy for luminal Pio release.

We therefore claim in the final sentence of the introduction that “…ZP domain proteins Pio and, Dumpy, as well as the protease Np respond to mechanical stresses when tracheal tubes elongate”. The according changes are marked in red.

2. The authors state that all pio CRISPR/Cas9 generated mutants display identical tracheal phenotypes, however these data are not shown. Tracheal phenotypes, in particular DT phenotypes, of all mutants generated should be shown in supplementary materials.

As requested by the reviewer, we included the data in the supplement. The *pio5M* and *pio11R* alleles showed embryonic lethality and a 100% gas filling defect resembling the *pio17C* allele. Additionally, we extended the tracheal analysis with the *pio5M* allele and identified tube size defects, irregular pattern of taenidial folds and apical membrane deformation, altogether resembling the *pio17C* allele. These new data are shown in the supplement Figure S1.

We clarify this in the Results section as follows:

“The tracheal phenotypes of *pio^5m^* are shown in the supplement (Figure S1B-F). In all other Figures, we show images of the *pio^17c^* allele. “

3. At stage 16, pio null mutants display DT overelongation phenotypes (Figure 1). The authors should quantify this phenotype.

As requested by the reviewer, we quantified the DT overelongation phenotypes for *pio^5M^* (Figure S1). The quantification of pio17C was shown already in Figure 6B, now Figure 8B.

4. The authors analyse Pio distribution under tubular stress, using mega mutants and Chitinase overexpression. Pio localization changes in these genetic backgrounds and this is shown in Figure 2 only in a qualitative manner. The authors should measure Pio localization at the lumen and at the membrane and provide quantitative data.

As requested by the referee, we measured Pio localization recognized by the anti-Pio antibody at the lumen and at the membrane to provide quantitative data. These are shown in Figure 2E. All images were taken with a Zeiss Airyscan. For statistical analysis we used the profile tool of the Zeiss ZEN 2.3 black software. This tool allows the measurement and comparison of fluorescence pixel intensities of individual channels. We determined the fluorescent intensities profile across the tube to identify values at apical membrane and tube lumen at minimum 10 different position of DTs (metameres 5 to 6) of two distinct embryos for each genetic background. The maximum values of membranes versus tube lumen were set into ratio and compared between control, *mega* mutant and Cht2 overexpression. The control embryos showed a ration below 0.4, the Cht2 overexpression a ratio of 1.2 and mega mutants a ratio of about ~0.9. These quantitative data confirm the statement that Pio localization increases at and near the apical cell membrane with respect to the lumen in *mega* mutants and in Cht2 overexpression embryos.

5. Surprisingly and interestingly, wurst;pio transheterozygotes display very strong tracheal defects. The authors say they observe gas filling defects; however it is not clear from figure 2E if this indeed the case. From the panel in the figure, it looks like these embryos suffer from strong tracheal morphogenetic defects. It would be necessary to have a better analysis of these embryos. What is the penetrance of this phenotype. If this is 100% penetrant, one would expect it to be lethal. Therefore, double mutant balanced stocks are not viable? Having analyzed the phenotypes and confirmed which morphogenetic defects the transheterozygote embryos present, how does this genetic interaction fit with the model presented?

We are thankful to the reviewer for this interesting point of view suggesting that the *wurst;pio* embryos display tracheal morphogenetic defects. First, our data show that only 11.6% of the *wurst;pio* transheterozygous embryos completed gas filling and survived until adulthood. In contrast, 88.4% of transheterozygous *wurst;pio* mutant embryos did not complete gas filling which is now presented in Figure 3B. The corresponding quantifications is presented in Figure 3D. Importantly, the 88.4% wurst;pio transheterozygous embryos which show gas filling defects do not hatch as larvae and die.

As requested, we performed a better morphogenetic analysis, which is presented in Figure 3C.

Analysis of the gas filling defects with light microscopy were repeated with a better objective (Zeiss Apochromat 25x Gly; 0.8 NA). Indeed, this analysis revealed a strongly compromised tube lumen morphology with irregular tube lumen pattern as if tubes twist and bend. This tube lumen deformation was further confirmed with the confocal analysis of chitin staining (cbp). The tube lumen of stage 17 transheterozygous *wurst;pio* mutant embryos showed irregular lumen pattern with unusual twists and even partially collapsed tubes.

Furthermore, as asked by the referee, we generated the *wurst,pio* double mutation. All *wurst,pio* double mutant embryos lacked gas filling. In a more in-depth analysis of the tube lumen with a high-performance objective we could not identify any normal tube lumen in stage 17 embryos. Instead the double mutant embryos revealed completely collapsed tracheal tubes. This was confirmed by the chitin staining and confocal analysis. All new data are presented in the supplement.

As shown in our manuscript and in previous publications, neither *pio* nor *wurst* mutant embryos affect cell polarity or gross organization of the actin and tubulin cytoskeleton. However, we found that wurst mutant embryos showed irregular apical membrane expansion at tube lumen (Behr et al., 2007; legend Figure 4), irregular chitin fiber organization and to some extend collapsed tube lumen. In *pio* mutant embryos we found deformed apical membrane of DTs, irregular pattern of taenidial folds and to some extend collapsed tube lumen. Thus, the apical membrane is their common target of both proteins in late embryonic development, suggesting that *pio* functions provide stability and *wurst* functions the internalization of proteins at the apical membrane.

We discussed it as follows:

“Nevertheless, Pio and its endocytosis depend on its interaction with the chitin matrix and the Np-mediated cleavage. In stage 16 *wurst* and *mega* mutant embryos, we detect Pio antibody staining at the chitin cable, suggesting that Pio is cleaved and released into the dorsal trunk tube lumen. Also, the Cht2 overexpression did not prevent the luminal release of Pio. However, reduced *wurst*, mega function, and Cht2 overexpression caused an enrichment of punctuate Pio staining at the apical cell membrane and matrix (Figures 1,2). Although the three proteins are involved in different subcellular requirements, they all contribute to the determination of tube size by affecting either the apical cell membrane or the formation of a well-structured apical extracellular chitin matrix, indicating that changes at the apical cell membrane and matrix in stage 16 embryos affect the Pio pattern at the membrane. It also shows that local Pio linkages at the cell membrane and matrix are still cleaved by the Np function for luminal Pio release, which explains why those mutant embryos do not show *pio* mutant-like membrane deformations and Np-mutant-like bulges. This is in line with our observations that tracheal Pio overexpression cannot cause tube size defects as the Np function is sufficient to organize local Pio linkages at the membrane and matrix. Therefore, it is unlikely that tracheal tube length defects in *wurst* and *mega* mutants as well as in Cht2 misexpression embryos are caused by the apical Pio density enrichment.”

“Heterozygous and homozygous *pio* mutant embryos generally do not show tubal collapse. However, the loss of Pio and accompanying lack of Dpy secretion in stage 17 *pio* mutant embryos led to the loss of a Pio/Dpy matrix, impacting the late embryonic maturation and differentiation of a normal chitin matrix at the apical cell surface. TEM images reveal reduced dense chitin matrix material at taenidial folds and misarranged taenidial fold pattern (Figures 1; S2), suggesting impaired taenidial function prevents tube lumen from collapsing after tube protein clearance. *Wurst* knockdown and mutant embryos do not show general tube collapse, but luminal chitin fiber organization is disturbed in stage 17 embryos (Behr et al., 2007). Therefore, transheterozygous *wurst;pio* mutant embryos may combine both defects and suffer from maturation deficits of the chitin/ZP matrix at the apical cell surface and within the tube lumen, which finally causes a high number of embryos with incomplete gas filling due to tube collapse. These maturation deficits are even more dramatic in the *wurst;pio* double mutants, which show no gas filling.”

6. mCherry::Pio Dpy::eYFP time lapse analysis and FRAP experiments is very interesting. However, it is not clear to which degree bleaching occurs in the tracheal lumen. The authors claim that recovery is very fast and can be seen from minute 2, however, frame-by-frame analysis of Movie S2 does not show a clear different between luminal Pio from minute 0 to minute 2. Rough comparison with the luminal area surrounding the bleached area, does not show a clear difference in luminal Pio before and after photobleaching. To claim fast recovery of luminal Pio after photobleaching, the authors should quantify luminal Pio, before and after bleaching.

We agree with the reviewer and deleted “fast”. The Video2 shows intracellular mCherry::Pio recovery within 2min after photobleaching. The Video 2 shows extracellular (luminal) recovery within 6min after photobleaching, when first large mCherry::Pio puncta appear at the apical surface of the bleached area. Nonetheless, mCherry::Pio puncta appear in the lumen indicating recovery, whereas Dpy::eYFP did not.

We state this in the Results section as follows:

“In stage 16 embryos mCherry::Pio puncta reappeared in tracheal cells within 2 minutes of bleaching and in the tubular lumen within 6 minutes.”

In addition, in figure 4D, the normalized mCherry::Pio fluorescence in the graph what does it refer to? Intracellular Pio?

Figure 4D, now 5D, shows Western Blot signals. We guess that you refer to Figure 4B which is Figure 5B.

We are sorry for confusion and named it now Figure 5B’.

We stated in the Material section:

“The bleaching was performed with 405nm full laser power (50mW) at the ROI for 20 seconds. A Z-stack covering the whole depth of the tracheal tubes in the ROI were taken at each imaging step. “Fluorescence intensity in the bleached ROIs was measured after correction for embryonic movements using Fiji.”

Thus, to clarify this point, we added to the legends:

“Fluorescence intensities refer to the bleached ROIs as indicated with the frame in corresponding Movie S2 and was measured after correction for embryonic movements.”

7. When mCherry::Pio Dpy::eYFP time lapse analysis and FRAP experiments was done in an Np mutant background, the authors describe lack of Pio recovery within the lumen (Movie S3). However, when comparing control and Np mutant background embryos, Pio is not properly released into the lumen of Np mutants (as stated by the authors and seen by comparing movies S1 and S4). Furthermore, on minute 0 of the FRAP experiment in Np embryos, there is no detectable Pio in the DT lumen. Therefore, recovery was not expected in Np mutants and should not be claimed as a conclusion for this experiment.

We thank the reviewer for careful reading and apologize our wrong description. We changed it accordingly as follows:

“In contrast to the control, extracellular mCherry::Pio is not released into the tube lumen within 56 min after bleaching in *Np* mutant embryos (Figure 6C, Video S3).”

8. Brodu et al. (Dev Cell 2010) have shown that Pio is important for cytoskeletal modulation during tracheal maturation. Pio is important for non-centrosomal microtubule (MT) arrays anchored at the tracheal cell apical membranes. In addition, MT disruption in tracheal cells leads to lumen formation defects (Brodu et al., Dev Cell 2010). In the absence of Pio, the tracheal cytoskeleton is altered, and this could explain some of the results observed. Ideally, the work should be complemented with a basic cytoskeletal analysis, but if this is not possible, the authors should discuss some of the phenotypes in light of this Pio function.

Dear reviewer, this is a great idea. Therefore, we analyzed F-actin with Phalloidin and β tubulin (E7 antibody, DSHB) in the dorsal trunk cells of stage 16 control and *pio* mutant embryos. However, tracheal cells are tiny and only gross irregularities can be realized. So, confocal Z-stack analysis of the stainings did not show gross differences between control and *pio* mutant embryos. We observe the expected apical subcortical accumulation for the actin and tubulin cytoskeleton in dorsal trunk cells of *pio* stage 16 mutant embryos which also has been shown for wt embryos elsewhere. These new data are presented in the supplement Figure S7.

Minor commentsThe model should not be in supplementary materials and should be moved to the main manuscript.

We thank the reviewer for this suggestion and moved the model to the main part – now Figure 9. As requested by the reviewer 1, we extended the model, showing the timing events of Pio function.

Throughout the manuscript embryonic stages are described using different nomenclature (stage X, stX and st X). Either way is correct, but the same nomenclature should be used throughout.

We apologize for the different nomenclature and use "stage X" in the manuscript and "stX" in the figures for space reasons. Legend 1 clarifies the abbreviation.

In Figure S1 B and C the authors should specify which pio allele is being analysed (as in Figure 7).The same should be done in the text.

That's a fairly good point. To be clear from the beginning, we now state the following in the first paragraph of the results:

“The tracheal phenotypes of *pio^5m^* are shown in the supplement (Figure S1B-F). In the all other Figures, we show phenotypes of the *pio^17c^* allele.”

Line 131, it is not correct to say that WGA visualizes cell membranes. WGA marks/stains cell membranes.

Thanks for finding this mistake, it’s now corrected.

Line 165 "leads to excessive tube dilation and length expansion due to strongly reduced luminal chitin" is not correct. Chitin reduction leads to excessive tube dilation but not to length expansion, as reported in the papers cited at the end of the sentence.

Thanks very much for careful reading, we deleted “and length expansion” from the sentence.

Line 220-221, what do authors refer to as "stage 16 wt-like control embryos"?

Thanks for finding these mistakes. We corrected as follows:

“In stage 16 embryos mCherry::Pio puncta….”

Line 221, "some minutes" should be replaced by a specific number of minutes. According to Movie S2 reappearance of tracheal cell Pio happens from minute 16.

We agree with the reviewer to state the time when mCerry::Pio puncta reappear. We observe first large puncta within two minutes after bleaching in tracheal cells at the ROI (Video S2, lower cell row at the movie). We further observe the reappearance of first large puncta at the ROI within 6 minutes in the tracheal tube lumen.

We corrected it as follows: “In stage 16 embryos mCherry::Pio puncta reappeared in tracheal cells within 2 minutes of bleaching and in the tubular lumen within 6 minutes.”

Line 291 "time laps" should be lapse.

Thanks for finding the typo, it is corrected now.

Line 302, "Pio was not shedded into the lumen but remained at the cell" should be "Pio was not shed into the lumen but remained in the cell".

Thanks for finding the typo, it is corrected now.

Referees cross-commentingI agree. Taken together, all the comments will improve the quality of the work and of a future manuscript. Also, everything seems quite doable and will not present any problems.Reviewer #2 (Significance (Required)):The findings shown in this manuscript shed light on the regulation of tubulogenesis by ZP proteins and how their interaction with the ECM can be regulated by proteolysis. It was known that Pio is involved in tracheal development, is secreted into the lumen, regulating tube elongation (Jaźwińska et al., Nat.Cell Biol., 2003) and anchoring MTs to the apical membrane during tubulogenesis (Brodu et al., Dev. Cell 2010). This work provides additional molecular insights into Pio dynamics and regulation during tube maturation.This work will be of interest to a broad cell and developmental biology community as they provide a mechanistic advance in ZP proteins involved in morphogenesis. It is of specific interest to the specialized field of tubulogenesis and tracheal morphogenesis.Field of expertise:*Drosophila*, morphogenesis, tracheal tubulogenesis, cytoskeletonReviewer #3SummaryIn this manuscript, Drees and colleagues analysed, during the formation and growth of tubular systems, how cells combine forces at the cell membranes while maintaining tubular network integrity. A fundamental question is to understand how cells manage to integrate the axial forces to stabilise the cell membrane and the apical extracellular matrix (aECM).To address this question, the authors study the formation of the tracheal system in *Drosophila* embryos, a well-established and detailed model system to investigate formation of tubular networks. In particular, they focused on the formation of the larger tube of the tracheal network, the dorsal trunk. The formation of this tube depends in part of axial extension along the anteroposterior axis.They concentrated their work on the function of Piopio (Pio), a Zona-Pellucida (ZP)-domain protein. They showed that Pio together with the protease Notopleural (Np) contribute the sense and support mechanical stresses when tracheal tubes elongate, thus ensuring normal membrane -aECM morphology.Major CommentsIn a previous work, Drees et al. (PLOS Genetics 2019), showed the matriptase-prostasin proteolytic cascade (MPPC), is conserved and essential for both *Drosophila* ECM morphogenesis and physiology.The functionally conserved components of the MPPC mediate cleavage of zona pellucidadomain (ZP-domain) proteins, which play crucial roles in organizing apical structures of the ECM in both vertebrates and invertebrates. They showed that ZP-proteins are molecular targets of the conserved MPPC and that cleavage within the ZP-domains is a conserved mechanism of ECM development and differentiation.Here, Drees et al. investigate further how the coupling between membrane and matrix takes place to ensure proper tube growth.Pio distribution and phenotypesThey first focused on the tracheal phenotypes observed in a pio null mutant context. So far, the only pio mutant characterised was a point mutation in the ZP domain. Using CRISPR/Cas9, they generated new alleles of pio which are lack of function alleles. In the context, Drees and colleagues observed over-elongated dorsal trunk tubes, with bulges appearing at stage 16 between the apical domain of tracheal cells and adjacent extra-luminal matrix.Additionally, pio mutant embryos showed impaired tube lumen clearance of the some of the aECM components, which prevent gas-filling of the airways.To detect Pio distribution, the authors used either anti-Pio antibody directed toward a short stretch with the Pio ZP domain or generated a CRISPR/Cas9 piomCherry::pio line.1. The Pio antibody shows a strong luminal staining as already published. But the authors reported an apical membrane signal in tracheal cells. I find this apical membrane signal really difficult to observe in panel Figure 2B. The overlap between the Pio dots and the apical membrane labelled with Uif showed in Figure 2C can be due to the 3D projection. It is only when endocytosis is unpaired (Suppl Figure 2), that data are more convincing.

We thank the reviewer for this important point, we are sorry for the unconvincing presentation and for having the chance to improve it.

We show the 3D image of Pio puncta as voxels overlapping with Uif at the apical cell membrane. The amount of Pio voxels overlapping with the Uif marked apical cell membrane increased in *mega* mutant and due to tracheal Cht2 overexpression. This result was indicated by a representative region (frame) and white arrows and is shown now in Figure 2C. We further used orthogonal projections across the tracheal tube of the airyscan Z-stacks. Random usage confirmed that puncta of Pio antibody staining overlap with Uif at the tube lumen. We observed overlap in controls, but increasing overlap in *mega* mutant and Cht2 overexpressing embryos. This result is shown now in Figure 2E.

However, to overcome any misinterpretations of projections, we used single images of the original airyscan Z-stacks for co-localization analysis with the Zeiss ZEN software (black, 2.3, sp1). We used two available and independent standard methods to compare fluorescence pixel intensities of different channels namely the ZEN co-localization and the ZEN profile tool. Both are described in the Materials section.

With the co-localization tool we compared directly fluorescence pixel intensities of Pio and Uif. Highest overlap of the intensities, shown in the ZEN tool as third quadrant, were set to white for better visualization in the images. These new images are included as Figure 2D and show recurrent overlap of Pio and Uif antibody stainings (punctuate pattern) along the apical cell membrane at the dorsal trunk of stage 16 control embryos. This overlap pattern increased in *mega* mutant and Cht2 overexpression embryos.A second approach for comparing fluorescence intensities is the ZEN “profile” tool. Drawing a line across the tube allowed us to compare peak fluorescence pixel intensities of the different channels at distinct regions, such as the apical cell membrane and the tube lumen including the cbp marked chitin cable. This tool detected overlap of peak fluorescence intensities of UIF and Pio antibody staining’s, confirming that Pio is located together with UIF at the apical membrane of dorsal trunk tracheal cells. These new intensity profiles and the corresponding images are presented in the supplement as Figure S4B-D. Quantifications of this method comparing the ration of Pio peak intensities between the apical cell membrane and the tube lumen are presented as Figure 2F (as requested by Reviewer 2).

2. When the author used their CRISPR/Cas9 piomCherry::pio line to characterise Pio distribution (Figure 4), Pio is localised at the apical plasma membrane before stage 16. Only at stage 16, Pio is detected within the lumen. This timing of Pio release in the lumen is critical for the model proposed by Drees at al. This is an important point to assess the difference between the use of the antibody (which mostly label the lumen) while piomCherry::pio line is mostly at the membrane.

We agree with the reviewer that the Pio antibody shows a different pattern within the tube lumen of earlier stages. The Pio antibody shows intense extracellular staining from early stage 12 onwards, presumably due to its early function at dorsal and ventral branches, as shown by Anna Jazwinska (Jazwinska et al., 2003). The intense luminal Pio antibody staining, predominantly at the chitin cable, persist until its disappearance due to airway protein clearance during stage 17. Unfortunately, this strong luminal Pio staining made it impossible to examine the Pio distribution pattern in more detail during stage 16. Nevertheless, Np overexpression experiments indicate that luminal Pio release occurs specifically in stage 16 embryos (Figure S13), which was tested with the Pio antibody, see results, second last paragraph:

“Our data assumes that Np overexpression may enhance Pio shedding in stage 16 embryos, affecting the Pio-mediated ZP matrix function. Upon *breathless (btl)*-Gal4-mediated expression of UAS-Np in tracheal cells, we observed a high amount of Pio puncta across the entire tracheal tube lumen, specifically in stage 16 embryos but not in earlier stages (Figure S13).”

We further agree with the reviewer that mCherry::Pio was used to characterize in vivo Pio distribution within the dorsal trunk cells and tube lumen during stage 16. The Figure 5A shows apical mCherry::Pio distribution pattern in early and late stage 16 embryos. Importantly, the appearance of luminal mCherry::Pio increased during stage 16 and mainly enriched at late stage 16. See Figure 5A, red arrowheads point to apical Pio and red arrows to luminal Pio staining.

Furthermore, as discussed above and shown by different ZEN tools, such as co-localization and fluorescence intensity profile tools, Pio antibody stainings revealed a punctuate pattern at the apical cell membrane of dorsal trunk cells in stage 16 embryos, which is reflected also by the appearance of apical mCherry::Pio puncta at the membrane surface. Additionally, we observed mCherry::Pio puncta also within the tube lumen (see the new Figures S4B & S8). Thus, subcellular Pio distribution at the apical cell membrane and lumen were observed for both, Pio antibody staining and mCherry::pio pattern.

Nonetheless, there is different luminal appearance between the Pio antibody staining and mCherry::Pio. Pio antibody detects a short stretch at the ZP domain and thus detects all possible Pio variants, uncleaved and cleaved. Due to early tracheal Pio function, Pio enriches within the tube lumen in an intense core-like structure, which is recognized by the Pio antibody and is comparable with the Dpy::eYFP pattern. Also mCherry::Pio labels all Pio variants, uncleaved and cleaved. The spatial temporal mCherry::Pio expression pattern (Figure S5) is comparable with the Pio antibody pattern and the staining at the membrane in stage 16 embryos. However, mCherry::Pio did not enrich in the lumen in a core-like structure, nonetheless, shows overlap with luminal Dpy::eYFP.

Jaswinska showed that Pio antibody staining is intracellular in the trachea of stage 11 *pio^2R-16^*

point mutation embryos (Jaswinska et al., 2003; Figure 2d). To understand more about the specificity of the antibody, we performed stainings in the null mutant embryos. In contrast, to the high number of intracellular Pio puncta in *pio^2R-16^* point mutation embryos, Pio stainings were much more reduced in *pio^5m^* and *pio^17c^* mutants, but a low number of Pio puncta were still detectable in the embryos (Figure S1G,H). It is of note that also *dpy* mutants showed strongly reduced Pio antibody staining (Figure S10E). Thus, discussing underlying causes of enriched (Pio antibody) versus non-enriched (mCherry::Pio) luminal staining are speculative. However, observations by Jaswinska et al. (2003) and our new observations, investigating the Pio antibody stainings in *pio* null mutants, *dpy* mutants, eYFP::Dpy embryos and NP overexpression may hint to the possibility of cross-reactivity of the Pio antibody to other ZP domains which may intensify the appearance of luminal Pio antibody staining in control embryos.

Anyway, we clarify the difference in luminal Pio pattern in the discussion as follows:

“Indeed, the anti-Pio antibody, which detects all different Pio variants, showed a punctuate Pio pattern overlapping with the apical cell membrane markers Crb and Uif at the dorsal trunk cells of stage 16 embryos (Figure 2; Figure S3,S4). Additionally, Pio antibody also revealed early tracheal expression from embryonic stage 11 onwards, and due to Pio function in narrow dorsal and ventral branches, strong luminal Pio antibody staining is detectable from early stage 14 until stage 17, when airway protein clearance removes luminal contents. In the *pio^5m^* and *pio^17c^* mutants Pio stainings were strongly reduced although some puncta were still detectable in the trachea (Figure S1G,H). Similarly, Pio antibody staining is intracellular in the trachea of stage 11 *pio^2R-16^* point mutation embryos (Jaźwińska et al., 2003). Interestingly, also *dpy* mutants showed strongly reduced and intracellular Pio antibody staining (Figure S10E).

We generated mCherry::Pio as a tool for in vivo Pio expression and localization pattern analysis during tube lumen length expansion. The mCherry::Pio resembled the Pio antibody expression pattern from early tracheal development onwards. However, luminal mCherry::Pio enrichment occurs specifically during stage 16, when tubes expand. The stage 16 embryos showed mCherry::Pio puncta accumulating apically in dorsal trunk cells. Moreover, mCherry::Pio puncta partially overlapped with Dpy::YFP and chitin at the taenidial folds, forming at apical cell membranes. Supported by several observations, such as antibody staining, Video monitoring, FRAP experiments, and Western Blot studies (Figures 4,5), these findings indicate that Pio may play a significant role at the apical cell membrane and matrix in dorsal trunk cells of stage 16 embryos.”

3. Another important point is to explain the discrepancy between the pio mutant alleles. The allele containing a point mutation in the ZP domain shows no over-elongated tubes (Dong et al. 2014, Jazwinska et al. 2003) while the lack of function alleles does.

The reviewer is correct that the *pio^2R-16^* mutation shows only a disintegration phenotype whereas our *pio* null mutations show in addition tube length defects. However, Dong et al. showed significantly increased dorsal trunk length in *shrub; pio^2R-16^* double mutant embryos when compared with *shrub* mutant embryos (Supplemental Figure S4A). Also, the *shrub;dpy^olvR^* double mutant embryos revealed increased tube length expansion when compared with *shrub* mutant embryos. Moreover, their quantifications show that the also *dpy^olvR^* mutant embryos revealed significantly increased tube expansion when compared with wt. Altogether these previous findings suggests that Pio and Dpy are involved in controlling tube length control during stage 16.

Furthermore, we generated three independent *pio* null mutation alleles, which lost all the essential Pio protein domains, and caused all embryonic lethality, gas-filling defects, branch disintegration phenotype and tube length defects (quantifications are shown in Figures 9 and S1). In addition, *pio* null mutations prevent Dpy::eYFP secretion. Thus, we are confident that the observed tube length defects as well as the air-filling defects are due to the loss of Pio, and in particular since these defects could be rescued by Pio Expression in the *pio* null mutation background, as shown in Figure 3B.

So, what could make the difference?

The described *pio^2R-16^* mutation allele contains a X-ray induced single point mutation that led to an amino acid replacement (V159D) in the ZP domain. It is not clear how the amino acid exchange affects the protein and the ZP domain. It may hamper *pio* function and maybe this amino acid replacement is problematic for the early tracheal function but not during stage 16. As stated by Jazwinska et al. 2003 (Figure 2 legend), Pio antibody staining is intracellular in the mutants and extracellular in the trachea of wt at stage 13.

They further speculate that the mutant Pio protein may retain in the secretory pathway, but this is not confirmed with co-markers. As luminal Pio function is required to provide a barrier for autocellular AJ formation, this fails in *pio^2R-16^* mutation. In contrast, it is still possible that Pio interacts and supports Dpy secretion in *pio^2R-16^* mutation and additionally it is thinkable that intracellular Pio may reach to some extend the apical cell membrane in *pio^2R-16^* mutation stage 16 and thus can support tube size control. But these assumptions are speculations.

Nevertheless, to clarify this point we explain the discrepancy between the *pio^2R-16^* mutation and *pio* null mutations alleles as follows:

“Using CRISPR/Cas9, we generated three *pio* lack of function alleles (Figure S1A), all exhibiting embryonic lethality and identical tracheal mutant phenotypes. The tracheal phenotypes of *pio^5m^* are shown in the supplement (Figure S1B-F). In all other Figures, we show images of the *pio^17c^* allele. The *pio^17c^* and *pio^5m^* null mutant embryos revealed the dorsal and ventral branch disintegration phenotype known from a previously described *pio^2R-16^* mutation allele which contains a X-ray induced single point mutation that led to an amino acid replacement (V159D) in the ZP domain (Jaźwińska et al., 2003). Additionally, the late stage 16 *pio^17c^* and *pio^5m^* null mutant embryos showed over-elongated tracheal dorsal trunk tubes (see below).”

4. A minor point, the author should provide hypothesis to explain why only the clearance of CBP, Obstructor-A and Knickkopf are affected in a pio mutant background and not Serpentine and Vermiform.

We thank the reviewer for careful reading and the comment on this point. We would be happy to see such a scenario which could give us a hind of Pio interaction partners at the chitinous matrix. However, we stated that luminal material, such as Obst-A and Knk are removed from the lumen (see Figure S5A). We further describe that in *pio* mutant embryos, luminal Serp and Verm staining appeared reduced but showed *wt*-like distribution (see Figure S6) in stage 16 embryos.

We do not show Serp and Verm in stage 17 embryos, but they are removed from the tube lumen (not shown). These data are received from immune-staining’s and confocal analysis. Nevertheless, we also state that *pio* mutant embryos revealed lumen clearance defects in TEM analysis, of undefined material in the tube lumen (see Figure 1D and Figure S2B).

To clarify this point we state in the results as follows:

“Fourth, ultrastructure TEM images revealed aECM remnants in the airway lumen of *pio* mutant stage 17 embryos, while control embryos cleared their airways (Figure S2B). Consistently, the in vivo analysis of airways in stage 17 *pio* mutant embryos revealed lack of tracheal air-filling (Figure 3B). The pan-tracheal expression of Pio in *pio* mutant embryos rescued the lack of gas filling (Figure 3B). Thus, TEM images suggest that *pio* mutant embryos showed impaired tube lumen clearance of aECM, which prevented subsequent airway gas-filling. “

And

“Also, the *pio* mutant embryos showed tracheal lumen clearance defects of chitin fibers in ultrastructure (TEM) analysis (Figures 1D, S2B). In contrast, confocal analysis revealed that wellknown chitin matrix proteins, such as Obstructor-A (Obst-A) and Knickkopf (Knk), are removed from the lumen of *pio* mutants (Figure S5A). These results suggest that the Pio function did not affect airway clearance of Obst-A and Knk and therefore did not play a central role in airway clearance like Wurst. Nevertheless, airway clearance defects observed in TEM images in *pio* null mutant embryos and, in addition, defective tube lumen morphology in *wurst;pio* transheterozygous mutant embryos explain the occurrence of airway gas filling defects.”

5. Pio and DumpyDumpy (Dpy) is another ZP domain protein secreted by the tracheal cells and detected in the lumen. To follow Dpy distribution, Drees and colleagues used a Dpy::eYFP protein trap line, the same used in Dong et al. However, in this latter paper, Dong et al. stated, using a Crb staining, that Dpy is not at the apical cell surface but only in the lumen. However, Drees and colleagues reported (line 227 and Figure 4C) that Dpy appears both at the apical cell surface and in the lumen of the tracheal system. But they did not show a co-localisation with an apical marker.Furthermore, in their previous work, (Drees et al. 2019) they called the apical staining a "peripheral shell" layer. In addition, in S2R+ cell culture, it is only when Pio and Dpy co-express that Dpy is detected at the cell membrane. The in vivo localisation of Dpy is an important point that needs to be clarified as it is of importance for the final model proposed Supp Figure 9. Drees at al. also performed FRAP experiments on Dpy::eYFP protein trap embryos. As excepted as already shown by Dong et al.

The referee is correct, we state “In stage 16 embryos *Dpy::eYFP* (Lye et al., 2014) appears at the tracheal apical cell surface and predominantly within the lumen (Figure 4C).” The corresponding Figure 4C reveals Dumpy::eYFP staining overlapping with chitin at two subcellular regions: Dpy is enriched as a core-like structure within the lumen overlapping with the chitin cable of the control embryos. Additionally, *Dpy::eYFP* overlaps with the chitin part that might be part of the apical cell surface. But this observation is hard to see in images in Figure 4C and we apologize it. We therefore repeated the *Dpy::eYFP* localization analysis and analyzed in more detail with the ZEN profile tools, which shows peak fluorescence pixel intensities of different channels and provides the possibility to prove, if they overlap in XY axis.

We asked first, if cbp (chitin) appears at the apical surface of dorsal trunk cells, when Pio becomes cleaved and released. In mid stage 16 embryos cbp staining appeared in the luminal chitin cable and additionally in a distinctive pattern, which fits to the pattern of taenidial folds that start to form. We therefore used the apical cell membrane marker Crumbs to co-stain cbp. Airycsan microscopy fluorescence intensity profile analysis and corresponding close ups images confirmed the overlap of Crb and cbp stainings at this distinctive pattern indicating this shows the chitin matrix at the apical cell surface (Figure S8A). But there was no overlap of cbp and Crb at the chitin cable structure. Thus, knowing the localization of the apical cell surface chitin matrix, we performed co-stainings of cbp with mCherry::Pio (RFP antibody). This revealed, as expected, overlap of cbp and RFP antibody staining at the apical cell surface chitin matrix (distinct pattern) and with the luminal chitin-cable (Figure S8B,C). Finally we repeated the stainings and analysis with cbp, mCherry::Pio (RFP antibody) and Dpy::eYFP (GFP antibody). First, these results revealed overlap of Dpy::eYFP and cbp at the apical cell surface and in the tube lumen (Figure S8D) and second, overlap of punctuate staining of Dpy::eYFP, cbp and mCherry::Pio at the apical cell surface chitin matrix and also at the luminal chitin cable (Figure S8E). Very obvious from images and Z-projection in Figure 4C is the lack of extracellular *Dpy::eYFP* staining in *pio* mutant embryos. *Dpy::eYFP* enriched intracellularly, and thus, the *pio* mutant caused *Dpy::eYFP* mis-expression fits well to our results from S2R+ cell culture. As the reviewer notes, it is only when Pio and Dpy co-express that Dpy is detected at the cell membrane. Altogether, Figure 4C, cell culture experiments and our new stainings support our model, that Pio and Dumpy interact and are co-secreted at the apical cell membrane/surface, where Np mediates Pio cleavage. As requested by reviewer 2, we moved the model to Figure 9. As requested by reviewer 1, we extended the model for timing events.

A minor point, the Dpy::eYFP protein trap line used in this study is not listed in the Materials and methods section of the supplementary data.

Thanks, we included it into the List of sources (Supplement). This YFP-trap line (called CPTI lines) was published by Claire M. Lye et al., Development, 141, 2014. We cite it in our manuscript.

6. The serine protease NP and Pio release.Drees and colleagues have pervious shown, preforming in vitro studies, that protease Notopleural (Np) cleaves the Pio ZP domain (Drees at al. 2019). Here the authors went a step further in demonstrating that it is also true in vivo at stage 17. In addition, they showed that, in Np mutant embryos, mCherry::Pio is mostly detected within tracheal cells and the luminal staining is strongly reduced. In this mutant context, the authors conducted FRAP experiment on the mCherry::Pio signal even very weak in the lumen. They showed hardly no recovery after photobleaching.In *Drosophila* S2 cells, Drees and colleagues showed that co-expression of the catalytically inactive NpS990A with mCherry::Pio in showed as a prominent signal the 90kDa mCherry::Pio variant in the cell lysate (Figure 5B), and live imaging revealed mCherry::Pio localisation at the cell surface (Figure S6B). However, in this inactive form context, a strong signal is also detected at 60kDA corresponding to a cleaved form of the Pio ZP domain (Figure 5B), and Pio localisation at the cell surface appears weaker than in controls. They authors did not consider that another protease could be at play.On the other hand, in their previous work, Drees et al. identified a mutant form of Pio(PioR196A) which is resistant to NP cleavage in vitro. It will be a step forward to establish by CRISPR/cas9, as the authors seems to be successful with this technique, a mutant line carrying this point mutation. It will be important to determine whether the observed phenotype resembles that of a mutant Np phenotype.In their previous work (PLOS Genetics 2019), in Np mutant embryos, Drees et al. did not report "budge-like" deformations from stage 16 onwards leading to the detachment of the tracheal cell from their adjacent aECM. Either the alleles or the allelic combination is different between the two studies which could explain this difference, or it is a new phenotype that has not been previously described. In the latter case, it becomes important to quantify the proportion of segments showing these bubbles. Is this a rare phenotype to observe?

We thank the reviewer for the very interesting comments and the careful reading of our manuscripts and the very useful suggestions. We agree, we cannot exclude the possibility that another protease is involved in the cleavage of Pio. Therefore, we included this important point in the Discussion section as follows:

“Unknown proteases may likely be involved in Pio processing since cleaved mCherry::Pio is also detectable in inactive NpS990A cells.”

We think the generation of the *pioR196A* mutant to address Pio localization and tracheal phenotypes is a great idea, which we would like to address in future experiments. Unfortunately, the production of this fly line with such a specific point mutation at this position will take several months, not included the subsequent evaluation and phenotypic analysis of this fly line and mutants. Therefore, we apologize that we cannot pursue this question experimentally.

Nevertheless, mentioning the possibility and the requirement of such an experiment is important and we discuss it as follows:

“Previously we identified a mutation at the Pio ZP domain (R196A) resistant to NP cleavage in cell culture experiments (Drees et al., 2019). Establishing a corresponding mutant fly line would be essential in determining whether the observed phenotype resembles the phenotype of the *Np* mutant embryos.”

However, knowing that we are not able to provide a new mutant fly line to evaluate the formation of the dorsal tube when an NP non-cleavable form of Pio is expressed, we sought to use an alternative approach by overexpressing Np in the trachea with btl-Gal4. This shows a clear pairing of Np overexpression and Pio release specifically at stage 16 dorsal trunk and associated tube overexpansion.

Finally, the reviewer is correct, we did not mention the appearance of bulges in Np mutant tracheal dorsal trunk cells in our previous publication. We used that same Np alleles in 2019 and a closer look at the publication of 2019 likewise shows the appearance of bulges in Np mutant embryos, e.g. Figure 1B (red-dextran, left part of the tracheal lumen shows bulges) and even the Dpy::YFP matrix tear off at the site of bulges (Figure 4F’’, above the arrowhead). But we did not know at the time the link with Pio and Dumpy.

However, we agree, it is important to know more about the appearance of the phenotype by means of quantifications. The quantifications of bulges per dorsal trunk (n=16) is shown in Figure 7B.

7. Minor point: I don't understand what the authors are trying to show in supplementary Figure 8. Tracheal cells detach and are found in the lumen?

We are sorry for the unclear description in the legend. We corrected it as follows in the legend of Figure S12:

“This indicates disintegration of apical cell membrane at bulges and subsequent leaking of cellular content into the lumen.”

8. Np function conserved matriptase.In this work, Drees and colleagues showed that Np controls in vivo the cleavage of the Pio ZP domain.Dumpy and Piopio are not conserved in vertebrates but they both contain a ZP domain which is conserved. The authors tested if other ZP proteins can be cleaved by Np or the human homolog Matriptase. The authors tested in cell culture the ability of the type III Transforming growth factor-β receptor which contains a ZP domain to be cleaved either by Np or Matriptase. This could be a general mechanism that needs to be extended to other ZP domain proteins and that could be at play to structure the matrix and give it its physical properties.However, as it is all speculative, I find the Discussion section related to these data, for too long and that does not help to understand better the work done in the formation of the tracheal tubes of the *Drosophila* embryo.

We show that Np mediates cleavage of the Pio ZP domain in vitro and in vivo in *Drosophila* embryos. We further showed that also the human matriptase was able to cleave the Pio ZP domain. To understand if this is a more general mechanism, we extended our studies with the human TβIII and its ZP domain. These data show that both *Drosophila* and human matriptases are able to cleave ZP domains of different proteins from different species. These data suggest that Matriptase-mediated ZP domain cleavage is not a *Drosophila* specific mechanism. We cannot follow the argumentation of the referee to state it all speculative. Nevertheless, we agree that it will need follow up studies to show that the mechanism is more general than two different species and ZP domain proteins. Anyway, as requested by the referee, we deleted the following sentences of the paragraph, since they are speculative in the context of our manuscript and do not directly describe a potential matriptase and ZP domain function:

“Matriptase degrades receptors and ECM in pulmonary fibrinogenesis in squamous cell carcinoma (Bardou et al., 2016; Martin and List, 2019). TβRIII is a membrane-bound proteoglycan that generates a soluble form upon shedding (López-Casillas et al., 1991), a potent neutralizing agent of TGF-β. Expression of the soluble TβRIII inhibits tumor growth due to the inhibition of angiogenesis (Bandyopadhyay et al., 2002). Idiopathic pulmonary fibrosis (IPF) is associated with a progressive loss of lung function due to fibroblast accumulation and relentless ECM deposition (King et al., 2011; Loomis-King et al., 2013). “

However, the comparisons of the tubular organ and the phenotypic expressions of the bulging membrane and the aortic aneurysm appear to us as an important element of the article. In both cases, cell membrane loses its integrity and can break in tubular networks. Thus, with our findings on the modification of extracellular ZP proteins, we offer a potential new molecular approach even for clinical investigation.

9.Minor points: Pio and cytoskeleton organisation.Line 78-79, the authors wrongly quoted a work from Brodu et al. (2010). Pio does not anchor the microtubule severing enzyme Spastin. Instead, Spastin releases the microtubule-organising centre from its centrosomal location, then Pio contributes to its apical membrane anchoring. It can therefore be assumed that the organisation of the microtubule network is affected in a pio null mutant. In addition, ZP proteins have been shown to link the aECM to the actin cytoskeleton. Therefore, it would be interesting to look at the organisation of the actin and microtubule cytoskeletons in a pio mutant context in which enlarged apical cell surface area are observed.

We are very thankful for finding this mistake in the introduction. We corrected it as follows: “Further, Pio is involved in relocating microtubule organizing center components γ-TuRC (γtubulin and Grips; γ-tubulin ring proteins). This requires Spastin-mediated release from the centrosome and Pio-mediated γ-TuRC anchoring in the apical membrane.”

Studying cytoskeleton in pio mutant embryos is a helpful idea. Therefore, we analyzed F-actin with Phalloidin and β tubulin (E7 antibody, DSHB) in the dorsal trunk cells of stage 16 control and *pio* mutant embryos. However, tracheal cells are tiny and only gross changes can be realized. The confocal Z-stack analysis of the stainings did not show gross differences between control and *pio* mutant embryos. We observe the expected apical subcortical accumulation for the actin and tubulin cytoskeleton in dorsal trunk cells of *pio* stage 16 mutant embryos which also has been shown for wt embryos elsewhere. These new data are presented in the supplement Figure S7.

Referees cross-commentingI have just read the comments of the other two reviewers, who like me are specialists in the formation of the tracheal system in the *Drosophila* embryo.I find the comments very fair and balanced. They are in the same spirit as my comments and are very complementary. I hope that all our comments will be constructive for the authors and will improve the quality of their work.Reviewer #3 (Significance (Required)):Overall, the methodology is sound, the quality of the data is good and the paper is very well written. Authors combine in vivo, in vitro studies as well a cell culture approach. Using CRISPR/Cas9, they generated a large number of new tools allowing in vivo studies. Drees and colleagues generated new alleles of pio which are lack of function alleles. They described a new phenotype for pio mutant embryos, namely over-elongated tubes. But they authors do not comment on why these new alleles reveal a new phenotype. Furthermore, using their piomCherry::pio line, the authors state that Pio is localised to the plasma membrane. This location is very difficult to assess. Both new results require clarification.The authors had already demonstrated that Np cleaves the ZP domain of Pio in vitro. Here they demonstrate this in vivo. It appears important to evaluate the formation of the dorsal tube when an NP non-cleavable form of Pio is expressed.Finally, the model proposing a coupling between the extracellular matrix and the membrane of tracheal cells is very interesting. The demonstration that cleavage of Pio by Np could participate in this coupling is very interesting for those interested in the integration of mechanical stress and cellular deformation. However, such a model has already been discussed in Dong et al. (2014). In this article, Dong et al. proposed that a "coupling of the apical membrane and Dpy matrix core is essential for tube length regulation".The audience for this article should be specialised and oriented towards basic research. It may be of interest to people working on tubular systems or working on ZP proteins.My field of expertise is cell biology and developmental biology in *Drosophila* and formation of tubular networks.